# Unleashing the Potential of Fractional Calculus in Graph Neural Networks with FROND

**Qiyu Kang**[1][*][†] **, Kai Zhao**[1][*]**, Qinxu Ding**[2]**, Feng Ji**[1]**, Xuhao Li**[3]**, Wenfei Liang**[1]**, Yang Song**[4]**,
Wee Peng Tay**[1]
[1]Nanyang Technological University [2]Singapore University of Social Sciences
[3]Anhui University [4]C3 AI, Singapore

## Abstract

We introduce the FRactional-Order graph Neural Dynamical network (FROND), a new continuous graph neural network (GNN) framework. Unlike traditional continuous GNNs that rely on integer-order differential equations, FROND employs the Caputo fractional derivative to leverage the non-local properties of fractional calculus. This approach enables the capture of long-term dependencies in feature updates, moving beyond the Markovian update mechanisms in conventional integer-order models and offering enhanced capabilities in graph representation learning. We offer an interpretation of the node feature updating process in FROND from a non-Markovian random walk perspective when the feature updating is particularly governed by a diffusion process. We demonstrate analytically that oversmoothing can be mitigated in this setting. Experimentally, we validate the FROND framework by comparing the fractional adaptations of various established integer-order continuous GNNs, demonstrating their consistently improved performance and underscoring the framework's potential as an effective extension to enhance traditional continuous GNNs. The code is available at `https://github.com/zknus/ICLR2024-FROND`.

## 1 Introduction

Graph Neural Networks (GNNs) have excelled in diverse domains, e.g., chemistry (Yue et al., 2019), finance (Ashoor et al., 2020), and social media (Kipf & Welling, 2017; Zhang et al., 2022; Wu et al., 2021). The message passing scheme (Feng et al., 2022), where features are aggregated along edges and iteratively propagated through layers, is crucial for the success of GNNs. Over the past few years, numerous types of GNNs have been proposed, including Graph Convolutional Networks (GCN) (Kipf & Welling, 2017), Graph Attention Networks (GAT) (Veličković et al., 2018), and GraphSAGE (Hamilton et al., 2017). Recent works, such as (Chamberlain et al., 2021c; Thorpe et al., 2022; Rusch et al., 2022; Song et al., 2022; Choi et al., 2023; Zhao et al., 2023a; Kang et al., 2023), have incorporated various continuous dynamical processes to propagate information over graph nodes, giving rise to a class of continuous GNNs based on integer-order differential equations. These continuous models have demonstrated notable performance, for instance, in enhancing robustness and addressing heterophilic graphs (Han et al., 2023).

Within these integer-order continuous GNNs, the differential operator $\mathrm{d}^\beta / \mathrm{d}t^\beta$ has been constrained to *integer values* of $\beta$, primarily 1 or 2. However, over recent decades, the wider scientific community has explored fractional-order differential operators, where $\beta$ can be any *real number*. These expansions have proven pivotal in various applications characterized by non-local and memory-dependent behaviors, with examples including viscoelastic materials (Bagley & Torvik, 1983), anomalous transport mechanisms (Gómez-Aguilar et al., 2016), and fractal media (Mandelbrot & Mandelbrot, 1982). Unlike conventional integer-order derivatives that measure the function's *instantaneous rate of change* and focus on the local vicinity, fractional-order derivatives (Tarasov, 2011) consider *the entire historical trajectory of the function*.

---

[*]First two authors contributed equally. [†]Correspondence to: Qiyu Kang <kang0080@e.ntu.edu.sg>.

We introduce the FRactional-Order graph Neural Dynamical network (FROND) framework, a new approach that broadens the capabilities of traditional integer-order continuous GNNs by incorporating fractional calculus. It naturally generalizes the integer-order derivative $\mathrm{d}^\beta / \mathrm{d}t^\beta$ in these GNNs to accommodate any positive real number $\beta$. This modification gives FROND the ability to incorporate *memory-dependent dynamics* for information propagation and feature updating, enabling refined graph representations and improved performance potentially. Importantly, this technique assures at least equivalent performance to integer-order models, as setting $\beta$ to integer values reverts the models to their traditional integer-order forms.

Several works like (Maskey et al., 2023) have combined fractional graph shift operators with integer-order ordinary differential equations (ODEs). These studies are distinct from our research, wherein we focus on incorporating time-fractional derivatives for updating graph node features, modeled as a memory-inclusive dynamical process. Other works like (Liu et al., 2022) have used fractional calculus in gradient propagation for the training process, which is different from leveraging fractional differential equations (FDEs) in modeling the node feature updating. We provide a detailed discussion of the differences between FROND and these works in Appendix A.

Many real-world graph datasets, such as the World Wide Web, the Internet, and various biological and social networks, are known to exhibit scale-free hierarchical structures. These structures suggest a pervasive self-similarity across different scales, hinting at an underlying fractal behavior (Song et al., 2005; Kim et al., 2007; Masters, 2004). It has been well-established that dynamical processes with self-similarity on such fractal media are more accurately described using FDEs. For instance, the dispersion of heat or mass over these structures is best modeled using fractional diffusion equations (Diaz-Diaz & Estrada, 2022). Further investigations have revealed a direct connection between the fractal dimension of these structures and the order $\beta$ in fractional derivatives $\mathrm{d}^\beta / \mathrm{d}t^\beta$ (Nigmatullin, 1992; Tarasov, 2011). This revelation births a compelling insight: the optimal $\beta$ in our models, which may differ from integers, can pave the way for enhanced node classification and potentially unearth insights into the inherent "fractality" of graph datasets.

**Main contributions.** Our objective in this paper is to formulate a generalized fractional-order continuous GNN framework. Our key contributions are summarized as follows:

- We propose a novel, generalized continuous GNN framework that incorporates non-local fractional derivatives $\mathrm{d}^\beta / \mathrm{d}t^\beta$. This framework generalizes the prior class of integer-order continuous GNNs, subsuming them as special instances with $\beta$ setting as integers. This approach also lays the groundwork for a diverse new class of GNNs that can accommodate a broad array of learnable memory-dependent feature-updating processes.

- We provide an interpretation from the perspective of a non-Markovian graph random walk when the feature-updating dynamics are inspired by the fractional heat diffusion process. Contrasting with the Markovian random walk implicit in traditional integer-order graph neural diffusion models whose convergence to the stationary equilibrium is exponentially swift, we establish that in FROND, convergence follows a slow algebraic rate. This characteristic enhances FROND's ability to mitigate oversmoothing, as verified by our experimental results.

- We underscore the compatibility of FROND, emphasizing its capability to be seamlessly integrated to augment the performance of existing integer-order continuous GNNs across diverse datasets. Our exhaustive experiments, encompassing the fractional differential extension of (Chamberlain et al., 2021c; Thorpe et al., 2022; Rusch et al., 2022; Song et al., 2022; Choi et al., 2023; Zhao et al., 2023a), substantiate this claim. Through detailed ablation studies, we provide insights into the choice of numerical schemes and parameters.

## 2 PRELIMINARIES

In this section, we briefly introduce fractional calculus and integer-order continuous GNNs. For a comprehensive review of fractional calculus, readers are referred to Appendix B.

### 2.1 CAPUTO FRACTIONAL DERIVATIVE

The literature offers various fractional derivative definitions, notably by Riemann, Liouville, Chapman, and Caputo (Tarasov, 2011). Our study leverages the *Caputo* fractional derivative, due to the reasons listed in Appendix B.4. The traditional first-order derivative of a scalar function $f(t)$ represents the

local rate of change of the function at a point, defined as: $\frac{\mathrm{d}f(t)}{\mathrm{d}t} = \lim_{\Delta t \to 0} \frac{f(t+\Delta t)-f(t)}{\Delta t}$. Let $F(s)$ denote the Laplace transform of $f(t)$, assumed to exist on $[s_0, \infty)$ for some $s_0 \in \mathbb{R}$. Under certain conditions (Korn & Korn, 2000), the Laplace transform of $\frac{\mathrm{d}f(t)}{\mathrm{d}t}$ is given by:

$$\mathcal{L}\left\{\frac{\mathrm{d}f(t)}{\mathrm{d}t}\right\} = sF(s) - f(0) \tag{1}$$

The Caputo fractional derivative of order $\beta \in (0, 1]$ for a function $f(t)$ is defined as follows:

$$D_t^\beta f(t) = \frac{1}{\Gamma(1-\beta)} \int_0^t (t-\tau)^{-\beta} f'(\tau)\, \mathrm{d}\tau, \tag{2}$$

where $\Gamma(\cdot)$ denotes the gamma function, and $f'(\tau)$ is the first-order derivative of $f$. The broader definition for any $\beta > 0$ is deferred to Appendix B. The Caputo fractional derivative inherently integrates the entire history of the system through the integral term, emphasizing its non-local nature. For $s > \max\{0, s_0\}$, the Laplace transform of the Caputo fractional derivative is given by (Diethelm, 2010)[Theorem 7.1]:

$$\mathcal{L}\left\{D_t^\beta f(t)\right\} = s^\beta F(s) - s^{\beta-1} f(0). \tag{3}$$

Comparing (1) and (3), it is evident that the Caputo derivative serves as a generalization of the first-order derivative. The alteration in the exponent of $s$ comes from the memory-dependent property in (2). As $\beta \to 1$, the Laplace transform of the Caputo fractional derivative converges to that of the traditional first-order derivative. When $\beta = 1$, $D_t^1 f = f'$ is uniquely determined through the inverse Laplace transform (Cohen, 2007).

In summary, from the frequency domain using the Laplace transform, we observe that the Caputo fractional derivative can be seen as a natural extension of the traditional first-order derivative. For vector-valued functions, the fractional derivative is defined component-wise for each dimension.

## 2.2 INTEGER-ORDER CONTINUOUS GNNS

We denote an undirected graph as $\mathcal{G} = (\mathcal{V}, \mathbf{W})$ without self-loops, where $\mathcal{V}$ is the set of $|\mathcal{V}| = N$ nodes. The feature matrix $\mathbf{X} = \left([\mathbf{x}_1]^\mathsf{T}, \cdots, [\mathbf{x}_N]^\mathsf{T}\right)^\mathsf{T} \in \mathbb{R}^{N \times d}$ consists of rows $\mathbf{x}_i \in \mathbb{R}^d$ as node feature vectors and $i$ is the node index. The $N \times N$ matrix $\mathbf{W} := (W_{ij})$ has elements $W_{ij}$ indicating the edge weight between the $i$-th and $j$-th node with $W_{ij} = W_{ji}$. The following integer-order continuous GNNs leverage ODEs to facilitate information propagation amongst graph nodes, where features evolve as $\mathbf{X}(t)$, starting from the initial condition $\mathbf{X}(0) = \mathbf{X}$.

**GRAND:** Inspired by the heat diffusion equation, GRAND (Chamberlain et al., 2021c) utilizes the following nonlinear autonomous dynamical system:

$$\frac{\mathrm{d}\mathbf{X}(t)}{\mathrm{d}t} = (\mathbf{A}(\mathbf{X}(t)) - \mathbf{I})\mathbf{X}(t). \tag{4}$$

where $\mathbf{A}(\mathbf{X}(t)) \in \mathbb{R}^{N \times N}$ is a learnable, time-variant attention matrix, calculated using the features $\mathbf{X}(t)$, and $\mathbf{I}$ denotes the identity matrix. The feature update outlined in (4) is referred to as the **GRAND-nl** version (due to the nonlinearity in $\mathbf{A}(\mathbf{X}(t))$). We define $d_i = \sum_{j=1}^n W_{ij}$ and let $\mathbf{D}$ be a diagonal matrix with $D_{ii} = d_i$. The *random walk Laplacian* is then represented as $\mathbf{L} = \mathbf{I} - \mathbf{W}\mathbf{D}^{-1}$. In a simplified context, we employ the following linear dynamical system:

$$\frac{\mathrm{d}\mathbf{X}(t)}{\mathrm{d}t} = (\mathbf{W}\mathbf{D}^{-1} - \mathbf{I})\mathbf{X}(t) = -\mathbf{L}\mathbf{X}(t). \tag{5}$$

The feature update process in (5) is the **GRAND-l** version. For implementations of (5), one may direct set $\mathbf{W}\mathbf{D}^{-1} = \mathbf{A}(\mathbf{X}(0))$ as a column-stochastic attention matrix, rather than using a plain weight. Notably, in this time-invariant setting, the attention weight matrix, reliant on the initial node features, stays unchanged throughout the feature evolution period.

**GRAND++** (Thorpe et al., 2022) adds a source term to GRAND, enhancing learning in scenarios with limited labeled nodes. **GraphCON** (Rusch et al., 2022) employs a second-order ODE, which is equivalent to two first-order ODEs, drawing inspiration from oscillator systems. **CDE** (Zhao et al., 2023a) incorporates convection-diffusion equations into GNNs to address heterophilic graph challenges. **GREAD** (Choi et al., 2023) introduces a reaction term in the GRAND model, improving its application to heterophilic graphs and formulating a diffusion-reaction equation within GNNs. The detailed formulation for each model is presented in Appendix E.1 due to space constraints.

## 3    FRACTIONAL-ORDER GRAPH NEURAL DYNAMICAL NETWORK

In this section, we introduce the FROND framework, a novel approach that augments traditional integer-order continuous GNNs by incorporating fractional calculus. We elucidate the fractional counterparts of several well-established integer-order continuous GNNs, including GRAND, GRAND++, GraphCON, CDE, and GREAD, as referenced in Section 2.2. We provide a detailed study of the fractional extension of GRAND, and present insights into the inherent memory mechanisms in our framework through a random walk interpretation. Our theoretical findings suggest a potential mitigation of oversmoothing due to the model's slow algebraic convergence to stationarity. Subsequently, we outline the numerical FDE solvers required to implement FROND.

### 3.1    FRAMEWORK

Consider a graph $\mathcal{G} = (\mathcal{V}, \mathbf{W})$ as defined in Section 2.2. Analogous to the implementation in traditional integer-order continuous GNNs, a preliminary learnable encoder function $\varphi : \mathcal{V} \to \mathbb{R}^d$ that maps each node to a feature vector can be applied. Stacking all the feature vectors together, we obtain $\mathbf{X} \in \mathbb{R}^{N \times d}$. Employing the Caputo fractional derivative outlined in Section 2.1, the information propagation and feature updating dynamics in FROND are characterized by the following FDE:

$$D_t^\beta \mathbf{X}(t) = \mathcal{F}(\mathbf{W}, \mathbf{X}(t)), \quad \beta > 0, \tag{6}$$

where $\beta$ denotes the fractional order of the derivative, and $\mathcal{F}$ is a dynamic operator on the graph like the models presented in Section 2.2. The initial condition for (6) is set as $\mathbf{X}^{[\lceil \beta \rceil - 1]}(0) = \ldots = \mathbf{X}(0) = \mathbf{X}$ consisting of the preliminary node features, with $\mathbf{X}^{[i]}(t)$ denoting the $i$-th order derivative and $\lceil \beta \rceil$ is the smallest integer not less than $\beta$, akin to the initial conditions seen in integer-order ODEs.[1] Similar to integer-order continuous GNNs, we set an integration time parameter $T$ to get $\mathbf{X}(T)$. The final node embeddings for downstream tasks are then decoded using a learnable decoder $\psi(\mathbf{X}(T))$.

When $\beta = 1$, (6) reverts to the class of integer-order continuous GNNs, with the infinitesimal variation of features dependent only on their present state. Conversely, when $\beta < 1$, the Caputo fractional derivative (2) dictates that the updating process for features encompasses their entire history, not just the present state. This paradigm facilitates memory-dependent dynamics in the framework.

For further insights into memory dependence, readers are directed to Section 3.3, which discusses time discretization techniques for numerically solving the system. It illustrates how, akin to integer-order neural ODE models, time consistently acts as an analog to the layer index and how the nonlocal properties of fractional derivatives facilitate nontrivial dense or skip connections between layers. In Section 3.2, when the dynamic operator $\mathcal{F}$ is designated as the diffusion process in (5), we offer a memory-dependent *non-Markovian* random walk interpretation of the fractional graph neural diffusion process. Here, as $\beta \to 1$, the non-Markovian random walk increasingly detaches from the path history, becoming a Markovian walk at $\beta = 1$, which is related to the normal diffusion process (Thorpe et al., 2022). The parameter $\beta$ provides flexibility to adjust the extent of memorized dynamics embedded in the framework. From a geometric perspective, as discussed in Section 1, the information propagation dynamics in fractal graph datasets might be more suitably described using FDEs. Choosing a non-integer $\beta$ could reveal the degree of fractality in graph datasets.

### 3.1.1    FROND MODEL EXAMPLES

When $\mathcal{F}$ in (6) is specified to the dynamics depicted in various integer-order continuous GNNs (cf. Section 2.2), we formulate FROND GNN variants such as F-GRAND, F-GRAND++, F-GREAD, F-CDE, and F-GraphCON, serving as fractional differential extensions of the original GNNs.

**F-GRAND**: Mirroring the GRAND model, the fractional-GRAND (F-GRAND) has two versions. The F-GRAND-nl version employs a time-variant FDE as follows:

$$D_t^\beta \mathbf{X}(t) = (\mathbf{A}(\mathbf{X}(t)) - \mathbf{I})\mathbf{X}(t), \quad 0 < \beta \leq 1. \tag{7}$$

---

[1] See Appendix B.3.2. We mainly consider $\beta \in (0, 1]$ and the initial condition is $\mathbf{X}(0) = \mathbf{X}$.

It is computed using $\mathbf{X}(t)$ and the attention mechanism derived from the Transformer model (Vaswani et al., 2017). The entries of $\mathbf{A}(\mathbf{X}(t)) = (a(\mathbf{x}_i, \mathbf{x}_j))$ are given by:

$$a(\mathbf{x}_i, \mathbf{x}_j) = \mathrm{softmax}\left(\left\{\frac{(\mathbf{W}_K \mathbf{x}_i^\mathsf{T})^\mathsf{T} \mathbf{W}_Q \mathbf{x}_j^\mathsf{T}}{\bar{d}_k}\right\}\right). \tag{8}$$

In this formulation, $\mathbf{W}_K$ and $\mathbf{W}_Q$ are the learned matrices, and $\bar{d}_k$ signifies a hyperparameter related to the dimensionality of $\mathbf{W}_K$. In parallel, the F-GRAND-l version stands as the fractional differential extension of (5):

$$D_t^\beta \mathbf{X}(t) = -\mathbf{L}\mathbf{X}(t), \quad 0 < \beta \leq 1. \tag{9}$$

Recall that the initial condition for F-GRAND-nl and F-GRAND-l is $\mathbf{X}(0) = \mathbf{X}$ due to $\beta \in (0, 1]$.

**F-GRAND++, F-GREAD, F-CDE, and F-GraphCON:** Due to space constraints, we direct the reader to Appendix E for detailed formulations. Succinctly, they represent the fractional differential extensions of GRAND++, GraphCON, CDE, and GREAD. To highlight FROND's compatibility and its potential to enhance the performance of existing integer-order continuous GNNs across a variety of datasets, exhaustive experiments are provided in Section 4 and Appendix E.

## 3.2 Random Walk Perspective of F-GRAND-l

The established Markov interpretation of GRAND-l (5), as outlined in (Thorpe et al., 2022), aligns with F-GRAND-l (9) when $\beta = 1$. We herein broaden this interpretation to encompass a non-Markovian random walk that considers the walker's complete path history when $\beta$ is a non-integer, thereby elucidating the memory effects inherent in FROND. In contrast to the Markovian walk, whose distribution converges exponentially to equilibrium, our strategy assures algebraic convergence, revealing F-GRAND-l's efficacy in mitigating oversmoothing as evidenced in Section 4.3.

To begin, we discretize the time domain into time instants as $t_n = n\sigma, \sigma > 0, n = 0, 1, 2, \ldots$, where $\sigma$ is assumed to be small enough to ensure the validity of the approximation. Let $\mathbf{R}(t_n)$ be a random walk on the graph nodes $\{\mathbf{x}_j\}_{j=1}^N$ that is not necessarily a Markov process and $\mathbf{R}(t_{n+1})$ may depend on the path history $(\mathbf{R}(t_0), \mathbf{R}(t_1), \ldots, \mathbf{R}(t_n))$ of the random walker. For convenience, we introduce the coefficients $c_k$ for $k \geq 1$ and $b_n$ for $n \geq 0$ from (Gorenflo et al., 2002), which are used later to define the random walk transition probability:

$$c_k(\beta) = (-1)^{k+1}\binom{\beta}{k} = \left|\binom{\beta}{k}\right|, \quad b_n(\beta) = \sum_{k=0}^n (-1)^k \binom{\beta}{k}, \tag{10}$$

where the generalized binomial coefficient $\binom{\beta}{k} = \frac{\Gamma(\beta+1)}{\Gamma(k+1)\Gamma(\beta-k+1)}$ and the gamma function $\Gamma(\cdot)$ are employed in the definition of the coefficients. The sequences $c_k$ and $b_n$ consist of positive numbers, not greater than 1, decreasing strictly monotonically to zero (see supplementary material for details) and satisfy $\sum_{k=1}^n c_k + b_n = 1$. Using these coefficients, we define the transition probabilities of the random walk starting from $\mathbf{x}_{j_0}$ as

$$\mathbb{P}\big(\mathbf{R}(t_{n+1}) = \mathbf{x}_{j_{n+1}} \,\big|\, \mathbf{R}(t_0) = \mathbf{x}_{j_0}, \mathbf{R}(t_1) = \mathbf{x}_{j_1}, \ldots, \mathbf{R}(t_n) = \mathbf{x}_{j_n}\big)$$

$$= \begin{cases} c_1 - \sigma^\beta & \text{if staying at current location with } j_{n+1} = j_n, \\ \sigma^\beta \frac{W_{j_n j_{n+1}}}{d_{j_n}} & \text{if jumping to neighboring nodes with } j_{n+1} \neq j_n, \\ c_{n+1-k} & \text{if revisiting historical positions with } j_{n+1} = j_k, 1 \leq k \leq n-1, \\ b_n & \text{if revisiting historical positions with } j_{n+1} = j_0. \end{cases} \tag{11}$$

This formulation integrates memory effects, considering the walker's time, position, and path history. The transition mechanism of the memory-inclusive random walk between $t_n$ and $t_{n+1}$ is elucidated as follows: Suppose the walker is at node $j_n$ at time $t_n$, having a full path history $(j_0, j_1, \ldots, j_n)$. We generate a random number $\rho \in [0, 1)$ uniformly, and divide the interval $[0, 1)$ into adjacent sub-intervals with lengths $c_1, c_2, \ldots, c_n, b_n$. We further subdivide the first interval (with length $c_1$) into sub-intervals of lengths $c_1 - \sigma^\beta$ and $\sigma^\beta$.

1. If $\rho$ is in the first interval with length $c_1$, the walker either moves to a neighbor $j_{n+1} = k$ with probability $\sigma^\beta \frac{W_{j_n k}}{d_{j_n}}$ or remains at the current position with probability $c_1 - \sigma^\beta$.

2. For $\rho$ in subsequent intervals, the walker jumps to a previously visited node in the history $(j_0, j_1, \ldots, j_{n-1})$, specifically, to $j_{n+1-k}$ if in $c_k$, or to $j_0$ if in $b_n$.

When $\beta < 1$, the random walk can, with positive probability, revisit its history, restricting extensive drift. We denote $\mathbb{P}(\mathbf{R}(t_n))$ as the probability column vector, with its $j$-th element given as $\mathbb{P}(\mathbf{R}(t_n) = \mathbf{x}_j)$. Additionally, we specify ${}_i\mathbb{P}(\mathbf{R}(t_n))$ to indicate the situation where the random walker initiates from the $i$-th node, i.e., $\mathbf{R}(0) = \mathbf{x}_i$, with probability 1. In this case, the initial probability vector ${}_i\mathbb{P}(\mathbf{R}(0))$ is represented as a one-hot vector with the $i$-th entry marked as 1. Using the technique from (Gorenflo et al., 2002), we can prove the following:

**Theorem 1.** *Consider the random walk defined in* (11)*, with the step size $\sigma$ and number of steps $n$. Under the conditions that $n \to \infty$ and $n\sigma = t$, the limiting probability distribution $\mathbf{P}(t) :=$ $\lim_{n\to\infty} \mathbb{P}(\mathbf{R}(t_n))$ satisfies* (9)*. In other words,*

$$D_t^\beta \mathbf{P}(t) = -\mathbf{L}\mathbf{P}(t) \tag{12}$$

Considering that initial conditions and dimensions affect the solutions of FDEs, $\mathbf{P}(t)$ and $\mathbf{X}(t)$ are not equivalent. However, due to the linearity of FDEs, the following conclusion is straightforward:

**Corollary 1.** *Under the conditions that $n \to \infty$ and $n\sigma = t$, we have $\lim_{n\to\infty} \sum_i {}_i\mathbb{P}(\mathbf{R}(t_n))\mathbf{x}_i = \mathbf{X}(t)$, i.e., $\sum_i {}_i\mathbf{P}(t)\mathbf{x}_i = \mathbf{X}(t)$ with ${}_i\mathbf{P}(t) := \lim_{n\to\infty} {}_i\mathbb{P}(\mathbf{R}(t_n))$, where $\mathbf{X}(t)$ is the solution to* (9) *with the initial condition $\mathbf{X}(0) = \mathbf{X}$.*

**Remark 1.** *Theorem 1 and Corollary 1 relate F-GAND-l* (9) *to the non-Markovian random walk in* (11)*, illustrating memory dependence in FROND. As $\beta \to 1$, this process reverts to the Markovian random walk found in GRAND-l (Thorpe et al., 2022) in* (13)*. This underscores the FROND framework's capability to apprehend more complex dynamics than integer-order continuous GNNs.*

$$\mathbb{P}\big(\mathbf{R}(t_{n+1}) = \mathbf{x}_{j_{n+1}} \,\big|\, \mathbf{R}(t_0) = \mathbf{x}_{j_0}, \mathbf{R}(t_1) = \mathbf{x}_{j_1}, \ldots, \mathbf{R}(t_n) = \mathbf{x}_{j_n}\big) \tag{13}$$

$$= \mathbb{P}\big(\mathbf{R}(t_{n+1}) = \mathbf{x}_{j_{n+1}} \,\big|\, \mathbf{R}(t_n) = \mathbf{x}_{j_n}\big) = \begin{cases} 1 - \sigma & \text{\textit{if staying at current location with }} j_{n+1} = j_n \\ \sigma \dfrac{W_{j_n j_{n+1}}}{d_{j_n}} & \text{\textit{if jumping to neighbors with }} j_{n+1} \neq j_n \end{cases}$$

*since we have that all these coefficients vanishing except $c_1 = 1$, i.e.,*

$$c_1 = 1, \quad \lim_{\beta\to 1} c_k(\beta) = 0, \quad k \geq 2, \quad \lim_{\beta\to 1} b_n(\beta) = 0, \quad n \geq 1. \tag{14}$$

### 3.2.1 OVERSMOOTHING MITIGATION OF F-GRAND-L COMPARED TO GRAND-L

The seminal research (Oono & Suzuki, 2020)[Corollary 3. and Remark 1] has highlighted that, when considering a GNN as a layered dynamical system, oversmoothing is a broad expression of the *exponential convergence* to stationary states that only retain information about graph connected components and node degrees. Under certain conditions, the stationary distribution for the Markovian random walk (13) is given by $\boldsymbol{\pi} = (\frac{d_1}{\sum_{j=1}^N d_j}, \ldots, \frac{d_N}{\sum_{j=1}^N d_j})$ (Thorpe et al., 2022), with an *exponentially rapid convergence rate* $\|\mathbb{P}(\mathbf{R}(t_n)) - \boldsymbol{\pi}^{\mathsf{T}}\|_2 \sim O(e^{-r'n})$ [2], where $r' > 0$ relates to the eigenvalues of the matrix $\mathbf{L}$ (Chung, 1997), and $\|\cdot\|_2$ denotes the $\ell^2$ norm. This behavior extends to the continuous limit, akin to a first-order linear ODE solution, exhibiting *exponential convergence* with some $r > 0$:

$$\|\mathbf{P}(t) - \boldsymbol{\pi}^{\mathsf{T}}\|_2 \sim O(e^{-rt}). \tag{15}$$

In contrast, we next prove that the non-Markovian random walk (11) converges to the stationary distribution at a *slow algebraic rate*, thereby helping to mitigate oversmoothing. As $\beta \to 0$, the convergence is expected to be *arbitrarily slow*. In real-world scenarios where we operate within a finite horizon, this slower rate of convergence may be sufficient to alleviate oversmoothing, particularly when it is imperative for a deep model to extract distinctive features instead of achieving exponentially fast convergence to a stationary equilibrium.

**Theorem 2.** *Under the assumption that the graph is strongly connected and aperiodic, the stationary probability for the non-Markovian random walk* (11)*, with $0 < \beta < 1$, is still $\boldsymbol{\pi}$, which is unique. This mirrors the stationary probability of the Markovian random walk as defined by* (13) *when $\beta = 1$. Notably, when $\beta < 1$, the convergence of the distribution (distinct from $\boldsymbol{\pi}$) to $\boldsymbol{\pi}$ is algebraic:*

$$\|\mathbf{P}(t) - \boldsymbol{\pi}^{\mathsf{T}}\|_2 \sim \Theta(t^{-\beta}). \tag{16}$$

**Remark 2.** *Corollary 1 and Theorem 2 indicate that $\mathbf{X}(t) = \sum_i {}_i\mathbf{P}(t)\mathbf{x}_i$, as the solution to F-GRAND-l* (9)*, converges to $\sum_i \boldsymbol{\pi}^{\mathsf{T}}\mathbf{x}_i = \boldsymbol{\pi}^{\mathsf{T}} \sum_i \mathbf{x}_i$ at a slow algebraic rate since $\|{}_i\mathbf{P}(t) - \boldsymbol{\pi}^{\mathsf{T}}\|_2 \sim \Theta(t^{-\beta})$ for all $i$. Notably, $\boldsymbol{\pi}^{\mathsf{T}} \sum_i \mathbf{x}_i$ forms a rank 1 invariant subspace under the dynamics of* (9)*, due to $\boldsymbol{\pi}$ being stationary. This underscores the difference in convergence rates, contrasting the slow algebraic rate in our case with the fast exponential rate (Oono & Suzuki, 2020; Zhao et al., 2023b).*

---

[2]We use the asymptotic order notations from (Notations, 2023) in this paper.

### 3.3 SOLVING FROND

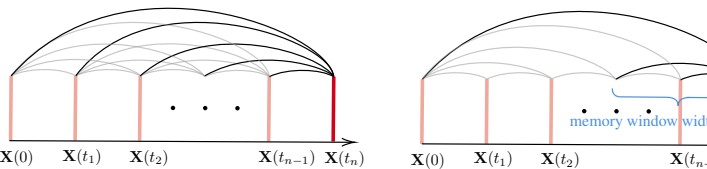

Figure 1: Diagrams of fractional Adams–Bashforth–Moulton method with full (left) and short (right) memory.

The studies by (Chen et al., 2018b; Quaglino et al., 2019; Yan et al., 2018) introduce numerical solvers specifically designed for integer-order neural ODE models. Our research, in contrast, engages with fractional-order ODEs, entities inherently more intricate than integer-order ODEs. To address the scenario *where $\beta$ is non-integer*, we introduce the *fractional explicit Adams–Bashforth–Moulton solver*, incorporating three variants employed in this study: the **basic predictor** discussed in this section, the **predictor-corrector** elaborated in Appendix C.2, and the **short memory principle** detailed in Appendix C.3. Additionally, we present one **implicit L1** solver in Appendix C.4. These methods exemplify how time still acts as a continuous analog to the layer index and elucidate how memory dependence manifests as nontrivial dense or skip connections between layers (see Figs. 1 and 4), stemming from the non-local properties of fractional derivatives.

**Basic Predictor:** We first employ a preliminary numerical solver called "predictor" (Diethelm et al., 2004) through time discretisation. Let $h$ be a small positive discretization parameter. We have

$$
{}_P\mathbf{X}^{(k)} = \sum_{j=0}^{\lceil\beta\rceil-1} \frac{t_k^j}{j!}\mathbf{X}^{[j]}(0) + \frac{1}{\Gamma(\beta)}\sum_{j=0}^{k-1}\mu_{j,k}\mathcal{F}(\mathbf{W}, \mathbf{X}^{(j)}), \tag{17}
$$

where $\mu_{j,k} = \frac{h^\beta}{\beta}\left((k-j)^\beta - (k-1-j)^\beta\right)$, $k$ denotes the discrete time index (iteration), and $t_k = kh$ represents the discretized time steps. $\mathbf{X}^{(k)}$ is the numerical approximation of $\mathbf{X}(t_k)$. When $\beta = 1$, this method simplifies to the Euler solver in (Chen et al., 2018b; Chamberlain et al., 2021c) as $\mu_{j,n} \equiv h$, yielding ${}_P\mathbf{X}^{(k)} = \mathbf{X}^{(k-1)} + h\mathcal{F}(\mathbf{W}, \mathbf{X}^{(k-1)})$. Thus, our basic predictor can be considered as the fractional Euler method or fractional Adams–Bashforth method, which is a generalization of the Euler method used in (Chen et al., 2018b; Chamberlain et al., 2021c). However, when $\beta < 1$, we need to utilize the full memory $\{\mathcal{F}(\mathbf{W}, \mathbf{X}^{(j)})\}_{j=0}^{k-1}$. The block diagram in Fig. 1 shows the basic predictor and the short memory variant, highlighting the inclusion of nontrivial dense or skip connections in our framework. A more refined visualization is conveyed in Fig. 4, elucidating the manner in which information propagates through layers and the graph's spatial domain.

## 4 EXPERIMENTS

We execute a series of experiments to illustrate that continuous GNNs formulated within the FROND framework using $D_t^\beta$ outperform their traditional counterparts based on integer-order derivatives. Importantly, our primary aim is not to achieve state-of-the-art results, but rather to demonstrate the additional effectiveness of the FROND framework when applied to existing integer-order continuous GNNs. In the main paper, we detail the impressive results achieved by F-GRAND, particularly emphasizing its efficacy on tree-structured data, and F-CDE, highlighting its proficiency in managing large heterophilic datasets. We also validate the slow algebraic convergence, as discussed in Theorem 2, by constructing deeper GNNs with non-integer $\beta < 1$. To maintain consistency in the experiments presented in the main paper, the basic predictor solver is used instead of other solvers when $\beta < 1$.

**More Experiments In the Appendix:** The Appendix D section provides additional details covering various aspects such as experimental settings, described in Appendices D.1 to D.3, the computational complexity of F-GRAND in Appendix D.6, and analysis of F-GRAND's robustness against adversarial attacks in Appendix D.9. Furthermore, results related to other FROND-based continuous GNNs are extensively presented in the Appendix E. In the main paper, we utilize the basic predictor, as delineated in (17), while the exploration of its variants is reserved for the Appendix D.5. Additional insights into the optimal fractional-derivative order $\beta$ and fractality in graph datasets are explored in Section Appendix D.11.

## 4.1 NODE CLASSIFICATION OF F-GRAND

**Datasets and splitting.** We utilize datasets with varied topologies, including citation networks (Cora (McCallum et al., 2004), Citeseer (Sen et al., 2008), Pubmed (Namata et al., 2012)), tree-structured datasets (Disease and Airport (Chami et al., 2019)), coauthor and co-purchasing graphs (CoauthorCS (Shchur et al., 2018), Computer and Photo (McAuley et al., 2015)), and the ogbn-arxiv dataset (Hu et al., 2020). We follow the same data splitting and pre-processing in (Chami et al., 2019) for Disease and Airport datasets. Consistent with experiment settings in GRAND (Chamberlain et al., 2021c), we use random splits for the largest connected component of each other dataset. We also incorporate the large-scale Ogbn-Products dataset (Hu et al., 2021) to demonstrate the scalability of the FROND framework, with the results displayed in Table 7.

**Methods.** For a comprehensive performance comparison, we select several prominent GNN models as baselines, including GCN (Kipf & Welling, 2017), and GAT (Veličković et al., 2018). Given the inclusion of tree-structured datasets, we also incorporate well-suited baselines: HGCN(Chami et al., 2019) and GIL (Zhu et al., 2020b). To highlight the benefits of memorized dynamics in FROND, we include GRAND (Chamberlain et al., 2021c) as a special case of F-GRAND with $\beta = 1$. In line with (Chamberlain et al., 2021c), we examine two F-GRAND variants: F-GRAND-nl (7) and F-GRAND-l (9). Graph rewiring is not explored in this study. Where available, results from the paper (Chamberlain et al., 2021c) are used.

Table 1: Node classification results(%) for random train-val-test splits. The best and the second-best results are highlighted in **red** and **blue**, respectively.

| Method | Cora | Citeseer | Pubmed | CoauthorCS | Computer | Photo | CoauthorPhy | ogbn-arxiv | Airport | Disease |
|---|---|---|---|---|---|---|---|---|---|---|
| GCN | 81.5±1.3 | 71.9±1.9 | 77.8±2.9 | 91.1±0.5 | 82.6±2.4 | 91.2±1.2 | 92.8±1.0 | 72.2±0.3 | 81.6±0.6 | 69.8±0.5 |
| GAT | 81.8±1.3 | 71.4±1.9 | 78.7±2.3 | 90.5±0.6 | 78.0±19.0 | 85.7±20.3 | 92.5±0.90 | 73.7±0.1 | 81.6±0.4 | 70.4±0.5 |
| HGCN | 78.7±1.0 | 65.8±2.0 | 76.4±0.8 | 90.6±0.3 | 80.6±1.8 | 88.2±1.4 | 90.8±1.5 | 59.6±0.4 | 85.4±0.7 | 89.9±1.1 |
| GIL | 82.1±1.1 | 71.1±1.2 | 77.8±0.6 | 89.4±1.5 | – | 89.6±1.3 | – | – | 91.5±1.7 | **90.8±0.5** |
| GRAND-l | **83.6±1.0** | 73.4±0.5 | 78.8±1.7 | 92.9±0.4 | 83.7±1.2 | 92.3±0.9 | 93.5±0.9 | 71.9±0.2 | 80.5±9.6 | 74.5±3.4 |
| GRAND-nl | 82.3±1.6 | 70.9±1.0 | 77.5±1.8 | 92.4±0.3 | 82.4±2.1 | 92.4±0.8 | 91.4±1.3 | 71.2±0.2 | 90.9±1.6 | 81.0±6.7 |
| F-GRAND-l | **84.8±1.1** | **74.0±1.5** | **79.4±1.5** | **93.0±0.3** | **84.4±1.5** | **92.8±0.6** | **94.5±0.4** | **72.6±0.1** | **98.1±0.2** | **92.4±3.9** |
| $\beta$ for F-GRAND-l | 0.9 | 0.9 | 0.9 | 0.7 | 0.98 | 0.9 | 0.6 | 0.7 | 0.5 | 0.6 |
| F-GRAND-nl | 83.2±1.1 | **74.7±1.9** | **79.2±0.7** | 92.9±0.4 | 84.1±0.9 | **93.1±0.9** | 93.9±0.5 | 71.4±0.3 | **96.1±0.7** | 85.5±2.5 |
| $\beta$ for F-GRAND-nl | 0.9 | 0.9 | 0.4 | 0.6 | 0.85 | 0.8 | 0.4 | 0.7 | 0.1 | 0.7 |

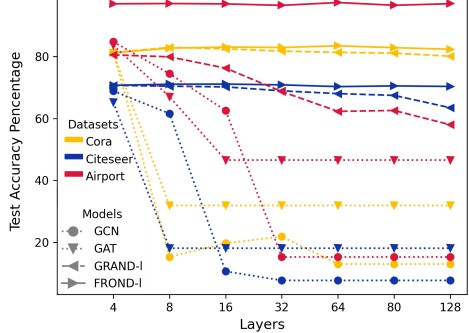

Figure 2: oversmoothing mitigation.

Table 2: Graph classification results.

| Feature | POL | | | GOS | | |
|---|---|---|---|---|---|---|
| | **Profile** | **word2vec** | **BERT** | **Profile** | **word2vec** | **BERT** |
| GraphSage | 77.60±0.68 | 80.36±0.68 | 81.22±4.81 | 92.10±0.08 | 96.58±0.22 | **97.07±0.23** |
| GCN | **78.28±0.52** | 83.89±0.53 | 83.44±0.38 | 89.53±0.49 | 96.28±0.08 | 95.96±0.75 |
| GAT | 74.03±0.53 | 78.69±0.78 | 82.71±0.19 | 91.18±0.23 | 96.57±0.34 | 96.61±0.45 |
| GRAND-l | 77.83±0.37 | **86.57±1.13** | **85.97±0.74** | **96.11±0.26** | **97.04±0.55** | 96.77±0.34 |
| F-GRAND-l | **79.49±0.43** | **88.69±0.37** | **89.29±0.93** | **96.40±0.19** | **97.40±0.03** | **97.53±0.14** |

Table 3: Node classification accuracy of F-GRAND-l under different value of $\beta$ when time $T = 8$.

| $\beta$ | 0.1 | 0.3 | 0.5 | 0.7 | 0.9 | 1.0 |
|---|---|---|---|---|---|---|
| Cora | 74.80±0.42 | 77.0±0.98 | 79.60±0.91 | 81.56±0.30 | 82.68±0.64 | 82.37±0.59 |
| Airport | 97.09±0.87 | 95.80±2.03 | 91.66±6.34 | 84.36±8.04 | 78.73±6.33 | 78.88±9.67 |

**Performance.** The results for graph node classification are summarized in Table 1, which also report the optimal $\beta$ obtained via hyperparameter tuning. Consistent with our expectations, F-GRAND surpasses GRAND across nearly all datasets, given that GRAND represents a special case of FROND with $\beta = 1$. This underscores the consistent performance enhancement offered by the integration of memorized dynamics. This advantage is particularly noticeable on tree-structured datasets such as Airports and Disease, where F-GRAND markedly outperforms the baselines. For instance, F-GRAND-l outperforms both GRAND and GIL by approximately 7% on the Airport dataset. Interestingly, our experiments indicate a smaller $\beta$ (signifying greater dynamic memory) is preferable for such fractal-structured datasets, aligning with previous studies on FDEs in biological and chemical systems (Nigmatullin, 1986; Ionescu et al., 2017). Further discussion on $\beta$ and its relation to the fractal dimension of graph datasets can be found in Section 4.4 and Appendix D.11.

## 4.2 Graph Classification of F-GRAND

We employ the Fake-NewsNet datasets (Dou et al., 2021), constructed from Politifact and Gossipcop fact-checking data. More details can be found in the Appendix D.2. This dataset features three types of node features: 768-dimensional BERT features, and 300-dimensional spaCy features, both extracted using pre-trained models, and 10-dimensional profile features from Twitter accounts. The graphs in the dataset exhibit a hierarchical tree structure. From Table 2, we observe that F-GRAND consistently outperforms GRAND with a notable edge on the POL dataset.

## 4.3 Oversmoothing of F-GRAND

To validate that F-GRAND mitigates the oversmoothing issue and performs well with numerous layers, we conducted an experiment using the basic predictor in the *Adams Bashforth Moulton* method as defined in (17). This allows us to generate architectures of varying depths. In this context, we utilize the fixed data splitting as described in (Chami et al., 2019). As illustrated in Fig. 2, optimal performance on the Cora dataset is attained with a network depth of 64 layers. When compared to GRAND-l, F-GRAND-l maintains a consistent performance level across all datasets as the number of layers increases, with virtually no performance drop observed up to 128 layers. This observation is consistent with our expectations, given that Theorem 2 predicts a slow algebraic convergence. In contrast, GRAND exhibits a faster rate of performance degradation particularly on the Airport dataset. Further details on oversmoothing mitigation are in Appendix D.7.

## 4.4 Ablation Study: Selection of $\beta$

In Table 3, we investigate the influence of $\beta$ across various graph datasets. Notably, for the Cora dataset, a larger $\beta$ is optimal, whereas, for tree-structured data, a smaller $\beta$ is preferable. This suggests that the quantity of memorized dynamics should be tailored to the dataset's topology, and a default setting of memoryless graph diffusion with $\beta = 1$ may not be optimal. More comprehensive details concerning the variations in $\beta$ can be found in the appendix, specifically in Table 15.

## 4.5 More integer-order continuous GNNs in FROND framework

Our FROND framework can be seamlessly applied to various other integer-order continuous GNNs, as elaborated in Appendix E. Specifically, here we outline the node classification results of FROND based on the CDE model in Table 4. It is evident from the results that F-CDE enhances the performance of the CDE model across almost all large heterophilic datasets. The optimal $\beta$ is determined through hyperparameter tuning. When $\beta = 1$, F-CDE seamlessly reverts to CDE, and the results from the original paper are reported. Additionally, we conduct comprehensive experiments detailed in Appendix E. The results for F-GRAND++, F-GREAD, and F-GraphCON are available in Table 19, Table 23, and Table 25, respectively. Collectively, these results demonstrate that our FROND framework can significantly bolster the performance of integer-order continuous GNNs, without introducing any additional training parameters to the backbones.

Table 4: Node classification accuracy(%) of large heterophilic datasets

| Model | Roman-empire | Wiki-cooc | Minesweeper | Questions | Workers | Amazon-ratings |
|---|---|---|---|---|---|---|
| CDE | 91.64±0.28 | 97.99±0.38 | 95.50±5.23 | 75.17±0.99 | 80.70±1.04 | 47.63±0.43 |
| F-CDE | **93.06±0.55** | **98.73±0.68** | **96.04±0.25** | 75.17±0.99 | **82.68±0.86** | **49.01±0.56** |
| $\beta$ for F-CDE | 0.9 | 0.6 | 0.6 | 1.0 | 0.4 | 0.1 |

## 5 Conclusion

We have introduced FROND, a novel graph learning framework that incorporates Caputo fractional derivatives to capture long-term memory in the graph feature updating dynamics. This approach has demonstrated superior performance compared to various traditional integer-order continuous GNNs. The resulting framework represents a significant advancement in graph representation learning, addressing key challenges in the field, such as oversmoothing. Our results highlight the potential of fractional calculus in enabling more effective graph learning algorithms.

## ACKNOWLEDGMENTS AND DISCLOSURE OF FUNDING

This research is supported by the Singapore Ministry of Education Academic Research Fund Tier 2 grant MOE-T2EP20220-0002, and the National Research Foundation, Singapore and Infocomm Media Development Authority under its Future Communications Research and Development Programme. The computational work for this article was partially performed on resources of the National Supercomputing Centre, Singapore (https://www.nscc.sg). Xuhao Li is supported by the National Natural Science Foundation of China (Grant No. 12301491) and the Anhui Provincial Natural Science Foundation (Grant No. 2208085QA02). To improve the readability, parts of this paper have been grammatically revised using ChatGPT OpenAI (2022).

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

## A    RELATED WORK

**Fractional Calculus and Its Applications**

The field of fractional calculus has seen a notable surge in interest recently due to its wide-ranging applications across various domains. These include, but are not limited to, numerical analysis (Yuste & Acedo, 2005), viscoelastic materials (Coleman & Noll, 1961), population growth models (Almeida et al., 2016), control theory (Podlubny, 1994), signal processing (Machado et al., 2011), financial mathematics (Scalas et al., 2000), and particularly in the representation of porous and fractal phenomena (Nigmatullin, 1986; Mandelbrot & Mandelbrot, 1982; Ionescu et al., 2017). Within these contexts, FDEs have been developed as a powerful extension to the conventional integer-ordered differential equations, offering a resilient mathematical framework for system analysis (Diethelm & Ford, 2002). To illustrate, in studies related to diffusion processes, researchers have utilized fractional calculus for delineating various natural and synthetic systems, from protein diffusion in cellular membranes (Krapf, 2015), to animal migration patterns (Brockmann et al., 2006), human mobility networks (Gustafson et al., 2017), and even biological phenomena pertinent to respiratory tissues and neuroscience (Ionescu et al., 2017). Interestingly, the occurrence of subdiffusion, as modeled by FDEs, has been observed in scenarios where diffusing entities encounter intermittent obstructions due to the complex geometrical structure or interaction dynamics of the environment (Diaz-Diaz & Estrada, 2022; Sornette, 2006).

Within the realm of deep learning, (Liu et al., 2022) proposes a novel approach to GNN parameter optimization using the fractional derivative. This marks a significant shift from the conventional integer-order derivative employed in optimization algorithms like SGD or Adam (Kingma & Ba, 2014) with respect to the weights. The essence of their work fundamentally differs from ours, which focuses on the fractional-order evolution of node embeddings, not gradient optimization. A detailed examination of the study by (Liu et al., 2022) is pivotal as it adopts fractional derivatives instead of the standard first-order derivatives *during the weight updating phase of a GNN in the gradient descent.* Specifically, attention is drawn to equation (16) in (Liu et al., 2022), elucidating that the fractional derivative is operational on the loss function. This stands in stark contrast to the FROND framework proposed in this work. As delineated in equation (6) of our paper, the fractional derivative is applied to the evolving node feature, representing an implementation of a fractional-order feature updating process, thereby showcasing a clear distinction in the application of fractional derivatives.

Additionally, (Antil et al., 2020) incorporates insights from fractional calculus and its L1 approximation of the fractional derivative to craft a densely connected neural network. Their aim is to adeptly handle non-smooth data and counteract the vanishing gradient problem. While our research operates within a similar sphere, we have introduced fractional calculus into integer-order continuous GNNs. Our work examines the potential of fractional derivatives in node embedding evolution to address the oversmoothing issue and establishes a connection to non-Markovian dynamic processes. Our framework paves the way for a new class of GNNs, enabling a wide spectrum of learnable feature-updating processes influenced by memory effects.

From the perspective of physics-informed machine learning, another line of research is dedicated to crafting neural networks rooted in physical laws to solve fractional PDEs. A pioneering work

in this domain is the Fractional Physics Informed Neural Networks (fPINNs) (Pang et al., 2019). Subsequent research, such as (Guo et al., 2022; Javadi et al., 2023; Wang et al., 2022a), has evolved in this direction. It is worth noting that this line of research is starkly different from our problem formulation.

**Integer-Order Continuous GNNs**

Recent research has illuminated a fascinating intersection between differential equations and neural networks. The concept of continuous dynamical systems as a framework for deep learning has been initially explored by (Weinan, 2017). The seminal work of (Chen et al., 2018b) introduces neural ODEs with open-source solvers to model continuous residual layers, which has subsequently been applied to the field of GNNs. By utilizing neural ODEs, we can align the inputs and outputs of a neural network with specific physical laws, enhancing the network's explainability (Weinan, 2017; Chamberlain et al., 2021c). Additionally, separate advancements in this domain have led to improvements in neural network performance (Dupont et al., 2019), robustness(Yan et al., 2018; Kang et al., 2021), and gradient stability (Haber & Ruthotto, 2017; Gravina et al., 2022). In practical applications, neural ODEs are demonstrating superior performance (She et al., 2024a;b; 2023b; Wang et al., 2023; She et al., 2023a). In a similar vein, (Avelar et al., 2019) models continuous residual layers in GCN, leveraging neural ODE solvers to produce output. Further, the work of (Poli et al., 2019) proposes a model that considers a continuum of GNN layers, merging discrete topological structures and differential equations in a manner compatible with various static and autoregressive GNN models. The study (Zhuang et al., 2019) introduces GODE, which enables the modeling of continuous diffusion processes on graphs. It also suggests that the oversmoothing issue in GNNs may be associated with the asymptotic stability of ODEs. Recently, GraphCON (Rusch et al., 2022) adopts the coupled oscillator model that preserves the graph's Dirichlet energy over time and mitigates the oversmoothing problem. In (Chamberlain et al., 2021a), the authors modeled information propagation as a diffusion process of a substance from regions of higher to lower concentration. The Beltrami diffusion model is utilized in (Chamberlain et al., 2021b; Song et al., 2022) to enhance rewiring and improve the robustness of the graph. The study by (Bodnar et al., 2022) introduces general sheaf diffusion operators to regulate the diffusion process and maintain non-smoothness in heterophilic graphs, leading to improved node classification performance. Meanwhile, ACMP (Wang et al., 2022b) is inspired by particle reaction-diffusion processes, taking into account repulsive and attractive force interactions between particles. Concurrently, the graph CDE model (Zhao et al., 2023a) is crafted to handle heterophilic graphs and is inspired by the convection-diffusion process. GRAND++ (Thorpe et al., 2022) leverages heat diffusion with sources to train models effectively with a limited amount of labeled training data. Concurrently, GREAD (Choi et al., 2023) articulates a GNN approach, which is premised on reaction-diffusion equations, aiming to negotiate heterophilic datasets effectively. In another development, the continuous GNN as an ODE (Maskey et al., 2023) encapsulates a graph spatial domain rewiring, leveraging the fractional order of the graph Laplacian matrix, presenting a substantial advancement in understanding graph structures. We also recommend that interested readers refer to the recent survey (Han et al., 2023) on continuous GNNs for a more thorough summarization.

*Our FROND extends the above integer-order continuous GNNs by incorporating the Caputo fractional derivative. The models mentioned can be reduced from our unified mathematical framework, with variations manifesting from the choice of the dynamic operator $\mathcal{F}(\mathbf{W}, \mathbf{X}(t))$ in (6) and as $\beta$ equals 1 in the fractional derivative operator $D_t^\beta$.*

**Skip Connections in GNNs**

The incorporation of skip or dense connections within network layers has been a transformative approach within deep learning literature. Initially popularized through the ResNet architecture (He et al., 2016), this strategy introduces shortcut pathways for gradient flow during backpropagation, thereby simplifying the training of more profound networks. While this architectural design has been instrumental in improving Convolutional Neural Networks (CNNs), it has also been employed in GNNs to bolster their representational capacity and mitigate the vanishing gradient problem. For example, the Graph U-Net (Gao & Ji, 2019) employs skip connections to enable efficient information propagation across layers. Similarly, the Jump Knowledge Network (Xu et al., 2018) implements a layer-aggregation mechanism that amalgamates outputs from all preceding layers, a strategy reminiscent of the dense connections found in DenseNet (Huang et al., 2017). Furthermore, the work (Chen et al., 2020) introduces GCNII, an extension of the standard GCN model that incorporates

two simple techniques, initial residual and identity mapping, to tackle the oversmoothing problem. Expanding on the idea of depth in GNNs, (Li et al., 2019; 2020a) propose DeepGCNs, an innovative architecture that employs residual/dense connections along with dilated convolutions. The work (Di Giovanni et al., 2023) suggests that gradient-flow message passing neural networks may be able to deal with heterophilic graphs provided that a residual connection is available. The paper (Gutteridge et al., 2023) proposes a spatial domain rewiring and focuses on long-range interactions. DRew in (Gutteridge et al., 2023) does not adhere to any ODE evolutionary structure. Additionally, the skip connection in the vDRew from (Gutteridge et al., 2023) specifically links an $n - k$-th layer to the $n$-th layer. This design is fundamentally different from our FDE approach.

By incorporating fractional calculus and memory effects into our framework, we not only offer a new perspective on understanding the structural design of skip connections in GNNs as a discretized fractional dynamical system, but we also establish a foundation for the development of more versatile and powerful mechanisms for graph representation learning.

## B    REVIEW OF CAPUTO TIME-FRACTIONAL DERIVATIVE

We appreciate the need for a more accessible explanation of the Caputo time-fractional derivative and its derivation, as the mathematical intricacies may be challenging for some readers in the GNN community. To address this, we are providing a more comprehensive background in this section. In the main paper, we briefly touched upon fractional calculus, with a particular focus on the *Caputo* fractional derivative that has been employed in our work. In this appendix, we aim to provide a more detailed overview of it and explain why it is widely employed in applications. We have based our FROND framework on the assumption that the solution to the fractional differential equation exists and is unique. The appendix provides explicit conditions for this, which are automatically satisfied in most neural network designs exhibiting local Lipschitz continuity. To simplify, these conditions are akin to those for ordinary differential equations, a common assumption implicitly made in integer-order continuous GNNs such as GRAND (Chamberlain et al., 2021c), GraphCON (Rusch et al., 2022), GRAND++ (Thorpe et al., 2022), GREAD (Choi et al., 2023) and CDE (Zhao et al., 2023a).

### B.1    CAPUTO FRACTIONAL DERIVATIVE AND ITS COMPATIBILITY OF INTEGER-ORDER DERIVATIVE

In the main paper, our focus is predominantly on the order $\beta \in (0, 1]$ for the sake of simplification. The Caputo fractional derivative of a function $f(t)$ over an interval $[0, b]$, of a general positive order $\beta \in (0, \infty)$, is defined as follows:

$$D_t^\beta f(t) = \frac{1}{\Gamma(\lceil\beta\rceil - \beta)} \int_0^t (t - \tau)^{\lceil\beta\rceil - \beta - 1} f^{[\lceil\beta\rceil]}(\tau) \mathrm{d}\tau, \tag{18}$$

Here, $\lceil\beta\rceil$ is the smallest integer greater than or equal to $\beta$, $\Gamma(\cdot)$ denotes the gamma function, and $f^{[\lceil\beta\rceil]}(\cdot)$ denotes the $\lceil\beta\rceil$-order derivative of $f(\cdot)$. Within this definition, it is presumed that $f^{[\lceil\beta\rceil]} \in L^1[0, b]$, i.e., $f^{[\lceil\beta\rceil]}$ is Lebesgue integrable, to ensure the well-defined nature of $D_t^\beta f(t)$ as per (18) (Diethelm, 2010). For a vector-valued function, the Caputo fractional derivative is defined on a component-by-component basis for each dimension, similar to the integer-order derivative. For ease of exposition, we discuss only the scalar case here, although all the following results can be generalized to vector-valued functions. The Laplace transform for a general order $\beta \in (0, \infty)$ is presented in Theorem 7.1 (Diethelm, 2010) as:

$$\mathcal{L}D_t^\beta f(s) = s^\beta \mathcal{L}f(s) - \sum_{k=1}^{\lceil\beta\rceil} s^{\beta - k} f^{[k-1]}(0). \tag{19}$$

where we assume that the Laplace transform $\mathcal{L}f$ exists on $[s_0, \infty)$ for some $s_0 \in \mathbb{R}$. In contrast, for the integer-order derivative $f^{[\beta]}$ where $\beta$ is a positive integer, we also have the formulation (19), with the only difference being the range of $\beta$. Therefore, as $\beta$ approaches some integer, the Laplace transform of the Caputo fractional derivative converges to the Laplace transform of the traditional integer-order derivative. *As a result, we can conclude that the Caputo fractional derivative operator*

*generalizes the traditional integer-order derivative since their Laplace transforms coincide when $\beta$ takes an integer value.* The inverse Laplace transform specifies the uniquely determined $D_t^\beta f = f^{[\beta]}$ when $\beta$ is an integer (in the sense of almost everywhere (Cohen, 2007)).

Under specific reasonable conditions, we can directly present this generalization as follows. Suppose $f^{[\lceil \beta \rceil]}(t)$ (18) is continuously differentiable. In this context, integration by parts can be utilized to demonstrate that

$$D_t^\beta f(t) = \frac{1}{\Gamma(\lceil \beta \rceil - \beta)} \left( - \left[ f^{[\lceil \beta \rceil]}(\tau) \frac{(t-\tau)^{\lceil \beta \rceil - \beta}}{\lceil \beta \rceil - \beta} \right] \Big|_0^t + \int_0^t f^{[\lceil \beta \rceil + 1]}(\tau) \frac{(t-\tau)^{\lceil \beta \rceil - \beta}}{\lceil \beta \rceil - \beta} \, \mathrm{d}\tau \right)$$

$$= \frac{t^{\lceil \beta \rceil - \beta} f^{[\lceil \beta \rceil]}(0)}{\Gamma(\lceil \beta \rceil - \beta + 1)} + \frac{1}{\Gamma(\lceil \beta \rceil - \beta + 1)} \times \int_0^t (t-\tau)^{\lceil \beta \rceil - \beta} f^{[\lceil \beta \rceil + 1]}(\tau) \, \mathrm{d}\tau. \tag{20}$$

As $\beta \to \lceil \beta \rceil$, we have

$$\begin{aligned} \lim_{\beta \to \lceil \beta \rceil} D_t^\beta f(t) &= f^{[[\lceil \beta \rceil]]}(0) + \int_0^t f^{[[\lceil \beta \rceil]+1]}(\tau) \mathrm{d}\tau \\ &= f^{[[\lceil \beta \rceil]]}(0) + f^{[[\lceil \beta \rceil]]}(t) - f^{[[\lceil \beta \rceil]]}(0) \\ &= f^{[[\lceil \beta \rceil]]}(t). \end{aligned} \tag{21}$$

In parallel to the integer-order derivative, *given certain conditions* ((Diethelm, 2010)[Lemma 3.13]), the Caputo fractional derivative possesses the semigroup property as illustrated in (Diethelm, 2010)[Lemma 3.13]:

$$D_t^\varepsilon D_t^n f = D_t^{n+\varepsilon} f. \tag{22}$$

Nonetheless, it is crucial to recognize that, in general, the Caputo fractional derivative does not exhibit the semigroup property, a characteristic inherent to integer-order derivatives, as detailed in (Diethelm, 2010)[Section 3.1]. The Caputo fractional derivative also exhibits *linearity*, but does not adhere to the same Leibniz and chain rules as its integer counterpart. As such properties are not utilized in our work, we refer interested readers to (Diethelm, 2010)[Theorem 3.17 and Remark 3.5.].

### B.2 Comparison between Riemann–Liouville and Caputo Derivative

Another well-known fractional derivative is the Riemann–Liouville derivative, which, however, sees less use in practical applications (see Appendix B.4 for more insights). In this section, we offer a succinct introduction to the Riemann–Liouville derivative and compare it with Caputo's definition. The Riemann–Liouville fractional derivative is given as

$$\widehat{D}_t^\beta f(t) := \frac{1}{\Gamma(\lceil \beta \rceil - \beta)} \frac{\mathrm{d}^{\lceil \beta \rceil}}{\mathrm{d}t^{\lceil \beta \rceil}} \int_0^t (t-\tau)^{\lceil \beta \rceil - \beta - 1} f(\tau) \mathrm{d}\tau \tag{23}$$

Here again, we make the assumption that sufficient conditions are satisfied to ensure well-definedness (refer to (Diethelm, 2010)[section 2.2] for details).

We compare the Taylor expansion for the two definitions of fractional derivatives, namely the Riemann-Liouville and Caputo derivatives, with the conventional integer-order derivative. This comparison allows us to clearly highlight the distinctions among the differential equations defined under these three different approaches.

- **Classical Integer-order Taylor Expansion:** (Diethelm, 2010)[Theorem 2.C] Assuming that $f$ has absolutely continuous $(m-1)$-st derivative, we have that for $t \in [0, b]$,

$$f(t) = \sum_{k=0}^{m-1} \frac{t^k}{k!} \frac{\mathrm{d}^k f(0)}{\mathrm{d}t^k} + J^m \frac{\mathrm{d}^m}{\mathrm{d}t^m} f(t) \tag{24}$$

where $J^n f(t) := \frac{1}{\Gamma(n)} \int_0^t (t-\tau)^{n-1} f(\tau) \, \mathrm{d}\tau$. Note that here, *k is an integer*.

- **Riemann-Liouville Fractional Taylor Expansion:** (Diethelm, 2010)[Theorem 2.24] Let $n > 0$ and $m = \lfloor n \rfloor + 1$. Assume that $f$ is such that $J^{m-n} f$ has absolutely continuous $(m-1)$-st derivative.

Then,

$$f(t) = \frac{t^{n-m}}{\Gamma(n-m+1)} J^{m-n} f(0) + \sum_{k=1}^{m-1} \frac{t^{k+n-m}}{\Gamma(k+n-m+1)} \widehat{D}_t^{k+n-m} f(0) + J^n \widehat{D}_t^n f(t). \quad (25)$$

Note that in the case $n \in \mathbb{N}$ we have $m = n + 1$ and $\Gamma(n - m + 1) = \Gamma(0) = \infty$, and the first term before the sum vanishes. Hence, we recover the classical result. For general $n$, *the order in $\widehat{D}_t^{k+n-m}$ is not a integer.*

• **Caputo Fractional Taylor Expansion:** (Diethelm, 2010)[Theorem 3.8.] Assume that $n \geq 0, m = \lceil n \rceil$, and $f$ has absolutely continuous $(m-1)$-st derivative. Then

$$f(t) = \sum_{k=0}^{m-1} \frac{t^k}{k!} D_t^k f(0) + J^n D_t^n f(t). \quad (26)$$

Note *the order in $D_t^k$ is an integer.* If we compare (24) to (26), it becomes evident that the Caputo derivative closely resembles the classical integer-order derivative in terms of Taylor expansion. This fact influences the initial conditions for the differential equations introduced in the following section.

### B.3 (CAPUTO) FRACTIONAL DIFFERENTIAL EQUATION

In this section, we first compare the initial conditions for FDEs under the Riemann-Liouville and Caputo definitions. Following this, we present the precise conditions for the existence and uniqueness of the solution to the fractional differential equation. These conditions closely align with those of ordinary differential equations, which are widely assumed by integer-order continuous GNNs (Chamberlain et al., 2021c; Rusch et al., 2022; Thorpe et al., 2022; Choi et al., 2023; Zhao et al., 2023a).

#### B.3.1 RIEMANN-LIOUVILLE CASE

Drawing from the Riemann-Liouville fractional Taylor expansion, let us assume that $e$ is a given function with the property that there exists some function $g$ such that $g = \widehat{D}_t^\beta e$. The solution of the Riemann-Liouville differential equation of the form

$$\widehat{D}_t^\beta f = g \quad (27)$$

is given by

$$f(t) = e(t) + \sum_{j=1}^{\lceil \beta \rceil} c_j t^{n-j} \quad (28)$$

where $c_j$ are arbitrary constants. In other words, to uniquely determine the solution from (25), we need to know the value of $\widehat{D}_t^{k+n-m} f(0)$. This is akin to a $k$ order ordinary differential equation where the initial conditions are assumed as $\frac{d^k}{dt^k} f(0)$, *with the distinction that the order in $\widehat{D}_t^{k+n-m}$ is not an integer.*

#### B.3.2 CAPUTO CASE

Similarly, if $e$ is a given function with the property that $e = D_t^\beta g$ and if we intend to solve

$$D_t^\beta f = g \quad (29)$$

then we find

$$f(t) = e(t) + \sum_{j=1}^{\lceil \beta \rceil} c_j t^{\lceil \beta \rceil - j} \quad (30)$$

once more, with $c_j$ as arbitrary constants. Thus, to obtain a unique solution, it is natural to prescribe the values of *integer order derivatives* $f(0), D_t^1 f(0), \ldots, D_t^{\lceil \beta \rceil - 1} f(0)$ in the Caputo setting, *mirroring traditional ordinary differential equations.*

B.3.3 EXISTENCE AND UNIQUENESS OF THE (CAPUTO) SOLUTION

Next, we delve into a general Caputo fractional differential equation, presented as follows:

$$D_t^\beta y(t) = g(t, y(t)) \tag{31}$$

conjoined with suitable initial conditions. As hinted in (29) and (30), the initial conditions take the form:

$$D_t^k y(0) = y_0^{(k)}, \quad k = 0, 1, \ldots, \lceil \beta \rceil - 1. \tag{32}$$

The following theorem addresses the existence and uniqueness of solutions:

- **Caputo existence and uniqueness theorem:** (Diethelm, 2010)[Theorem 6.8] Let $y_0^{(0)}, \ldots, y_0^{(m-1)} \in \mathbb{R}$ and $h^* > 0$. Define the set $G := [0, h^*] \times \mathbb{R}$ and let the function $g : G \to \mathbb{R}$ be continuous and fulfill a *Lipschitz condition* with respect to the second variable, i.e.,

$$|g(x, y_1) - g(x, y_2)| \leq L |y_1 - y_2|$$

  for some constant $L > 0$ independent of $x, y_1$, and $y_2$. Then there *uniquely exists* function $y \in C[0, h^*]$ solving the initial value problem (31) and (32).

For a point of reference, we also provide the well-known Picard–Lindelöf uniqueness theorem for first-order ordinary differential equations.

- **Picard–Lindelöf theorem** (Hartman, 2002)[Page 8] Let $D \subseteq \mathbb{R} \times \mathbb{R}^n$ be a closed rectangle with $(t_0, y_0) \in \text{int } D$, the interior of $D$. Let $g : D \to \mathbb{R}^n$ be a function that is continuous in $t$ and *Lipschitz continuous* in $y$. Then, there exists some $\varepsilon > 0$ such that the initial value problem

$$y'(t) = g(t, y(t)), \quad y(t_0) = y_0.$$

  has a *unique solution* $y(t)$ on the interval $[t_0, t_0 + \varepsilon]$.

This allows us to draw parallels between the existence and uniqueness theorem of the Caputo fractional differential equation and its integer-order ordinary differential equation equivalent. We also remind readers that standard neural networks, as compositions of linear maps and pointwise non-linear activation functions with bounded derivatives (such as fully-connected and convolutional networks), satisfy global Lipschitz continuity with respect to the input. For attention neural networks, which are compositions of softmax and matrix multiplication, we observe local Lipschitz continuity. To see this, suppose $\mathbf{v} = \text{softmax}(\mathbf{u}) \in \mathbb{R}^{n \times 1}$. Then

$$\frac{d\mathbf{v}}{\partial \mathbf{u}} = \text{diag}(\mathbf{v}) - \mathbf{v}\mathbf{v}^\top = \begin{bmatrix} v_1(1 - v_1) & -v_1 v_2 & \ldots & -v_1 v_n \\ -v_2 v_1 & v_2(1 - v_2) & \ldots & -v_2 v_n \\ \vdots & \vdots & \ddots & \vdots \\ -v_n v_1 & -v_n v_2 & \ldots & v_n(1 - v_n) \end{bmatrix}.$$

For bounded input, we have a bounded Jacobian. All the integer-order continuous GNN works, such as recent contributions like (Chamberlain et al., 2021c; Rusch et al., 2022; Thorpe et al., 2022; Choi et al., 2023; Zhao et al., 2023a) assume the uniqueness of the ODE solutions. *This means that all the integer-order continuous GNNs can be extended by our FROND framework with fractional dynamics.*

B.4 REASONS FOR CHOOSING CAPUTO DERIVATIVE

We now explain the reasons behind our preference for the Caputo fractional derivative:

1. As previously discussed, Caputo fractional differential equations align with integer-order differential equations concerning initial conditions.

2. The Caputo fractional derivative maintains a more intuitive resemblance to the integer-order derivative and satisfies the significant property of equating to zero when applied to a constant. This property is not satisfied by the Riemann-Liouville fractional derivative. Refer to (Diethelm, 2010)[Example 2.4. and Example 3.1.] for further clarification.

3. Given its widespread application in the literature for practical use cases, numerical methods for solving Caputo fractional differential equations have been meticulously developed and exhaustively analyzed (Diethelm, 2010; Diethelm et al., 2004; Deng, 2007).

## C  NUMERICAL SOLVERS FOR FROND

We remind readers that numerous methods for training neural ODEs, and consequently updating the weights $\theta$ in the neural network have been proposed. These include the autodifferentiation technique in PyTorch (Yan et al., 2018; Paszke et al., 2017), the adjoint sensitivity method (Chen et al., 2018b), and Snode (Quaglino et al., 2019). In our work, we employ the most straightforward autodifferentiation technique for training FROND with fractional neural differential equations, leveraging the numerical solvers outlined in (Diethelm, 2010; Diethelm et al., 2004; Deng, 2007). While we plan to investigate more sophisticated techniques for training FROND in future work, we have open-sourced our current solver implementations in `https://github.com/zknus/torchfde`. We believe these will serve as valuable tools for the GNN community, encouraging the advancement of a unique class of GNNs that incorporate memory effects.

In traditional integer-order continuous GNNs (Chamberlain et al., 2021c; Thorpe et al., 2022; Rusch et al., 2022; Song et al., 2022; Choi et al., 2023; Zhao et al., 2023a), the time parameter $t$ serves as a continuous analog to GNN layers, resembling the concept of neural ODEs (Chen et al., 2018b) as continuous residual networks. Time discretization plays a crucial role in many numerical solvers for neural ODEs. For example, the explicit Euler scheme reduces neural ODEs to residual networks with shared hidden layers (Chen et al., 2018b). More sophisticated discretization methods, such as adaptive step size solvers (Atkinson et al., 2011), provide accurate solutions but require additional computational resources.

Unlike prior studies, our work involves fractional-order ODEs, which are more complex than ODEs when the derivative order $\beta$ takes non-integer values. We present the *fractional Adams–Bashforth–Moulton method* with three variants utilized in this work, demonstrating how the time parameter continues to serve as a continuous analog to the layer index and how the non-local nature of fractional derivatives leads to nontrivial dense or skip connections between layers. Additionally, we also present one implicit L1 solver for solving FROND when $\beta$ is not an integer. It is worth noting that various neural ODE solvers remain applicable for FROND when $\beta$ is an integer.

We first recall the FROND framework

$$D_t^\beta \mathbf{X}(t) = \mathcal{F}(\mathbf{W}, \mathbf{X}(t)), \quad \beta > 0,$$

where $\beta$ denotes the fractional order of the derivative, and $\mathcal{F}$ is a dynamic operator on the graph like the models presented in Section 2.2. The initial condition is set as $\mathbf{X}^{[\lceil \beta \rceil - 1]}(0) = \ldots = \mathbf{X}(0) = \mathbf{X}$ consisting of the preliminary node features, akin to the initial conditions seen in ODEs.

### C.1  BASIC PREDICTOR

Referencing (Diethelm et al., 2004), we first employ a preliminary numerical solver called "predictor" through time discretisation $t_j = jh$, where the discretisation parameter $h$ is a small positive value:

$$_\mathrm{P}\mathbf{X}^{(k)} = \sum_{j=0}^{\lceil \beta \rceil - 1} \frac{t_k^j}{j!} \mathbf{X}^{[j]}(0) + \frac{1}{\Gamma(\beta)} \sum_{j=0}^{k-1} \mu_{j,k} \mathcal{F}(\mathbf{W}, \mathbf{X}^{(j)}), \tag{33}$$

where $\mu_{j,n} = \frac{h^\beta}{\beta} \left( (n-j)^\beta - (n-1-j)^\beta \right)$, $k$ denotes the discrete time index (iteration), and $t_k = kh$ represents the discretized time steps. $\mathbf{X}^{(k)}$ is the numerical approximation of $\mathbf{X}(t_k)$. When $\beta = 1$, this method simplifies to the Euler solver in (Chen et al., 2018b; Chamberlain et al., 2021c) as $\mu_{j,n} \equiv h$, yielding $_\mathrm{P}\mathbf{X}^{(k)} = \mathbf{X}^{(k-1)} + h\mathcal{F}(\mathbf{W}, \mathbf{X}^{(k-1)})$. Thus, our basic predictor can be considered as the fractional Euler method or fractional Adams–Bashforth method, which is a generalization of the Euler method used in (Chen et al., 2018b; Chamberlain et al., 2021c). However, when $\beta < 1$, we need to utilize the full memory $\{\mathcal{F}(\mathbf{W}, \mathbf{X}^{(j)})\}_{j=0}^{k-1}$.

The block diagram of this basic predictor, shown in Fig. 3, reveals that our framework introduces nontrivial dense or skip connections between layers. A more refined visualization is conveyed in Fig. 4, elucidating the manner in which information propagates through layers and the graph's spatial domain.

## C.2 PREDICTOR-CORRECTOR

The corrector formula from (Diethelm et al., 2004), a fractional variant of the one-step Adams-Moulton method, refines the initial approximation using the predictor $_P\mathbf{X}^{(k)}$ as follows:

$$\mathbf{X}^{(k)} = \sum_{j=0}^{\lceil\beta\rceil-1} \frac{t_k^j}{j!} \mathbf{X}^{[j]}(0) + \frac{1}{\Gamma(\beta)} \sum_{j=0}^{k-1} \eta_{j,k} \mathcal{F}(\mathbf{W}, \mathbf{X}^{(j)}) + \frac{1}{\Gamma(\beta)} \eta_{k,k} \mathcal{F}(\mathbf{W}, _P\mathbf{X}^{(k)}). \quad (34)$$

Here we show the coefficients $\eta_{j,n}$ in the predictor-corrector variant (34) from (Diethelm et al., 2004):

$$\eta_{j,k}(\beta) = \frac{h^\beta}{\beta(\beta+1)} \times \begin{cases} (k-1)^{\beta+1} - (k-1-\beta)k^\beta & \text{if } j = 0, \\ (k-j+1)^{\beta+1} + (k-1-j)^{\beta+1} - 2(k-j)^{\beta+1} & \text{if } 1 \leq j \leq k-1, \\ 1 & \text{if } j = k. \end{cases} \quad (35)$$

## C.3 SHORT MEMORY PRINCIPLE

When $T$ is large, computational time complexity becomes a challenge due to the non-local nature of fractional derivatives. To mitigate this, (Deng, 2007; Podlubny, 1999) suggest leveraging the short memory principle to modify the summation in (17) and (34) to $\sum_{j=n-K}^{n-1}$. This corresponds to employing a shifting memory window with a fixed width $K$. The block diagram is depicted in Fig. 3.

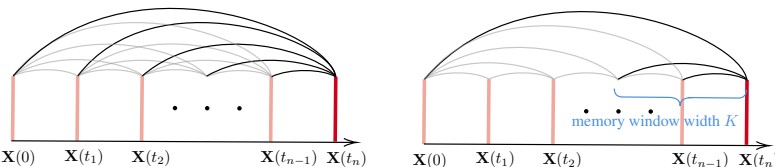

Figure 3: Diagrams of fractional Adams–Bashforth–Moulton method with full (left) and short (right) memory.

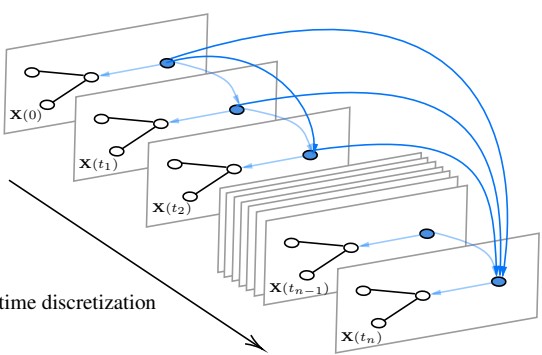

Figure 4: Model discretization in FROND with the basic predictor solver. Unlike the Euler discretization in ODEs, FDEs incorporate connections to historical times, introducing memory effects. Specifically, the dark blue connections observed in FDEs are absent in ODEs. The weight of these skip connections correlates with $\mu_{j,k}(\beta)$ as detailed in (17).

## C.4 L1 SOLVER

The L1 scheme is one of the most popular methods to approximate the Caputo fractional derivative in time. It utilizes a backward differencing method for effective approximation of derivatives. Referencing (Gao & Sun, 2011; Sun & Wu, 2006), we have the L1 approximation of Caputo fractional

derivative as follows:

$$D_t^\beta \mathbf{X}^{(k)} \approx \mu \sum_{j=0}^{k-1} R_{k,j}^\beta (\mathbf{X}^{(j+1)} - \mathbf{X}^{(j)})$$

where $h$ is the temporal step size,

$$\mu = \frac{1}{h^\beta \Gamma(2-\beta)}, \qquad R_{k,j}^\beta = (k-j)^{1-\beta} - (k-j-1)^{1-\beta}, \qquad 0 \le j \le k-1.$$

Applying the L1 solver for our problem, we obtain

$$\mu \sum_{j=0}^{k-1} R_{k,j}^\beta (\mathbf{X}^{(j+1)} - \mathbf{X}^{(j)}) = (\mathbf{A}(\mathbf{X}^{(k)}) - \mathbf{I})\mathbf{X}^{(k)}.$$

Manipulating the above equation, we have

$$\mathbf{X}^{(k)} - \frac{1}{\mu}(\mathbf{A}(\mathbf{X}^{(k)}) - \mathbf{I})\mathbf{X}^{(k)} = \mathbf{X}^{(k-1)} - \sum_{j=0}^{k-2} R_{k,j}^\beta (\mathbf{X}^{(j+1)} - \mathbf{X}^{(j)})$$

The above formula is an implicit nonlinear scheme. To solve it without calculating the inversion of a matrix, we propose the following iteration method:

(1) Compute a basic approximation of $\mathbf{X}(t_k)$ with the following formula:

$$_{\mathrm{P}}\mathbf{X}^{(k)} - \frac{1}{\mu}(\mathbf{A}(\mathbf{X}^{(k-1)}) - \mathbf{I})\mathbf{X}^{(k-1)} = \mathbf{X}^{(k-1)} - \sum_{j=0}^{k-2} R_{k,j}^\beta (\mathbf{X}^{(j+1)} - \mathbf{X}^{(j)}).$$

(2) Substitute the above $_{\mathrm{P}}\mathbf{X}^{(k)}$ into the implicit scheme to update $\mathbf{X}^{(k)}$:

$$\mathbf{X}^{(k)} - \frac{1}{\mu}(\mathbf{A}(_{\mathrm{P}}\mathbf{X}^{(k)}) - \mathbf{I})_{\mathrm{P}}\mathbf{X}^{(k)} = \mathbf{X}^{(k-1)} - \sum_{j=0}^{k-2} R_{k,j}^\beta (\mathbf{X}^{(j+1)} - \mathbf{X}^{(j)}). \tag{36}$$

The step (2) can be repeated multiple times to obtain an accurate approximation of $\mathbf{X}(t_k)$.

## D  DATASETS, SETTINGS AND MORE EXPERIMENTS FOR **F-GRAND** MODEL

### D.1  DATASETS

The dataset statistics used in Table 1 are provided in Table 5. Following the experimental framework in (Chamberlain et al., 2021c), we select the largest connected component from each dataset, except for the tree-like graph datasets (Airport and Disease). However, for the study of oversmoothing, we use a fixed data splitting approach over the entire datasets, as described in (Chami et al., 2019).

### D.2  GRAPH CLASSIFICATION DETAILS

We use the Fake-NewsNet datasets from (Dou et al., 2021), constructed based on fact-checking information obtained from Politifact and Gossipcop. The dataset incorporates four distinct node feature categories, including 768-dimensional BERT features and 300-dimensional spaCy features, which are derived using pre-trained BERT and spaCy word2vec models, respectively. Additionally, a 10-dimensional profile feature is extracted from individual Twitter accounts' profiles. Each graph within the dataset is characterized by a hierarchical tree structure, with the root node representing the news item and the leaf nodes representing Twitter users who have retweeted said news. An edge exists between a user node and the news node if the user retweeted the original news tweet, while an edge between two user nodes is established when one user retweets the news tweet from another user. This hierarchical organization facilitates the analysis of the spread and influence of both genuine and fabricated news within the Twitter ecosystem. The datasets statistics are summarized in Table 6.

Table 5: Dataset Statistics used in Table 1

| Dataset | Type | Classes | Features | Nodes | Edges |
|---------|------|---------|----------|-------|-------|
| Cora | citation | 7 | 1433 | 2485 | 5069 |
| Citeseer | citation | 6 | 3703 | 2120 | 3679 |
| PubMed | citation | 3 | 500 | 19717 | 44324 |
| Coauthor CS | co-author | 15 | 6805 | 18333 | 81894 |
| Computers | co-purchase | 10 | 767 | 13381 | 245778 |
| Photos | co-purchase | 8 | 745 | 7487 | 119043 |
| CoauthorPhy | co-author | 5 | 8415 | 34493 | 247962 |
| OGB-Arxiv | citation | 40 | 128 | 169343 | 1166243 |
| Airport | tree-like | 4 | 4 | 3188 | 3188 |
| Disease | tree-like | 2 | 1000 | 1044 | 1043 |

Table 6: Dataset and graph statistics used in Table 2

| Dataset | Graphs (Fake) | Total Nodes | Total Edges | Avg. Nodes per Graph |
|---------|---------------|-------------|-------------|----------------------|
| Politifact (POL) | 314 (157) | 41,054 | 40,740 | 131 |
| Gossipcop (GOS) | 5464 (2732) | 314,262 | 308,798 | 58 |

## D.3 IMPLEMENTATION DETAILS

Our FROND framework adheres to the experimental settings of the foundational integer-order continuous GNNs, diverging only in the introduction of fractional derivatives in place of integer derivatives. In implementing FROND, we employ one fully-connected (FC) layer on the raw input features to obtain the initial node representations $\mathbf{X}(0)$. Subsequently, we utilize another FC layer as the decoder function to process the FDE output, $\mathbf{X}(T)$, for executing downstream tasks. For more detailed information regarding the hyperparameter settings, we kindly direct the readers to the accompanying supplementary material, which includes the provided code for reproducibility. Our experiments were conducted using NVIDIA RTX A5000 graphics cards.

## D.4 LARGE SCALE OGBN-PRODUCTS DATASET

In this section, we extend our evaluation to include another large-scale dataset, Ogbn-products, adhering to the experimental settings outlined in (Hu et al., 2021). For effective handling of this large dataset, we employ a mini-batch training approach, which involves sampling nodes and constructing subgraphs, as proposed by GraphSAINT (Zeng et al., 2020). Upon examination, we observe that F-GRAND-l demonstrates superior performance compared to both GRAND-l and the GCN model, although it falls slightly short of the performance exhibited by GraphSAGE. This outcome could potentially be attributed to the insufficient dynamic setting in (9). As such, the more advanced dynamic $\mathcal{F}(\mathbf{W}, \mathbf{X}(t))$ in (6) may require additional refinement.

Table 7: Node classification accuracy(%) on Ogbn-products dataset

| Model | MLP | Node2vec | Full-batch GCN | GraphSAGE | GRAND-l | F-GRAND-l |
|-------|-----|----------|----------------|-----------|---------|-----------|
| Acc | 61.06±0.08 | 72.49±0.10 | 75.64±0.21 | 78.29±0.16 | 75.56±0.67 | 77.25±0.62 |

## D.5 PERFORMANCE OF DIFFERENT SOLVER VARIANTS

In this work, we introduce two types of solvers with distinct variants. We evaluate the performance of these variants in Table 8. Specifically, we run F-GRAND on the Cora and Airport datasets with $h = 1$ and $T = 64$. The solver variants perform comparably. For the Cora dataset, the fractional Adams–Bashforth–Moulton method with a short memory parameter of $K = 10$ performs slightly worse than the other variants. However, it demonstrates comparable performance to other solver variants on the Airport dataset.

Table 8: Node classification accuracy(%) under different solver when time $T = 64$

|  | Predictor(17) | Predictor-Corrector (34) | Short Memory | Implicit L1 |
|---|---|---|---|---|
| Cora($\beta = 0.6$) | 83.44±0.91 | 83.45±1.09 | 81.51±1.07 | 82.85±1.08 |
| Airport($\beta = 0.1$) | 97.41±0.42 | 96.85±0.36 | 97.23±0.59 | 96.06±1.59 |

Table 9: Node classification accuracy based on memory $K$ on the Cora dataset when time $T = 40$.

| memory $K$ | 1 | 5 | 10 | 15 | 20 | 25 | 30 | 35 | 40 |
|---|---|---|---|---|---|---|---|---|---|
| Accuracy (%) | 74.9±0.8 | 80.8±0.8 | 83.3±1.1 | 83.9±1.2 | 84.2±1.1 | 84.1±1.2 | 84.5±1.1 | 84.1±1.1 | 84.8±1.1 |
| Inference (ms) | 9.81 | 17.53 | 24.97 | 32.03 | 38.79 | 42.99 | 45.27 | 48.70 | 48.35 |

### D.5.1 FURTHER CLARIFICATION ON TWO ACCURACIES

This section aims to clarify potential ambiguities surrounding the term "accuracy" by distinguishing between "task accuracy" and "numerical accuracy." Task accuracy pertains to the performance of GNNs on tasks such as node classification. In contrast, numerical accuracy relates to the precision of numerical solutions to FDEs, a critical concern in mathematics.

For example, generally, a larger $K$ value in the Short Memory solver might enhance both numerical and GNN task accuracy. However, it comes with the trade-off of demanding more computational resources. Furthermore, the two accuracies are related, but not equivalent to each other. For added clarity, we conducted an ablation study on the Cora dataset, keeping all parameters constant except for the memory parameter $K$. The outcomes of this study are detailed in Table 9. Our observations indicate that while increasing the value of $K$ can improve numerical accuracy and potentially GNN task accuracy, the computational cost also rises. Notably, the gains in task accuracy plateau beyond a $K$ value of 15.

We also remind the readers that in the literature, to solve FDEs, there exist other more numerically accurate solvers like (Jin et al., 2017; Tian et al., 2015; Lv & Xu, 2016) that use higher convergence order. In general, these kinds of solvers can theoretically reduce computation cost and memory storage, as we can obtain the same numerical accuracy using larger step sizes compared to lower-order solvers. It does not aim to improve GNN task accuracy as we can take smaller step sizes to achieve this, but it may be helpful for other performances like computation cost and memory storage reduction. In our paper, we focus on task accuracy. Therefore, classical solvers are used in our work. Nonetheless, more numerically accurate solvers could potentially benefit other applications of fractional dynamics, particularly when GNNs are utilized to simulate and forecast real physical systems.

### D.6 COMPUTATION TIME

It should be emphasized that our FROND framework *does not introduce any additional training parameters* to the backbone integer-order continuous GNNs. Instead, we simply modify the integration method from standard integration to fractional integration.

In this section, we report the inference time of the different solver variants in Tables 10 to 13. For comparison, we consider the neural ODE solver for $\beta = 1$, which includes Euler, RK4, Implicit Adams, and dopri5 methods as per in the paper (Chen et al., 2018b). We observe that when $T = 4$, the inference time required by the FROND solver variants is similar to that of the ODE Euler solver. However, for larger $T = 64$, the basic Predictor (17) solver requires more inference time than Euler and is comparable to RK4. For more accurate approximation solver variants (34) and (36) incorporating the corrector formula, Tables 12 and 13 show that these methods require more computational time as the number of iterations increases. While the advantages of these solvers might not be pronounced for GNN node classification tasks, they could provide benefits for other applications of fractional dynamics, such as when GNNs are used to simulate and forecast real physical systems.

### D.7 CONTINUED STUDY OF OVERSMOOTHING

To corroborate that FROND mitigates the issue of oversmoothing and performs well with an increasing number of layers, we conducted an experiment employing the basic predictor with up to 128 layers in

Table 10: Average time under different solvers when time $T = 4$ and hidden dimension is 64 on Cora dataset

|  | Predictor(17) | Predictor-Corrector(34) | Short Memory | Implicit L1 | Euler | RK4 | Implicit Adams | dopri5 |
|---|---|---|---|---|---|---|---|---|
| Inference time (ms) | 0.98 | 1.67 | 0.98 | 0.62 | 0.96 | 2.06 | 3.20 | 11.91 |

Table 11: Average time under different solvers when time $T = 64$ and hidden dimension is 64 on Cora dataset

|  | Predictor(17) | Predictor-Corrector(34) | Short Memory | Implicit L1 | Euler | RK4 | Implicit Adams | dopri5 |
|---|---|---|---|---|---|---|---|---|
| Inference time (ms) | 44.46 | 160.92 | 30.26 | 221.74 | 12.16 | 42.66 | 103.46 | 66.15 |

Table 12: Average time of (34) and (36) with correctors, used to refine the approximation, when time $T = 4$ and hidden dimension is 64 on the Cora dataset.

| Predictor-Corrector (34) | 1 | 3 | 5 | 10 |
|---|---|---|---|---|
| Inference time (ms) | 1.67 | 3.31 | 4.74 | 8.34 |
| Implicit-L1 (36) | 1 | 3 | 5 | 10 |
| Inference time (ms) | 0.62 | 1.04 | 1.48 | 2.55 |

Table 13: Average time of (34) and (36) with correctors, used to refine the approximation, when time $T = 64$ and hidden dimension is 64 on the Cora dataset.

| Predictor-Corrector (34) | 1 | 3 |
|---|---|---|
| Inference time (ms) | 160.92 | 442.88 |
| Implicit-L1 (36) | 1 | 3 |
| Inference time (ms) | 221.74 | 441.60 |

the main paper. The results are presented in Fig. 2. For this experiment, we utilized the fixed data splitting approach for the Cora and Citeseer dataset without using the Largest Connected Component (LCC) as described in (Chami et al., 2019).

In the supplementary material, we further probe oversmoothing by conducting experiments with an increased number of layers, reaching up to 256. The results of these experiments are illustrated in Table 14. From our observations, F-GRAND-l maintains a consistent performance level even as the number of layers escalates. This contrasts with GRAND-l, where there is a notable performance decrease with the increase in layers. For instance, on the Cora datasets, the accuracy of GRAND-l drops from 81.29% with 4 layers to 73.37% with 256 layers. In stark contrast, our F-GRAND-l model exhibits minimal performance decrease on this dataset. On the Airport dataset, F-GRAND-l registers a slight decrease to 94.91% with 256 layers from 97.0% with 4 layers. However, the performance of GRAND-l significantly drops to 53.0%. These observations align with our expectations, as Theorem 2 predicts a slow algebraic convergence rate, while GRAND exhibits a more rapid performance degradation.

Additionally, we note that the optimal number of layers for F-GRAND is 64 on the Cora and Airport datasets, whereas on the Cirtesser dataset, the best performance is achieved with 16 layers.

## D.8 ABLATION STUDY: SELECTION OF $\beta$ CONTINUED

In the main paper, we explore the impact of the fractional order parameter $\beta$ across a variety of graph datasets, with the results of these investigations presented in Table 3. More comprehensive details concerning the variations in $\beta$ can be found in Table 15.

Table 14: oversmoothing mitigation under fixed data splitting without LCC

| Dataset | Model | 4 | 8 | 16 | 32 | 64 | 80 | 128 | 256 |
|---|---|---|---|---|---|---|---|---|---|
| Cora | GCN | 81.35±1.27 | 15.3±3.63 | 19.70±7.06 | 21.86±6.09 | 13.0±0.0 | 13.0±0.0 | 13.0±0.0 | 13.0±0.0 |
| | GAT | 80.95±2.28 | 31.90±0.0 | 31.90±0.0 | 31.90±0.0 | 31.90±0.0 | 31.90±0.0 | 31.90±0.0 | 31.90±0.0 |
| | GRAND-l | 81.29±0.43 | 82.95±0.52 | 82.48±0.46 | 81.72±0.35 | 81.33±0.22 | 81.07±0.44 | 80.09±0.43 | 73.37±0.59 |
| | F-GRAND-l | 81.17±0.75 | 82.68±0.64 | 83.05±0.81 | 82.90±0.81 | 83.44±0.91 | 82.85±0.89 | 82.34±0.83 | 81.74±0.53 |
| Citeseer | GCN | 68.84±2.46 | 61.58±2.09 | 10.64±1.79 | 7.7±0.0 | 7.7±0.0 | 7.7±0.0 | 7.7±0.0 | 7.7±0.0 |
| | GAT | 65.20±0.57 | 18.10±0.0 | 18.10±0.0 | 18.10±0.0 | 18.10±0.0 | 18.10±0.0 | 18.10±0.0 | 18.10±0.0 |
| | GRAND-l | 70.68±1.23 | 70.39±0.68 | 70.18±0.56 | 68.90±1.50 | 68.01±1.47 | 67.44±1.25 | 63.45±2.86 | 56.98±1.26 |
| | F-GRAND-l | 70.68±1.23 | 71.04±0.68 | 71.08±1.12 | 70.83±0.90 | 70.27±0.86 | 70.50±0.76 | 70.32±1.67 | 71.0±0.45 |
| Airport | GCN | 84.77±1.45 | 74.43±8.19 | 62.56±2.16 | 15.27±0.0 | 15.27±0.0 | 15.27±0.0 | 15.27±0.0 | 15.27±0.0 |
| | GAT | 83.59±1.51 | 67.02±4.70 | 46.56±0.0 | 46.56±0.0 | 46.56±0.0 | 46.56±0.0 | 46.56±0.0 | 46.56±0.0 |
| | GRAND-l | 80.53±9.59 | 79.88±9.67 | 76.24±3.80 | 68.67±4.02 | 62.28±10.83 | 50.38±2.98 | 57.96±11.63 | 53.0±14.85 |
| | F-GRAND-l | 97.0±0.79 | 97.09±0.87 | 96.97±0.84 | 96.50±0.60 | 97.41±0.42 | 96.53±0.74 | 97.03±0.55 | 94.91±3.72 |

Table 15: Node classification accuracy(%) under different value of $\beta$ when time $T = 8$.

| $\beta$ | 0.1 | 0.2 | 0.3 | 0.4 | 0.5 | 0.6 | 0.7 | 0.8 | 0.9 | 1.0 |
|---|---|---|---|---|---|---|---|---|---|---|
| Cora | 74.80±0.42 | 76.10±0.34 | 77.0±0.98 | 77.80±0.75 | 79.60±0.91 | 80.79±0.58 | 81.56±0.30 | 82.44±0.51 | 82.68±0.64 | 82.37±0.59 |
| Airport | 97.09±0.87 | 96.67±0.91 | 95.80±2.03 | 94.04±3.62 | 91.66±6.34 | 89.24±7.87 | 84.36±8.04 | 79.29±6.01 | 78.73±6.33 | 78.88±9.67 |

## D.9 ROBUSTNESS AGAINST ADVERSARIAL ATTACKS

Despite the significant advancements GNNs have made in inference tasks on graph-structured data, they are recognized as being susceptible to adversarial attacks (Zügner et al., 2018). Adversaries, aiming to deceive a trained GNN, can either introduce new nodes into the graph during the inference phase, known as an injection attack (Wang et al., 2020; Zheng et al., 2022; Zou et al., 2021; Hussain et al., 2022), or manipulate the graph's topology by adding or removing edges, termed as a modification attack (Chen et al., 2018a; Waniek et al., 2018; Du et al., 2017). In this section, we present preliminary experiments assessing the robustness of our model against adversarial attacks. Specifically, we carry out graph modification adversarial attacks using the Metattack method (Zügner & Günnemann, 2019). Our approach adheres to the attack setting described in Pro-GNN (Jin et al., 2020), and we utilize the perturbed graph provided by the DeepRobust library (Li et al., 2020b) to ensure a fair comparison. The perturbation rate, indicating the proportion of altered edges, is incrementally adjusted in 5% steps from 0% to 25%.

The results of these experiments are presented in Table 16. It should be noted that the impact of Meta-attacks with higher strengths detrimentally affects the performance of all models under test. However, our FROND-nl model consistently demonstrates enhanced resilience against adversarial attacks compared to the baselines, including GRAND-nl. For instance, at a perturbation rate of 25%, F-GRAND-nl outshines the baselines by an estimated margin of 10-15% on the Cora dataset.

Comprehensive testing against various adversarial attack methods and a theoretical understanding are detailed in our recent work (Kang et al., 2024).

Table 16: Node classification accuracy (%) under **modification, poisoning, non-targeted** attack (Metattack) in **transductive** learning.

| Dataset | Ptb Rate(%) | GGN | GAT | GRAND-nl | F-GRAND-nl |
|---|---|---|---|---|---|
| Cora | 0 | 83.50±0.44 | 83.97±0.65 | 83.14±1.06 | 83.48±1.08 |
| | 5 | 76.55±0.79 | 80.44±0.74 | 80.54±1.17 | 80.25±0.90 |
| | 10 | 70.39±1.28 | 75.61±0.59 | 76.59±1.21 | 77.94±0.48 |
| | 15 | 65.10±0.71 | 69.78±1.28 | 71.62±1.39 | 75.14±1.16 |
| | 20 | 59.56±2.72 | 59.94±0.92 | 57.52±1.20 | 69.04±1.13 |
| | 25 | 47.53±1.96 | 54.78±0.74 | 53.70±1.91 | 63.40±1.44 |
| Citeseer | 0 | 71.96±0.55 | 73.26±0.83 | 71.40±1.08 | 70.14±0.83 |
| | 5 | 70.88±0.62 | 72.89±0.83 | 70.99±1.12 | 70.0±1.72 |
| | 10 | 67.55±0.89 | 70.63±0.48 | 68.83±1.31 | 68.64±1.11 |
| | 15 | 64.52±1.11 | 69.02±1.09 | 66.78±0.92 | 67.90±0.41 |
| | 20 | 62.03±3.49 | 61.04±1.52 | 58.95±1.33 | 65.84±0.75 |
| | 25 | 56.94±2.09 | 61.85±1.12 | 60.52±1.29 | 66.50±1.16 |

## D.10 COMPARISON BETWEEN RIEMANN-LIOUVILLE (RL) DERIVATIVE AND CAPUTO DERIVATIVE

The underlying rationale for opting for the Caputo derivative over the Riemann-Liouville (RL) derivative is extensively delineated in Appendix B.4. However, a supplementary experiment was conducted utilizing the RL derivative in lieu of the Caputo derivative, the results of which are documented in Table 17. It can be observed that the task accuracies for both approaches are very similar. Further investigations on the use of different fractional derivatives and how to optimize the whole model architecture to adapt to a particular choice will be explored in future work.

Table 17: Comparison between RL-GRAND-l (using Riemann-Liouville derivative) and the original F-GRAND-l (using Caputo derivative).

| Method | Cora | Citeseer | Pubmed | CoauthorCS | Computer | Photo | CoauthorPhy | Airport | Disease |
|---|---|---|---|---|---|---|---|---|---|
| GRAND-l | 83.6±1.0 | 73.4±0.5 | 78.8±1.7 | 92.9±0.4 | 83.7±1.2 | 92.3±0.9 | 93.5±0.9 | 80.5±9.6 | 74.5±3.4 |
| RL-GRAND-l | 84.6±1.2 | 74.2±1.0 | 80.1±1.2 | 92.8±0.3 | 87.4±1.1 | 93.3±0.7 | 94.1±0.3 | 96.2±0.2 | 90.7±1.3 |
| F-GRAND-l | 84.8±1.1 | 74.0±1.5 | 79.4±1.5 | 93.0±0.3 | 84.4±1.5 | 92.8±0.6 | 94.5±0.4 | 98.1±0.2 | 92.4±3.9 |

## D.11 FRACTAL DIMENSION OF GRAPH DATASETS

Table 18: Comparison between the estimated fractal dimension, the best order $\beta$ and the $\delta$-hyperbolicity

| Dataset | Disease | Airport | Pubmed | Citeseer | Cora |
|---|---|---|---|---|---|
| fractal dimension | 2.47 | 2.17 | 2.25 | 0.62 | 1.22 |
| best $\beta$ (F-GRAND-l) | 0.6 | 0.5 | 0.9 | 0.9 | 0.9 |
| best $\beta$ (F-GRAND-nl) | 0.7 | 0.1 | 0.4 | 0.9 | 0.9 |
| $\delta$-hyperbolicity | 0.0 | 1.0 | 3.5 | 4.5 | 11.0 |

In Fig. 5, using the Compact-Box-Burning algorithm from (Song et al., 2007), we compute the fractal dimension for some datasets that have moderate sizes. As noted in Table 1, there is a clear trend between $\delta$-hyperbolicity (as referenced in (Chami et al., 2019) for assessing tree-like structures—with lower values suggesting more tree-like graphs) and the fractal dimension of datasets. Specifically, a lower $\delta$-hyperbolicity corresponds to a larger fractal dimension. As discussed in Sections 1 and 4, we believe that our fractional derivative $D_t^{\beta}$ effectively captures the fractal geometry in the datasets. Notably, we discerned a trend: a larger fractal dimension typically corresponds to a smaller optimal $\beta$.

## E MORE DYNAMICS IN FROND FRAMEWORK

### E.1 REVIEW OF GRAPH ODE MODELS

**GRAND++:** The work by (Thorpe et al., 2022) introduces graph neural diffusion with a source term, aimed at graph learning in scenarios with a limited quantity of labeled nodes. This approach leverages a subset of feature vectors, those associated with labeled nodes, indexed by $\mathcal{I}$, and considered "trustworthy" to act as a source term. It adheres to (4) and (5), incorporating an additional source term, facilitating the propagation of information from nodes in $\mathcal{I}$ to node $i$.

$$\frac{d\mathbf{X}(t)}{dt} = F(\mathbf{X}(t)) + s(\{\mathbf{x}_i\}_{i \in \mathcal{I}}) \tag{37}$$

Here, $\mathcal{I}$ denotes the set of source nodes, $s(\cdot)$ represents a source function, and $F(\cdot)$ embodies the function depicting the right-hand side of (4) and (5). The model is manifested in two variations, respectively denoted as GRAND++-nl and GRAND++-l.

**GraphCON:** Inspired by oscillator dynamical systems, GraphCON (Rusch et al., 2022) is defined through the employment of second-order ODEs. It is crucial to highlight that, for computation, the

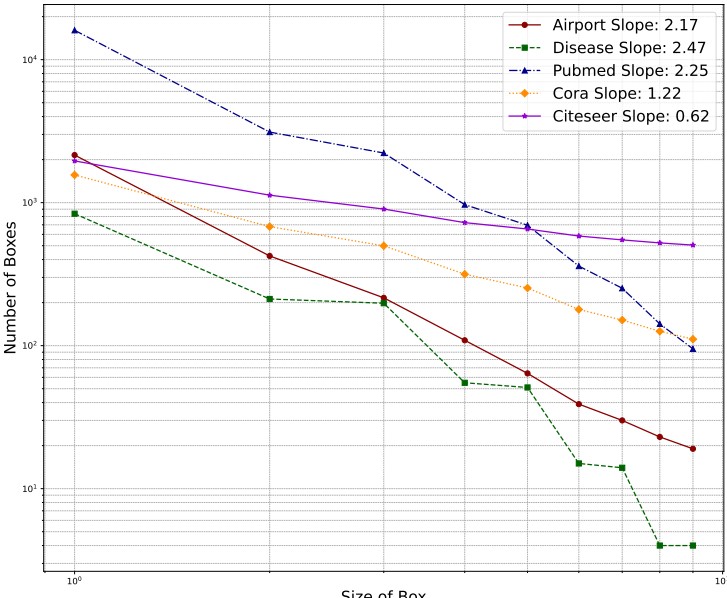

Figure 5: The fractal dim of datasets. We use the Compact-Box-Burning algorithm in (Song et al., 2007) to compute the log-log slope (fractal dim) of the box size and the minimum number of boxes needed to cover the graph.

second-order ODE is decomposed into two first-order ODEs:

$$\frac{\mathrm{d}\mathbf{Y}(t)}{\mathrm{d}t} = \sigma(\mathbf{F}_\theta(\mathbf{X}(t), t)) - \gamma \mathbf{X}(t) - \tilde{\alpha}\mathbf{Y}(t), \quad \frac{\mathrm{d}\mathbf{X}(t)}{\mathrm{d}t} = \mathbf{Y}(t), \tag{38}$$

where $\sigma(\cdot)$ is the activation function, $\mathbf{F}_\theta(\mathbf{X}(t), t)$ is the neural network function with parameters $\theta$, $\gamma$ and $\tilde{\alpha}$ are learnable coefficients, and $\mathbf{Y}(t)$ is the velocity term converting the second-order ODE to two first-order ODEs.

Analogous to the GRAND model, the GraphCON model is also available in both linear (GraphCON-l) and non-linear (GraphCON-nl) versions concerning time. The differentiation between these versions is determined by whether the function $\mathbf{F}_\theta$ undergoes updates based on time $t$.

**CDE:** With the objective of addressing heterophilic graphs, the paper (Zhao et al., 2023a) integrates the concept of convection-diffusion equations (CDE) into GNNs, leading to the proposition of the neural CDE model: This innovative model incorporates a convection term and introduces a unique velocity for each node, aiming to preserve diversity in heterophilic graphs. The corresponding formula is illustrated in (39).

$$\frac{\mathrm{d}\mathbf{X}(t)}{\mathrm{d}t} = (\mathbf{A}(\mathbf{X}(t)) - \mathbf{I})\mathbf{X}(t) + \mathrm{div}(\mathbf{V}(t) \circ \mathbf{X}(t)) \tag{39}$$

In this equation, $\mathbf{V}(t)$ represents the velocity field of the graph at time $t$, $\mathrm{div}(\cdot)$ denotes the divergence operator as defined in the paper (Chamberlain et al., 2021c; Song et al., 2022), and $\circ$ symbolizes the element-wise (Hadamard) product.

**GREAD:** To address the challenges posed by heterophilic graphs, the authors in (Choi et al., 2023) present the GREAD model. This model enhances the GRAND model by incorporating a reaction term, thereby formulating a diffusion-reaction equation within GNNs. The respective formula is depicted in (40), and the paper offers various alternatives for the reaction term.

$$\frac{\mathrm{d}\mathbf{X}(t)}{\mathrm{d}t} = -\alpha \mathbf{L}(\mathbf{X}(t)) + \alpha r(\mathbf{X}(t)) \tag{40}$$

In this equation, $r(\mathbf{X}(t))$ represents the reaction term, and $\alpha$ is a trainable parameter used to balance the impact of each term.

## E.2 F-GRAND++

Building upon the GRAND++ model (Thorpe et al., 2022), we define F-GRAND++ as follows:

$$D_t^\beta \mathbf{X}(t) = F(\mathbf{X}(t)) + s(\{\mathbf{x}_i\}_{i \in \mathcal{I}}) \tag{41}$$

We follow the same experimental settings as delineated in the GRAND++ paper. Given that the primary focus of GRAND++ is the model's performance under limited-label scenarios, our experiments also align with this setting. The sole distinction lies in the incorporation of fractional dynamics. Within this framework, we substitute the ordinary differential equation $\frac{d\mathbf{X}(t)}{dt}$ used in GRAND++ with our FROND fractional derivative $D_t^\beta \mathbf{X}(t)$. The optimal $\beta$ is determined through hyperparameter tuning. When $\beta = 1$, F-GRAND++ seamlessly reverts to GRAND++, and the results from the original paper are reported. Our observations distinctly indicate that the Fractional-GRAND++ consistently surpasses the performance of the original GRAND++ in nearly all scenarios. We also present the complete comparison results in Table 20, where it is evident that F-GRAND++ demonstrates greater effectiveness in learning with low labeling rates compared to GRAND++, GRAND, and other baseline methods.

Table 19: Node classification results (%) under limited-label scenarios

| Model | pre class | Cora | Citeseer | Pubmed | CoauthorCS | Computer | Photo |
|---|---|---|---|---|---|---|---|
| GRAND++ | 1 | 54.94±16.09 | 58.95±9.59 | 65.94±4.87 | 60.30±1.50 | 67.65±0.37 | 83.12±0.78 |
| F-GRAND++ | 1 | **57.31±8.89** | **59.11±6.73** | **65.98±2.72** | **67.71±1.91** | 67.65±0.37 | 83.12±0.78 |
|  | $\beta$ | 0.95 | 0.95 | 0.85 | 0.7 | 1.0 | 1.0 |
| GRAND++ | 2 | 66.92±10.04 | 64.98±8.31 | 69.31±4.87 | 76.53±1.85 | 74.47±1.48 | 83.71±0.90 |
| F-GRAND++ | 2 | **70.09±8.36** | 64.98±8.31 | **69.37±5.36** | **77.97±2.35** | **78.85±0.96** | 83.71±0.90 |
|  | $\beta$ | 0.9 | 1.0 | 0.95 | 0.5 | 0.8 | 1.0 |
| GRAND++ | 5 | 77.80±4.46 | 70.03±3.63 | 71.99±1.91 | 84.83±0.84 | 82.64±0.56 | 88.33±1.21 |
| F-GRAND++ | 5 | **78.79±1.66** | **70.26±2.36** | **73.38±5.67** | **86.09±2.09** | 82.64±0.56 | **88.56±0.67** |
|  | $\beta$ | 0.9 | 0.8 | 0.9 | 0.8 | 1.0 | 0.75 |
| GRAND++ | 10 | 80.86±2.99 | 72.34±2.42 | 75.13±3.88 | 86.94±0.46 | 82.99±0.81 | 90.65±1.19 |
| F-GRAND++ | 10 | **82.73±0.81** | **73.52±1.44** | **77.15±2.87** | **87.85±1.44** | **83.26±0.41** | **91.15±0.52** |
|  | $\beta$ | 0.95 | 0.9 | 0.95 | 0.6 | 0.7 | 0.95 |
| GRAND++ | 20 | 82.95±1.37 | 73.53±3.31 | 79.16±1.37 | 90.80±0.34 | 85.73±0.50 | 93.55±0.38 |
| F-GRAND++ | 20 | **84.57±1.07** | **74.81±1.78** | **79.96±1.68** | **91.03±0.72** | **85.78±0.43** | 93.55±0.38 |
|  | $\beta$ | 0.9 | 0.85 | 0.95 | 0.9 | 0.9 | 1.0 |

## E.3 F-CDE

Drawing inspiration from the graph neural CDE model (Zhao et al., 2023a), we further define the F-CDE model as follows:

$$D_t^\beta \mathbf{X}(t) = (\mathbf{A}(\mathbf{X}(t)) - \mathbf{I})\mathbf{X}(t) + \mathrm{div}(\mathbf{V}(t) \circ \mathbf{X}(t)) \tag{42}$$

In this expression, $\mathbf{V}(t)$ represents the velocity field of the graph at time $t$. The divergence operator, $\mathrm{div}(\cdot)$, is defined as per the formulation given in (Song et al., 2022), and $\circ$ symbolizes the element-wise (Hadamard) product.

We follow the same experimental setting as in the CDE paper(Zhao et al., 2023a). Given that the primary focus of CDE is on evaluating model performance on large heterophilic datasets, our experiments are also conducted under similar conditions. The statistics for the dataset are available in Table 21. The sole distinction in our approach lies in incorporating fractional dynamics; we achieve this by replacing the ODE used in CDE with our FROND fractional derivative. The complete comparison results in Table 22 conspicuously reveal that Fractional CDE exhibits superior performance compared to the conventional CDE and other baselines across various datasets.

Table 20: Full table: Classification accuracy of different GNNs trained with different number of labeled data per class (#per class) on six benchmark graph node classification tasks. The highest accuracy is highlighted in bold for each number of labeled data per class. These results show that F-GRAND++ is more effective in learning with low-labeling rates than GRAND++ and GRAND. Where available, baseline results are cited from (Thorpe et al., 2022).

| Model | #per class | CORA | CiteSeer | PubMed | CoauthorCS | Computer | Photo |
|---|---|---|---|---|---|---|---|
| F-GRAND++ | 1 | **57.31 ± 8.89** | **59.11 ± 6.73** | **65.98 ± 2.72** | **67.71 ± 1.91** | **67.65 ± 0.37** | **83.12 ± 0.78** |
| | 2 | **70.09 ± 8.36** | **64.98 ± 8.31** | **69.37 ± 5.36** | 77.97 ± 2.35 | **78.85 ± 0.96** | **83.71 ± 0.90** |
| | 5 | **78.79 ± 1.66** | **70.26 ± 2.36** | 73.38 ± 5.67 | 86.09 ± 2.09 | **82.64 ± 0.56** | 88.56 ± 0.67 |
| | 10 | **82.73 ± 0.81** | **73.52 ± 1.44** | **77.15 ± 2.87** | 87.85 ± 1.44 | **83.26 ± 0.41** | **91.15 ± 0.52** |
| | 20 | **84.57 ± 1.07** | **74.81 ± 1.78** | **79.96 ± 1.68** | 91.03 ± 0.72 | **85.78 ± 0.43** | **93.55 ± 0.38** |
| GRAND++ | 1 | 54.94 ± 16.09 | 58.95 ± 9.59 | 65.94 ± 4.87 | 60.30 ± 1.50 | **67.65 ± 0.37** | **83.12 ± 0.78** |
| | 2 | 66.92 ± 10.04 | **64.98 ± 8.31** | 69.31 ± 4.87 | 76.53 ± 1.85 | 76.47 ± 1.48 | **83.71 ± 0.90** |
| | 5 | 77.80 ± 4.46 | 70.03 ± 3.63 | 71.99 ± 1.91 | 84.83 ± 0.84 | 82.64 ± 0.56 | 88.33 ± 1.21 |
| | 10 | 80.86 ± 2.99 | 72.34 ± 2.42 | 75.13 ± 3.88 | 86.94 ± 0.46 | 82.99 ± 0.81 | 90.65 ± 1.19 |
| | 20 | 82.95 ± 1.37 | 73.53 ± 3.31 | 79.16 ± 1.37 | 90.80 ± 0.34 | 85.73 ± 0.50 | **93.55 ± 0.38** |
| GRAND | 1 | 52.53 ± 16.40 | 50.06 ± 17.98 | 62.11 ± 10.58 | 59.15 ± 5.73 | 48.67 ± 1.66 | 81.25 ± 2.50 |
| | 2 | 64.82 ± 11.16 | 59.55 ± 10.89 | 69.00 ± 7.55 | 73.83 ± 5.58 | 74.77 ± 1.85 | 82.13 ± 3.27 |
| | 5 | 76.07 ± 5.08 | 68.37 ± 5.00 | **73.98 ± 5.08** | 85.29 ± 2.19 | 80.72 ± 1.09 | 88.27 ± 1.94 |
| | 10 | 80.25 ± 3.40 | 71.90 ± 7.66 | 76.33 ± 3.41 | 87.81 ± 1.36 | 82.42 ± 1.10 | 90.98 ± 0.93 |
| | 20 | 82.86 ± 2.39 | 73.02 ± 5.89 | 78.76 ± 1.69 | 91.03 ± 0.47 | 84.54 ± 0.90 | 93.53 ± 0.47 |
| GCN | 1 | 47.72 ± 15.33 | 48.94 ± 10.24 | 58.61 ± 12.83 | 65.22 ± 2.25 | 49.46 ± 1.65 | 82.94 ± 2.17 |
| | 2 | 60.85 ± 14.01 | 58.06 ± 9.76 | 60.45 ± 16.20 | **83.61 ± 1.49** | 76.90 ± 1.49 | 83.61 ± 0.71 |
| | 5 | 73.86 ± 7.97 | 67.24 ± 4.19 | 68.69 ± 7.93 | 86.66 ± 0.43 | 82.47 ± 0.97 | **88.86 ± 1.56** |
| | 10 | 78.82 ± 5.38 | 72.18 ± 3.47 | 72.59 ± 3.19 | 88.60 ± 0.50 | 82.53 ± 0.74 | 90.41 ± 0.35 |
| | 20 | 82.07 ± 2.03 | 74.21 ± 2.90 | 76.89 ± 3.27 | 91.09 ± 0.35 | 82.94 ± 1.54 | 91.95 ± 0.11 |
| GAT | 1 | 47.86 ± 15.38 | 50.31 ± 14.27 | 58.84 ± 12.81 | 51.13 ± 5.24 | 37.14 ± 7.81 | 73.58 ± 8.15 |
| | 2 | 58.30 ± 13.55 | 55.55 ± 9.19 | 60.24 ± 14.44 | 63.12 ± 6.09 | 65.07 ± 8.86 | 76.89 ± 4.89 |
| | 5 | 71.04 ± 5.74 | 67.37 ± 5.08 | 68.54 ± 5.75 | 71.65 ± 4.53 | 71.43 ± 7.34 | 83.01 ± 3.64 |
| | 10 | 76.31 ± 4.87 | 71.35 ± 4.92 | 72.44 ± 3.50 | 74.71 ± 3.35 | 76.04 ± 0.35 | 87.42 ± 2.38 |
| | 20 | 79.92 ± 2.28 | 73.22 ± 2.90 | 75.55 ± 4.11 | 79.95 ± 2.88 | 80.05 ± 1.81 | 89.38 ± 2.48 |
| GraphSage | 1 | 43.04 ± 14.01 | 48.81 ± 11.45 | 55.53 ± 12.71 | 61.35 ± 1.35 | 27.65 ± 2.39 | 45.36 ± 7.13 |
| | 2 | 53.96 ± 12.18 | 54.39 ± 11.37 | 58.97 ± 12.65 | 76.51 ± 1.31 | 42.63 ± 4.29 | 51.93 ± 4.21 |
| | 5 | 68.14 ± 6.95 | 64.79 ± 5.16 | 66.07 ± 6.16 | **89.06 ± 0.69** | 64.83 ± 1.62 | 78.26 ± 1.93 |
| | 10 | 75.04 ± 5.03 | 68.90 ± 5.08 | 70.74 ± 3.11 | **89.68 ± 0.39** | 74.66 ± 1.29 | 84.38 ± 1.75 |
| | 20 | 80.04 ± 2.54 | 72.02 ± 2.82 | 74.55 ± 3.09 | **91.33 ± 0.36** | 79.98 ± 0.96 | 91.29 ± 0.67 |
| MoNet (Monti et al., 2017) | 1 | 47.72 ± 15.53 | 39.13 ± 11.37 | 56.47 ± 4.67 | 58.99 ± 5.17 | 23.78 ± 7.57 | 34.72 ± 8.18 |
| | 2 | 60.85 ± 14.01 | 48.52 ± 9.52 | 61.03 ± 6.93 | 76.57 ± 4.06 | 38.19 ± 3.72 | 43.03 ± 8.22 |
| | 5 | 73.86 ± 7.97 | 61.66 ± 6.61 | 67.92 ± 2.50 | 87.02 ± 1.67 | 59.38 ± 4.73 | 71.80 ± 5.02 |
| | 10 | 78.82 ± 5.38 | 68.08 ± 6.29 | 71.24 ± 1.54 | 88.76 ± 0.49 | 68.66 ± 3.30 | 78.66 ± 3.17 |
| | 20 | 82.07 ± 2.03 | 71.52 ± 4.11 | 76.49 ± 1.75 | 90.31 ± 0.41 | 73.66 ± 2.87 | 88.61 ± 1.18 |

Table 21: Dataset statistics used in Table 4

| Dataset | Nodes | Edges | Classes | Node Features |
|---|---|---|---|---|
| Roman-empire | 22662 | 32927 | 18 | 300 |
| Wiki-cooc | 10000 | 2243042 | 5 | 100 |
| Minesweeper | 10000 | 39402 | 2 | 7 |
| Questions | 48921 | 153540 | 2 | 301 |
| Workers | 11758 | 519000 | 2 | 10 |
| Amaon-ratings | 24492 | 93050 | 5 | 300 |

Table 22: Full table: Node classification accuracy(%) of large heterophilic datasets.

| Model | Roman-empire | Wiki-cooc | Minesweeper | Questions | Workers | Amazon-ratings |
|---|---|---|---|---|---|---|
| ResNet | 65.71±0.44 | 89.36±0.71 | 50.95±1.12 | 70.10±0.75 | 73.08±1.28 | 45.70±0.69 |
| H2GCN(Zhu et al., 2020a) | 68.09±0.29 | 89.24±0.32 | 89.95±0.38 | 66.66±1.84 | 81.76±0.68 | 41.36±0.47 |
| CPGNN(Zhu et al., 2021) | 63.78±0.50 | 84.84±0.66 | 71.27±1.14 | 67.09±2.63 | 72.44±0.80 | 44.36±0.35 |
| GPR-GNN(Chien et al., 2020) | 73.37±0.68 | 91.90±0.78 | 81.79±0.98 | 73.41±1.24 | 70.59±1.15 | 43.90±0.48 |
| GloGNN(Li et al., 2022) | 63.85±0.49 | 88.49±0.45 | 62.53±1.34 | 67.15±1.92 | 73.90±0.95 | 37.28±0.66 |
| FAGCN(Bo et al., 2021) | 70.53±0.94 | 91.88±0.37 | 89.69±0.60 | **77.04±1.56** | 81.87±0.94 | 46.32±2.50 |
| GBK-GNN(Du et al., 2022) | 75.87±0.43 | 97.81±0.32 | 83.56±0.84 | 72.98±1.05 | 78.06±0.91 | 43.47±0.51 |
| ACM-GCN(Luan et al., 2022) | 68.35±1.95 | 87.48±1.06 | 90.47±0.57 | OOM | 78.25±0.78 | 38.51±3.38 |
| GRAND(Chamberlain et al., 2021a) | 71.60±0.58 | 92.03±0.46 | 76.67±0.98 | 70.67±1.28 | 75.33±0.84 | 45.05±0.65 |
| GraphBel(Song et al., 2022) | 69.47±0.37 | 90.30±0.50 | 76.51±1.03 | 70.79±0.99 | 73.02±0.92 | 43.63±0.42 |
| Diag-NSD(Bodnar et al., 2022) | 77.50±0.67 | 92.06±0.40 | 89.59±0.61 | 69.25±1.15 | 79.81±0.99 | 37.96±0.20 |
| ACMP(Wang et al., 2022b) | 71.27±0.59 | 92.68±0.37 | 76.15±1.12 | 71.18±1.03 | 75.03±0.92 | 44.76±0.52 |
| CDE | 91.64±0.28 | 97.99±0.38 | 95.50±5.23 | 75.17±0.99 | 80.70±1.04 | 47.63±0.43 |
| F-CDE | **93.06±0.55** | **98.73±0.68** | **96.04±0.25** | 75.17±0.99 | **82.68±0.86** | **49.01±0.56** |
| $\beta$ for F-CDE | 0.9 | 0.6 | 0.6 | 1.0 | 0.4 | 0.1 |

### E.4 F-GREAD

Our FROND framework is also extendable to the GREAD model (Choi et al., 2023), as defined in (43).

$$D_t^\beta \mathbf{X}(t) = -\alpha \mathbf{L}(\mathbf{X}(t)) + \alpha r(\mathbf{X}(t)) \tag{43}$$

where $r(\mathbf{X}(t))$ represents a reaction term, and $\alpha$ is a trainable parameter used to emphasize each term.

We adhere to the same experimental setting outlined in the GREAD paper (Choi et al., 2023), concentrating exclusively on heterophilic datasets. We choose the Blurring-Sharpening (BS) as the reaction term to formulate both GREAD-BS and F-GREAD-BS, as GREAD-BS exhibits strong performance according to Table 4 in the GREAD paper (Choi et al., 2023). The results presented in Table 23 (refer to Table 24 for comprehensive comparisons with other baselines) demonstrate that our FROND framework enhances the performance of GREAD across all examined datasets.

Table 23: Node classification accuracy(%) of heterophilic datasets

| Model | Chameleon | Squirrel | Film | Texas | Wisconsin |
|---|---|---|---|---|---|
| GREAD-BS | 71.38±1.31 | 59.22±1.44 | 37.90±1.17 | 88.92±3.72 | 89.41±3.30 |
| F-GREAD-BS | **71.45±1.98** | **60.86±1.05** | **38.28±0.74** | **92.97±4.39** | **90.59±3.80** |
| $\beta$ | 0.9 | 0.9 | 0.8 | 0.9 | 0.9 |

Table 24: Full table: Node classification accuracy(%) of heterophilic datasets

| Model | Chameleon | Squirrel | Film | Texas | Wisconsin |
|---|---|---|---|---|---|
| Geom-GCN(Pei et al., 2020) | 60.00±2.81 | 38.15±0.92 | 31.59±1.15 | 66.76±2.72 | 64.51±3.66 |
| H2GCN(Zhu et al., 2020a) | 60.11±2.15 | 36.48±1.86 | 35.70±1.00 | 84.86±7.23 | 87.65±4.98 |
| GGCN(Yan et al., 2022) | 71.14±1.84 | 55.17±1.58 | 37.54±1.56 | 84.86±4.55 | 86.86±3.29 |
| LINKX(Lim et al., 2021) | 68.42±1.38 | **61.81±1.80** | 36.10±1.55 | 74.60±8.37 | 75.49±5.72 |
| GloGNN(Li et al., 2022) | 69.78±2.42 | 57.54±1.39 | 37.35±1.30 | 84.32±4.15 | 87.06±3.53 |
| ACM-GCN(Luan et al., 2022) | 66.93±1.85 | 54.40±1.88 | 36.28±1.09 | 87.84±4.40 | 88.43±3.22 |
| GCNII(Chen et al., 2020) | 63.86±3.04 | 38.47±1.58 | 37.44±1.30 | 77.57±3.83 | 80.39±3.40 |
| CGNN(Xhonneux et al., 2020) | 46.89±1.66 | 29.24±1.09 | 35.95±0.86 | 71.35±4.05 | 74.31±7.26 |
| GRAND(Chamberlain et al., 2021a) | 54.67±2.54 | 40.05±1.50 | 35.62±1.01 | 75.68±7.25 | 79.41±3.64 |
| BLEND(Chamberlain et al., 2021b) | 60.11±2.09 | 43.06±1.39 | 35.63±1.01 | 83.24±4.65 | 84.12±3.56 |
| Sheaf(Bodnar et al., 2022) | 68.04±1.58 | 56.34±1.32 | 37.81±1.15 | 85.05±5.51 | 89.41±4.74 |
| GRAFF(Di Giovanni et al., 2022) | 71.08±1.75 | 54.52±1.37 | 36.09±0.81 | 88.38±4.53 | 87.45±2.94 |
| GREAD-BS | 71.38±1.31 | 59.22±1.44 | 37.90±1.17 | 88.92±3.72 | 89.41±3.30 |
| F-GREAD-BS | **71.45±1.98** | 60.86±1.05 | **38.28±0.74** | **92.97±4.39** | **90.59±3.80** |
| $\beta$ | 0.9 | 0.9 | 0.8 | 0.9 | 0.9 |

### E.5 F-GRAPHCON

We also incorporate the following fractional-order oscillators dynamics, inspired by (Radwan et al., 2008; Rusch et al., 2022):

$$\begin{aligned} D_t^\beta \mathbf{Y} &= \sigma\left(\mathbf{F}_\theta(\mathbf{X}, t)\right) - \gamma \mathbf{X} - \alpha \mathbf{Y} \\ D_t^\beta \mathbf{X} &= \mathbf{Y} \end{aligned} \tag{44}$$

which represent the fractional dynamics version of GraphCON (Rusch et al., 2022). We denote this as F-GraphCON, with two variants, F-GraphCON-GCN and F-GraphCON-GAT. Here, $\mathbf{F}_\theta$ is set as GCN and GAT, as in the setting described in (Rusch et al., 2022). We refer readers to (Rusch et al., 2022) for further details. Notably, when $\beta = 1$, F-GraphCON simplifies to GraphCON, devoid of memory functionality.

Table 25: Node classification accuracy(%) based on GraphCON model

|  | Cora | Citeseer | Pubmed | Airport | Disease |
|---|---|---|---|---|---|
| GraphCON-GCN | 81.9±1.7 | 72.9±2.1 | 78.8±2.6 | 68.6±2.1 | 87.5±4.1 |
| GraphCON-GAT | 83.2±1.4 | 73.2±1.8 | 79.4±1.3 | 74.1±2.7 | 65.7±5.9 |
| F-GraphCON-GCN | **84.6±1.4** | **75.3±1.1** | **80.3±1.3** | **97.3±0.5** | **92.1±2.8** |
| $\beta$ | 0.9 | 0.8 | 0.9 | 0.1 | 0.1 |
| F-GraphCON-GAT | 83.9±1.2 | 73.4±1.5 | 79.4±1.3 | 97.3±0.8 | 86.9±4.0 |
| $\beta$ | 0.7 | 0.9 | 1.0 | 0.1 | 0.1 |

Table 26: Full table: Node classification accuracy(%) based on GraphCON model.

|  | Cora | Citeseer | Pubmed | Airport | Disease |
|---|---|---|---|---|---|
| GCN | 81.5±1.3 | 71.9±1.9 | 77.8±2.9 | 81.6±0.6 | 69.8±0.5 |
| GAT | 81.8±1.3 | 71.4±1.9 | 78.7±2.3 | 81.6±0.4 | 70.4±0.5 |
| HGCN | 78.7±1.0 | 65.8±2.0 | 76.4±0.8 | 85.4±0.7 | 89.9±1.1 |
| GIL | 82.1±1.1 | 71.1±1.2 | 77.8±0.6 | 91.5±1.7 | 90.8±0.5 |
| GRAND-l | 83.6±1.0 | 73.4±0.5 | 78.8±1.7 | 80.5±9.6 | 74.5±3.4 |
| GRAND-nl | 82.3±1.6 | 70.9±1.0 | 77.5±1.8 | 90.9±1.6 | 81.0±6.7 |
| GraphCON-GCN | 81.9±1.7 | 72.9±2.1 | 78.8±2.6 | 68.6±2.1 | 87.5±4.1 |
| GraphCON-GAT | 83.2±1.4 | 73.2±1.8 | 79.4±1.3 | 74.1±2.7 | 65.7±5.9 |
| F-GraphCON-GCN | **84.6±1.4** | **75.3±1.1** | **80.3±1.3** | **97.3±0.5** | **92.1±2.8** |
| $\beta$ | 0.9 | 0.8 | 0.9 | 0.1 | 0.1 |
| F-GraphCON-GAT | 83.9±1.2 | 73.4±1.5 | 79.4±1.3 | 97.3±0.8 | 86.9±4.0 |
| $\beta$ | 0.7 | 0.9 | 1.0 | 0.1 | 0.1 |

Table 27: Node classification accuracy(%) of undirected graphs based on F-FLODE model

|  | Film | Squirrel | Chameleon |
|---|---|---|---|
| FLODE | 37.16±1.42 | 64.23±1.84 | 73.60±1.55 |
| F-FLODE | **37.95±1.27** | **65.53±1.83** | **74.17±1.59** |
| $\beta$ | 0.8 | 0.9 | 0.9 |

Table 28: Node classification accuracy(%) of directed graphs based on F-FLODE model

|  | Film | Squirrel | Chameleon |
|---|---|---|---|
| FLODE | 37.41±1.06 | 74.03±1.58 | 77.98±1.05 |
| F-FLODE | **37.97±1.15** | **75.03±1.42** | **78.51±1.09** |
| $\beta$ | 0.9 | 0.9 | 0.9 |

## E.6 F-FLODE

In the work of (Maskey et al., 2023), the authors introduce the FLODE model, which incorporates fractional graph shift operators within integer-order continuous GNNs. Specifically, instead of utilizing a Laplacian matrix $\mathbf{L}$, they employ the fractional power of $\mathbf{L}$, denoted as $\mathbf{L}^\alpha$ (see (45)). Our research diverges from this approach, focusing on the incorporation of time-fractional derivative $D_t^\beta$ for updating graph node features in a memory-inclusive dynamical process. It is pivotal to differentiate the term "fractional" as used in our work from that in (Maskey et al., 2023), as they signify fundamentally distinct concepts in the literature. Fundamentally, FLODE differs from our work in key aspects:

- FLODE employs the fractional (real-valued) power of $\mathbf{L}$, namely $\mathbf{L}^\alpha$. The feature evolution model used by FLODE, specifically in its first heat diffusion-type variant, is given by:

$$\frac{\mathrm{d}\mathbf{X}(t)}{\mathrm{d}t} = -\mathbf{L}^\alpha \mathbf{X}(t)\mathbf{\Phi}. \tag{FLODE}$$

  This is a graph spatial domain rewiring technique, as $\mathbf{L}^\alpha$ introduces dense connections compared to $\mathbf{L}$. As a result, FLODE introduces space-based long-range interactions during the feature updating process.

- In contrast, our FROND model incorporates the time-fractional derivative $D_t^\beta$ to update graph node features in a memory-inclusive dynamical process. In this context, time acts as a continuous counterpart to the layer index, leading to significant dense skip connections between layers due to memory dependence. Thus, FROND induces time/layer-based long-range interactions in the feature update process. Note that FLODE does not utilize time-fractional derivatives. Our method is not only *compatible with various integer-order continuous GNNs, including FLODE (see* (F-FLODE)*)*, but also extends them to graph FDE models.

We next introduce the F-FLODE model, which utilizes time-fractional derivatives for updating graph node features in FLODE:

$$D_t^\beta \mathbf{X}(t) = -\mathbf{L}^\alpha \mathbf{X}(t)\mathbf{\Phi}, \tag{F-FLODE}$$

where $\mathbf{L}$ denotes the symmetrically normalized adjacency matrix. The $\alpha$-fractional power of the graph Laplacian, $\mathbf{L}^\alpha$, is given by:

$$\mathbf{L}^\alpha := \mathbf{U}\mathbf{\Sigma}^\alpha \mathbf{V}^{\mathrm{H}}. \tag{45}$$

In this formulation, $\mathbf{U}$, $\mathbf{\Sigma}$, and $\mathbf{V}$ are obtained from the SVD decomposition of $\mathbf{L} = \mathbf{U}\mathbf{\Sigma}\mathbf{V}^{\mathrm{H}}$, and $\alpha \in \mathbb{R}$ represents the order. The channel mixing matrix $\mathbf{\Phi}$, a symmetric matrix, follows the setting in (Maskey et al., 2023).

Following the experimental setup outlined in (Maskey et al., 2023), we present our results in Tables 27 and 28, demonstrating that our FROND framework enhances the performance of FLODE across all evaluated datasets. Note the difference in the equations in (FLODE) and (F-FLODE), where the two are equivalent only when $\beta = 1$. This example illustrates that the FROND framework encompasses the FLODE model as a special case when $\beta = 1$. Our experimental results indicate that F-FLODE outperforms FLODE with the optimal $\beta \neq 1$ in general.

# F  PROOFS OF RESULTS

In this section, we provide detailed proofs of the results stated in the main paper.

## F.1  PROOF OF THEOREM 1

*Proof.* We observe that for $0 < \beta < 1$ they possess the properties, the coefficients $c_k$, $b_m$ defined in (10) satisfying the following properties (Gorenflo et al., 2002).

$$\sum_{k=1}^\infty c_k = 1, \quad 1 > \beta = c_1 > c_2 > c_3 > \ldots \to 0,$$

$$b_0 = 1, \quad b_m = 1 - \sum_{k=1}^m c_k = \sum_{k=m+1}^\infty c_k, 1 = b_0 > b_1 > b_2 > b_3 > \ldots \to 0.$$

From the definition of the transition probability (11), we have

$$\mathbb{P}(\mathbf{R}(t_{n+1}) = \mathbf{x}_h)$$
$$= b_n\mathbb{P}(\mathbf{R}(t_0) = \mathbf{x}_h) + c_n\mathbb{P}(\mathbf{R}(t_1) = \mathbf{x}_h) + \ldots + c_2\mathbb{P}(\mathbf{R}(t_{n-1}) = \mathbf{x}_h)+$$
$$+ (c_1 - \sigma^\beta)\mathbb{P}(\mathbf{R}(t_n) = \mathbf{x}_h) + \sum_{j=1}^{n}\sigma^\beta\frac{W_{jh}}{d_j}\mathbb{P}(\mathbf{R}(t_n) = \mathbf{x}_j)$$
$$= b_n\mathbb{P}(\mathbf{R}(t_0) = \mathbf{x}_h) + c_n\mathbb{P}(\mathbf{R}(t_1) = \mathbf{x}_h) + \ldots + c_2\mathbb{P}(\mathbf{R}(t_{n-1}) = \mathbf{x}_h)+$$
$$+ c_1\mathbb{P}(\mathbf{R}(t_n) = \mathbf{x}_h) - \sigma^\beta\mathbb{P}(\mathbf{R}(t_n) = \mathbf{x}_h) + \sum_{j=1}^{N}\sigma^\beta\frac{W_{jh}}{d_j}\mathbb{P}(\mathbf{R}(t_n) = \mathbf{x}_j). \qquad (46)$$

By rearranging, we have

$$\mathbb{P}(\mathbf{R}(t_{n+1}) = \mathbf{x}_h) - \sum_{k=1}^{n}c_k\mathbb{P}(\mathbf{R}(t_{n+1-k}) = \mathbf{x}_h) - b_n\mathbb{P}(\mathbf{R}(t_0) = \mathbf{x}_h)$$

$$= (-1)^0\binom{\beta}{0}\mathbb{P}(\mathbf{R}(t_{n+1}) = \mathbf{x}_h) - \sum_{k=1}^{n}(-1)^{k+1}\binom{\beta}{k}\mathbb{P}(\mathbf{R}(t_{n+1-k}) = \mathbf{x}_h)$$

$$- \sum_{k=0}^{n}(-1)^k\binom{\beta}{k}\mathbb{P}(\mathbf{R} = \mathbf{x}_h)$$

$$= \sum_{k=0}^{n}(-1)^k\binom{\beta}{k}\mathbb{P}(\mathbf{R}(t_{n+1-k}) = \mathbf{x}_h) - \sum_{k=0}^{n}(-1)^k\binom{\beta}{k}\mathbb{P}(\mathbf{R} = \mathbf{x}_h)$$

$$= \sum_{k=0}^{n}(-1)^k\binom{\beta}{k}\left[\mathbb{P}(\mathbf{R}(t_{n+1-k}) = \mathbf{x}_h) - \mathbb{P}(\mathbf{R} = \mathbf{x}_h)\right]$$

$$= -\sigma^\beta\mathbb{P}(\mathbf{R}(t_n) = \mathbf{x}_h) + \sum_{j=1}^{n}\sigma^\beta\frac{W_{jh}}{d_j}\mathbb{P}(\mathbf{R}(t_n) = \mathbf{x}_j).$$

Dividing both sides of the final equality by $\sigma^\beta$, it follows that

$$\sum_{k=0}^{n}(-1)^k\binom{\beta}{k}\frac{\mathbb{P}(\mathbf{R}(t_{n+1-k}) = \mathbf{x}_h) - \mathbb{P}(\mathbf{R} = \mathbf{x}_h)}{\sigma^\beta}$$

$$= -\mathbb{P}(\mathbf{R}(t_n) = \mathbf{x}_h) + \sum_{j=1}^{N}\frac{W_{jh}}{d_j}\mathbb{P}(\mathbf{R}(t_n) = \mathbf{x}_j). \qquad (47)$$

From the Griinwald-Letnikov fractional derivatives formulation (Podlubny, 1999)[eq. (2.54)], the limit of LHS of (47) is

$$\lim_{\substack{\sigma \to 0 \\ n\sigma = t}}\sum_{k=0}^{n}(-1)^k\binom{\beta}{k}\frac{\mathbb{P}(\mathbf{R}(t_{n+1-k}) = \mathbf{x}_h) - \mathbb{P}(\mathbf{R} = \mathbf{x}_h)}{\sigma^\beta} = D_t^\beta\mathbb{P}(\mathbf{R}(t) = \mathbf{x}_h) \equiv [D_t^\beta\mathbf{P}(t)]_h.$$
$$(48)$$

where $\mathbf{P}(t) := \lim_{n \to \infty}\mathbb{P}(\mathbf{R}(t_n))$ and $[D_t^\beta\mathbf{P}(t)]_h$ denotes the $h$-th element of the vector. On the other hand, the RHS of (47) is

$$-\mathbb{P}(\mathbf{R}(t_n) = \mathbf{x}_h) + \sum_{j=1}^{N}\frac{W_{jh}}{d_j}\mathbb{P}(\mathbf{R}(t_n) = \mathbf{x}_j) = [-\mathbf{L}\mathbb{P}(\mathbf{R}(t_n))]_h \qquad (49)$$

where $\mathbb{P}(\mathbf{R}(t_n))$ is the probability (column) vector with $j$-th element being $\mathbb{P}(\mathbf{R}(t_n) = \mathbf{x}_j)$, and $[-\mathbf{L}\mathbb{P}(\mathbf{R}(t_n))]_h$ denotes the $h$-th element of the vector $-\mathbf{L}\mathbb{P}(\mathbf{R}(t_n))$.

Putting them together, we have

$$D_t^\beta\mathbf{P}(t) = -\mathbf{L}\mathbf{P}(t) \qquad (50)$$

since we assume $t_n = t$ in the limit. The proof of Theorem 1 is now complete. $\qquad\square$

### F.2 PROOF OF COROLLARY 1

It directly follows from the the linearity of FDEs and $\mathbf{X}(0) = \mathbf{X} = \sum_i {}_i\mathbf{P}(0)\mathbf{x}_i$ where recall that the initial probability vector ${}_i\mathbb{P}(\mathbf{R}(0)) \equiv {}_i\mathbf{P}(0)$ is represented as a one-hot vector with the $i$-th entry marked as 1.

### F.3 PROOF OF THEOREM 2

Before presenting the formal proof, we aim to provide additional insights and intuition regarding the algebraic convergence from two perspectives.

- Fractional Random Walk Perspective: In a standard random walk, a walker moves to a new position at each time step without delay. However, in a fractional random walk, which is more reflective of our model's behavior, the walker has a probability of revisiting past positions. This revisitation is not arbitrary; it is governed by a waiting time that follows a power-law distribution with a long tail. This characteristic fundamentally changes the walk's dynamics, introducing a memory component and leading to a slower, algebraic rate of convergence. This behavior is intrinsically different from normal random walks, where the absence of waiting times facilitates a quicker, exponential, convergence.

- Analytic Perspective: From an analytic perspective, the essential slow algebraic rate primarily stems from the slow convergence of the Mittag-Leffler function towards zero. To elucidate this, let us consider the scalar scenario. Recall that the Mittag-Leffler function $E_\beta$ is defined as:

$$E_\beta(z) := \sum_{j=0}^{\infty} \frac{z^j}{\Gamma(j\beta + 1)}$$

for values of $z$ where the series converges. Specifically, when $\beta = 1$,

$$E_1(z) = \sum_{j=0}^{\infty} \frac{z^j}{\Gamma(j+1)} = \sum_{j=0}^{\infty} \frac{z^j}{j!} = \exp(z)$$

corresponds to the well-known exponential function. According to [A1, Theorem 4.3.], the eigenfunctions of the Caputo derivative are expressed through the Mittag-Leffler function. In more precise terms, if we define $y(t)$ as

$$y(t) := E_\beta\left(-\lambda t^n\right), \quad t \geq 0,$$

it follows that

$$D_t^\beta y(t) = -\lambda y(t).$$

Notably, when $\beta = 1$, this reduces to $\frac{d\exp(-\lambda t)}{dt} = -\lambda\exp(-\lambda t)$. We examine the behavior of $E_\beta\left(-\lambda t^n\right)$. From (Diethelm, 2010)[Theorem 7.3.], when $0 < \beta < 1$, it is noted that:

(a) The function $y(t)$ is completely monotonic on $(0, \infty)$.

(b) As $x \to \infty$,

$$y(t) = \frac{t^{-\beta}}{\lambda\Gamma(1-\beta)}(1 + o(1)).$$

Thus, the function $E_\beta\left(-\lambda t^\beta\right)$ converges to zero at a rate of $\Theta\left(t^{-\beta}\right)$. Our paper extends this to the general high-dimensional case by replacing the scalar $\lambda$ with the Laplacian matrix $\mathbf{L}$, wherein the eigenvalues of $\mathbf{L}$ play a critical role analogous to $\lambda$ in the scalar case.

For a diagonalizable Laplacian matrix $\mathbf{L}$, the proof essentially reverts to the scalar case as outlined above (refer to (56) in our paper). However, in scenarios where $\mathbf{L}$ is non-diagonalizable and has a general Jordan normal form, it becomes necessary to employ the Laplace transform technique to demonstrate that the algebraic rate remains valid (refer to the context between (56) and (58)).

*Proof.* We first prove the stationary probability $\boldsymbol{\pi} = \left( \frac{d_1}{\sum_{j=1}^N d_j}, \ldots, \frac{d_N}{\sum_{j=1}^N d_j} \right)$ by induction. Assume that for $i = 1, \ldots, n$, the probability distribution $\mathbb{P}(\mathbf{R}(t_n))$ always equals $\boldsymbol{\pi}^{\mathsf{T}}$. For $i = n + 1$, from (46), it follows that

$$
\begin{aligned}
[\mathbb{P}(\mathbf{R}(t_{n+1}))]_h &= \mathbb{P}(\mathbf{R}(t_{n+1}) = \mathbf{x}_h) \\
&= b_n \mathbb{P}(\mathbf{R}(t_0) = \mathbf{x}_i) + \sum_k c_k \mathbb{P}(\mathbf{R}(t_{n+1-k}) = \mathbf{x}_h) \\
&\quad - \sigma^\beta \mathbb{P}(\mathbf{R}(t_n) = \mathbf{x}_h) + \sum_{j=1}^N \sigma^\beta \frac{W_{jh}}{d_j} \mathbb{P}(\mathbf{R}(t_n) = \mathbf{x}_j) \\
&= \boldsymbol{\pi}_h b_n + \sum_{k=1}^n \boldsymbol{\pi}_h c_k - \boldsymbol{\pi}_h \sigma^\beta + \sum_{j=1}^N \boldsymbol{\pi}_j \sigma^\beta \frac{W_{jh}}{d_j} \\
&= \boldsymbol{\pi}_h (b_n + \sum_{k=1}^n c_k) - \boldsymbol{\pi}_h \sigma^\beta + \sum_{j=1}^N \frac{d_j}{\sum_{j=1}^N d_j} \sigma^\beta \frac{W_{jh}}{d_j} \\
&= \boldsymbol{\pi}_h - \boldsymbol{\pi}_h \sigma^\beta + \sigma^\beta \sum_{j=1}^N \frac{W_{jh}}{\sum_{j=1}^N d_j} \\
&= \boldsymbol{\pi}_h - \boldsymbol{\pi}_h \sigma^\beta + \sigma^\beta \frac{d_h}{\sum_{j=1}^N d_j} \\
&= \boldsymbol{\pi}_h.
\end{aligned}
$$

This proves the existence of stationary probability. The uniqueness follows from this observation: if $\mathbb{P}(\mathbf{R}(t_1)) = \boldsymbol{\pi}' \neq \boldsymbol{\pi}$, we do not have $\mathbb{P}(\mathbf{R}(t_2)) = \mathbb{P}(\mathbf{R}(t_1))$ since otherwise it indicates that the Markov chain defined by

$$
\mathbb{P}\big(\mathbf{R}(t_{n+1}) = \mathbf{x}_{j_{n+1}} \,\big|\, \mathbf{R}(t_0) = \ldots, \mathbf{R}(t_1) = \ldots, \ldots, \mathbf{R}(t_n) = \mathbf{x}_{j_n}\big) \tag{51}
$$

$$
= \mathbb{P}(\mathbf{R}(t_{n+1}) = \mathbf{x}_j \,|\, \mathbf{R}(t_n) = \mathbf{x}_i) \tag{52}
$$

$$
= \mathbb{P}(\mathbf{R}(t_2) = \mathbf{x}_j \,|\, \mathbf{R}(t_1) = \mathbf{x}_i) \tag{53}
$$

$$
= \begin{cases} c_1 - \sigma^\beta + b_1 & \text{if staying at current location with } j = i \\ \sigma^\beta \frac{W_{ij}}{d_i} & \text{if jumping to neighboring nodes with } j \neq j \end{cases} \tag{54}
$$

has stationary distribution other than $\boldsymbol{\pi}$, which contradicts the assumption of a strongly connected and aperiodic graph.

We next establish the algebraic convergence as $0 < \beta < 1$.

It is evident that for the matrix $\mathbf{W}\mathbf{D}^{-1}$, given that it is column stochastic and the graph is strongly connected and aperiodic, the Perron-Frobenius theorem Horn & Johnson (2012)[Lemma 8.4.3., Theorem 8.4.4] confirms that the value 1 is the unique eigenvalue of this matrix that equals its spectral radius, which is also 1. Consequently, it follows that the matrix $\mathbf{L} = \mathbf{I} - \mathbf{W}\mathbf{D}^{-1}$ has an eigenvalue of 0, with all other eigenvalues possessing positive real parts. Considering the Jordan canonical form of $\mathbf{L}$, denoted as $\mathbf{L} = \mathbf{S}\mathbf{J}\mathbf{S}^{-1}$, it is observed that $\mathbf{J}$ contains a block that consists solely of a single 0, while the other blocks are characterized by eigenvalues $\lambda_k$ possessing positive real parts.

WLOG, we assume that the dimension of $\mathbf{X} \in \mathbb{R}^N$ in (9), as this is consistent with handling the probability vector $\mathbf{P}(t)$ described in (12). We rewrite it as

$$
D_t^\beta \mathbf{Y}(t) = -\mathbf{J}\mathbf{Y}(t) \tag{55}
$$

where $\mathbf{S}^{-1}\mathbf{X}(t) = \mathbf{Y}(t) \in \mathbb{R}^N$ representing a transformation of the feature space, and the transformed initial condition is defined as $\mathbf{S}^{-1}\mathbf{X}(0) = \mathbf{Y}(0)$.

If the matrix $\mathbf{L}$ is diagonalizable, then its Jordan canonical form $\mathbf{J}$ becomes a diagonal matrix, with the diagonal elements representing the eigenvalues of $\mathbf{L}$. In this scenario, the differential equation can be decoupled into a set of independent equations, each described by

$$
D_t^\beta \mathbf{Y}_k(t) = -\lambda_k \mathbf{Y}_k(t). \tag{56}
$$

Here, $\mathbf{Y}_k$ signifies the $k$-th component of the vector $\mathbf{Y}$. According to Diethelm (2010)[Theorem 4.3.], the solution to each differential equation in the given context is represented as:

$$\mathbf{Y}_k(t) = \mathbf{Y}_k(0) E_\beta(-\lambda_k t^\beta) \tag{57}$$

where is $E_\beta(\cdot)$ is the Mittag-Leffler function define as $E_\beta(z) = \sum_{j=0}^{\infty} \frac{z^j}{\Gamma(\beta j + 1)}$ and $\Gamma(\cdot)$ is the gamma function. This formulation leads to two important observations:

1. For the index $j$ such that the eigenvalue $\lambda_j = 0$, the solution simplifies to $\mathbf{Y}_j(t) = \mathbf{Y}_j(0)$. This corresponds to a stationary vector in the original space when transformed back to $\mathbf{X}(t)$.
2. According to Podlubny (1999)[Theorem 1.4.], for indices $k \neq j$, since $\lambda_k$ has a positive real part, the convergence to zero is characterized by the following order:

$$\mathbf{Y}_k(t) = \Theta(t^{-\beta}).$$

Asymptotically, this indicates that all components $\mathbf{Y}_k(t)$, except $\mathbf{Y}_j(t)$, will converge to zero at an algebraic rate. In terms of $\mathbf{X}(t)$, this translates into a convergence towards a stationary vector in the eigenspace corresponding to the eigenvalue 0, while components associated with other eigenspaces diminish at an algebraic rate.

If the matrix $\mathbf{J}$ is not diagonal, the entries of $\mathbf{Y}(t)$ corresponding to distinct Jordan blocks in $\mathbf{J}$ remain uncoupled. Therefore, it suffices to consider a single Jordan block corresponding to a non-zero eigenvalue $\lambda_k$. In this case, employing the Laplace transform technique becomes useful for demonstrating that the algebraic rate of convergence remains valid. We assume the Jordan block $\mathbf{J}(\lambda_k)$, associated with $\lambda_k$, is of size $m$. It follows that for this Jordan block we have

$$D_t^\beta \mathbf{Y}_1(t) = -\lambda_k \mathbf{Y}_1(t) - \mathbf{Y}_2(t),$$

$$\vdots \quad \vdots$$

$$D_t^\beta \mathbf{Y}_{m-1}(t) = -\lambda_k \mathbf{Y}_{m-1}(t) - \mathbf{Y}_m(t),$$

$$D_t^\beta \mathbf{Y}_m(t) = -\lambda_k \mathbf{Y}_m(t),$$

which can be solved from the bottom up. Beginning with the last equation, we obtain:

$$\mathbf{Y}_m(t) = \mathbf{Y}_m(0) E_\beta(-\lambda_k t^\beta) = \Theta(t^{-\beta}).$$

Further, the differential equation for $\mathbf{Y}_{m-1}(t)$ is given by:

$$D_t^\beta \mathbf{Y}_{m-1}(t) = -\lambda_k \mathbf{Y}_{m-1}(t) - \mathbf{Y}_m(0) E_\beta(-\lambda_k t^\beta)$$

Applying the Laplace transform and referring to (3), we obtain:

$$\mathcal{L}\left\{ D_t^\beta \mathbf{Y}_{m-1}(t) \right\} = s^\beta Y_{m-1}(s) - s^{\beta-1} \mathbf{Y}_{m-1}(0)$$

where $Y_{m-1}(s)$ is the Laplace transform of $\mathbf{Y}_{m-1}(t)$. For the right-hand side of the differential equation, we have $\mathcal{L}\{\lambda_k \mathbf{Y}_{m-1}(t)\} = \lambda_k Y_{m-1}(s)$. Additionally, the Laplace transform of the Mittag-Leffler function $E_\beta\left(-\lambda_k t^\beta\right)$ known to be $\frac{s^{\beta-1}}{s^\beta + \lambda_k}$ Podlubny (1999)[eq 1.80]. Consequently, the equation in the Laplace domain is represented as:

$$s^\beta Y_{m-1}(s) - s^{\beta-1} \mathbf{Y}_{m-1}(0) = -\lambda_k Y_{m-1}(s) - \mathbf{Y}_m(0) \frac{s^{\beta-1}}{s^\beta + \lambda_k}$$

Rearranging this equation to isolate $Y_{m-1}(s)$ yields:

$$Y_{m-1}(s) = \frac{s^{\beta-1} \mathbf{Y}_{m-1}(0) - \mathbf{Y}_m(0) \frac{s^{\beta-1}}{s^\beta + \lambda_k}}{s^\beta + \lambda_k}$$

As $s \to 0$, it follows that $Y_{m-1}(s) = \Theta(s^{\beta-1})$. Applying the same process recursively, we find that $Y_i(s) = \Theta(s^{\beta-1})$ for all $i = 1, \ldots, m$. Invoking the Hardy–Littlewood Tauberian theorem Wikipedia (2023), we can conclude that for all indices $i = 1, \ldots, m$, the following relationship holds:

$$\mathbf{Y}_i(t) = \Theta(t^{-\beta}). \tag{58}$$

Consequently, we can deduce that, akin to the scenarios involving diagonalizable matrices, the feature components associated with other eigenspaces in non-diagonalizable cases also diminish at an algebraic rate.

The proof now is complete. $\qquad\square$

LIMITATIONS

Our research proposes an advanced graph diffusion framework that integrates *time-fractional deriva-tives*, effectively encompassing many GNNs. Nonetheless, it presents certain limitations. A crucial element we have overlooked is the application of the *fractional derivative in the spatial domain*. In fractional diffusion equations, this implies substituting the standard second-order spatial derivative with a Riesz-Feller derivative (Gorenflo & Mainardi, 2003), thus modeling a random walk with space-based long-range jumps. Incorporating such a space-fractional diffusion equation within GNNs could potentially alleviate issues like the bottleneck and over-squashing highlighted in (Alon & Yahav, 2021). This represents a current limitation of our work and suggests a compelling future research trajectory that merges both time and space fractional derivatives in GNNs.

BROADER IMPACT

The introduction of FROND holds significant potential for applications such as sensor networks, transportation, and manufacturing. FROND's ability to encapsulate long-term memory in neural dy-namical processes can enhance the representation of complex interconnections, improving predictive modeling and efficiency. This could lead to more responsive sensor networks, optimized routing in transportation, and improved visibility into manufacturing process networks. However, the advent of FROND and similar models may also have mixed labor implications. While these technologies might render certain repetitive tasks obsolete, potentially displacing jobs, they may also generate new opportunities focused on developing and maintaining such advanced systems. Moreover, the shift from mundane tasks could enable workers to focus more on strategic and creative roles, enhancing job satisfaction and productivity. It's paramount that the deployment of FROND is done ethically, with ample support for reskilling those whose roles may be affected. This helps ensure that the broader impact of this technology is beneficial to society as a whole.

