# Unleashing the Potential of Fractional Calculus in Graph Neural Networks with FROND

## Abstract

We introduce the FRactional-Order graph Neural Dynamical network (FROND), a learning framework that extends traditional graph neural ordinary differential equation (ODE) models by incorporating the time-fractional Caputo derivative. Due to its non-local nature, fractional calculus allows our framework to capture long-term memories in the feature updating process, in contrast to the Markovian nature of updates in traditional graph neural ODE models. This can lead to improved graph representation learning. We offer an interpretation of the feature updating process on graphs from a non-Markovian random walk perspective when the feature updating is governed by a diffusion process. We demonstrate analytically that over-smoothing can be mitigated in this setting. To experimentally demonstrate the versatility of the FROND framework, we evaluate the fractional counterparts of various established graph ODE models. Their consistently superior performance, compared to their original counterparts, highlights the potential of the FROND framework as an effective extension to boost the efficacy of various graph neural ODE models.

## 1 Introduction

Graph Neural Networks (GNNs) have excelled in diverse domains, e.g., chemistry (Yue et al., 2019), finance (Ashoor et al., 2020), and social media (Kipf & Welling, 2017; Zhang et al., 2022; Wu et al., 2021). The message passing scheme (Feng et al., 2022), where features are aggregated along edges and iteratively propagated through layers, is crucial for the success of GNNs. Over the past few years, numerous types of GNNs have been proposed, including Graph Convolutional Networks (GCN) (Kipf & Welling, 2017), Graph Attention Networks (GAT) (Veličković et al., 2018), and GraphSAGE (Hamilton et al., 2017). Recent works, such as (Chamberlain et al., 2021c; Thorpe et al., 2022; Rusch et al., 2022; Song et al., 2022; Choi et al., 2023; Zhao et al., 2023), have incorporated various continuous dynamical processes to propagate information over the graph nodes, inspiring a new class of GNNs based on ordinary differential equations (ODEs)[1] on graphs which enables the interpretation of GNNs as evolutionary dynamical systems. These models have demonstrated notable performance, for instance, in enhancing robustness and addressing heterophilic graphs.

Within these graph neural ODE models, the differential operator $\mathrm{d}^\beta / \mathrm{d}t^\beta$ is conventionally constrained to *integer values* of $\beta$, primarily 1 or 2. However, over recent decades, the wider scientific community has delved into the domains of fractional-order differential operators, where $\beta$ can be any *real number*. These expansions have proven pivotal in various applications characterized by nonlocal and memory-dependent behaviors, with prime examples including viscoelastic materials (Bagley & Torvik, 1983), anomalous transport mechanisms (Gómez-Aguilar et al., 2016), and fractal media (Mandelbrot & Mandelbrot, 1982). The distinction lies in the fact that the conventional integer-order derivative measures the function's *instantaneous change rate*, concentrating on the proximate vicinity of the point. *In contrast, the fractional-order derivative (Tarasov, 2011) is influenced by the entire historical trajectory of the function,* which substantially diverges from the localized impact found in integer-order derivatives. For detailed definitions of fractional-order derivatives, readers are referred to Section 2.1 and Appendix B. We introduce the FRactional-Order graph Neural Dynamical network (FROND) framework, a new approach that broadens the capabilities of traditional graph neural

---

[1]Models like GRAND (Chamberlain et al., 2021c) primarily utilize ODEs on graphs, albeit inspired by partial differential equations. We consistently refer to such models as graph neural ODE models.

ODE models by incorporating fractional calculus. It naturally generalizes the integer-order derivative $\mathrm{d}^\beta / \mathrm{d}t^\beta$ in graph neural ODE models to accommodate any positive real number $\beta$. This modification gives FROND the ability to incorporate *memory-dependent dynamics* for information propagation and feature updating, enabling refined graph representations and improved performance potentially. Importantly, this technique assures at least equivalent performance to integer-order models, as, when $\beta$ assumes integer values, the models revert to conventional graph ODE models without memory.

Several works like (Maskey et al., 2023) have incorporated fractional graph shift operators within graph neural ODE models. These studies are distinct from our research, wherein we focus on incorporating time-fractional derivatives for updating graph node features, modeled as a memory-inclusive dynamical process. Other works like (Liu et al., 2022) have used fractional calculus in gradient propagation for the training process, which is different from leveraging fractional differential equations (FDEs) in modeling the node feature updating. We provide a detailed discussion of the differences between FROND and these other works in Appendix A.

It is worth noting that the further enhancement garnered from employing fractional calculus can be contingent on the graph dataset's topology and features. Our proposed feature updating mechanism, leveraging fractional derivatives, demonstrates proficiency in processing datasets with prominent tree-like structures. Hyperbolic GNNs (Chami et al., 2019; Liu et al., 2019) have proposed to embed graph nodes in hyperbolic spaces instead of the familiar Euclidean spaces. This is based on a pivotal work in network science (Krioukov et al., 2010), which established that hyperbolic geometry is aptly designed to encapsulate complex networks, especially those manifesting scale-free hierarchical structures reminiscent of trees. By scale-free, we refer to the characteristic where the node degree distribution adheres to a power law: $P(k) \propto k^{-\alpha}$. This exponent $\alpha$ can be viewed as a reflection of the negative curvature inherent to the underlying hyperbolic geometry (Krioukov et al., 2010).

*Our work leads us down a slightly different but related geometric path: that of fractal geometry*. The scale-free attribute hints at a pervasive self-similarity across varied scales, indicative of inherent fractal behavior (Kim et al., 2007; Masters, 2004). Here, "scale" refers to the clustering of interconnected nodes at various granularities, reminiscent of hierarchical tree branching. This phenomenon means that the power law distribution, even post scaling, continues to adhere to the identical distribution law, i.e., $P(ck) \propto k^{-\alpha}$. The degree distribution's exponent $\alpha$ also naturally acts as a reflection of the fractal dimension of the underlying fractal geometry (Song et al., 2005). Dynamical processes with self-similarity on such fractal media are well known to be better described using FDEs. For example, when heat or mass disperses over such structures, its concentration is best described using fractional diffusion equations (Diaz-Diaz & Estrada, 2022). The non-integer order derivatives elegantly encapsulate the fractal characteristics of the media. Further exploration reveals that the fractal dimension is intrinsically linked to the order of fractional derivatives (Nigmatullin, 1992; Tarasov, 2011). In other words, the exponent $\alpha$ has a profound connection to the parameter $\beta$ in $\mathrm{d}^\beta / \mathrm{d}t^\beta$. This revelation births a compelling insight: the optimal $\beta$ in our models, which may differ from integers, can pave the way for enhanced node classification and potentially unearth insights into the inherent "fractal" nature of the graph datasets.

**Main contributions.** Our objective in this paper is to formulate a generalized fractional-order graph learning framework that can serve as a reliable plugin for various graph ODE models. Our key contributions are summarized as follows:

- We propose a novel, generalized graph framework that incorporates time-fractional derivatives. This framework generalizes prior graph neural ODE models (Chamberlain et al., 2021c; Thorpe et al., 2022; Rusch et al., 2022; Song et al., 2022; Choi et al., 2023; Zhao et al., 2023), subsuming them as special instances. Specifically, when the fractional order $\beta$ equals 1, the non-local fractional derivative operator $\mathrm{d}^\beta / \mathrm{d}t^\beta$ reverts to the conventional local first-order derivative $\mathrm{d}/ \mathrm{d}t$ utilized in graph neural ODE models. This approach also lays the groundwork for a diverse new class of GNNs that can accommodate a broad array of learnable feature-updating processes with memory.
- We provide an interpretation from the perspective of a non-Markovian graph random walk when the model feature-updating dynamics is inspired by the fractional heat diffusion process (cf. F-GRAND-L in (9)). Contrasting with the traditional Markovian random walk implicit in traditional graph neural diffusion models whose convergence to the stationary equilibrium is exponentially swift, we establish that in FROND, convergence follows an algebraic rate. This characteristic enhances FROND's ability to mitigate over-smoothing, as verified by our experimental results.
- We underscore the compatibility of FROND, emphasizing its capability to be seamlessly integrated to augment the performance of existing graph ODE models across diverse datasets. Our exhaustive

experiments, encompassing the fractional differential extension of (Chamberlain et al., 2021c; Thorpe et al., 2022; Rusch et al., 2022; Song et al., 2022; Choi et al., 2023; Zhao et al., 2023), substantiate this claim. Through detailed ablation studies, we provide insights into the choice of numerical schemes and parameters.

## 2 PRELIMINARIES

This work proposes a novel GNN framework based on fractional calculus. We succinctly outline fractional calculus principles and prevalent graph neural ODE models. In Section 3, we augment these models with fractional differential extensions, introducing a new GNN class featuring memory-inclusive feature updating dynamics. For an extensive overview of fractional calculus, readers are directed to Appendix B.

### 2.1 THE CAPUTO TIME-FRACTIONAL DERIVATIVE

The literature offers various fractional derivative definitions, notably by Riemann, Liouville, Chapman, and Caputo (Tarasov, 2011). Our study mainly leverages the *Caputo* fractional derivative, due to the reasons listed in Appendix B.4. The traditional first-order derivative of a scalar function $f(t)$ represents the local rate of change of the function at a point, defined as: $\frac{\mathrm{d}f(t)}{\mathrm{d}t} = \lim_{\Delta t \to 0} \frac{f(t+\Delta t)-f(t)}{\Delta t}$. Let $F(s)$ denote the Laplace transform of $f(t)$, assumed to exist on $[s_0, \infty)$ for some $s_0 \in \mathbb{R}$. Under certain conditions (Korn & Korn, 2000), the Laplace transform of $\frac{\mathrm{d}f(t)}{\mathrm{d}t}$ is given by:

$$\mathcal{L}\left\{\frac{\mathrm{d}f(t)}{\mathrm{d}t}\right\} = sF(s) - f(0) \tag{1}$$

The Caputo fractional derivative of order $\beta \in (0, 1]$ for a function $f(t)$ is defined as follows:

$$D_t^\beta f(t) = \frac{1}{\Gamma(1-\beta)} \int_0^t (t-\tau)^{-\beta} f'(\tau)\,\mathrm{d}\tau, \tag{2}$$

where $\Gamma(\cdot)$ denotes the gamma function, and $f'(\tau)$ is the first-order derivative of $f$. The broader definition for $\beta > 0$ is deferred to Appendix B. In the primary models of this paper, we focus on cases where $\beta \in (0, 1]$. The Caputo fractional derivative inherently integrates the entire history of the system through the integral term, emphasizing its non-local nature. For $s > \max\{0, s_0\}$, the Laplace transform of the Caputo fractional derivative is given by (Diethelm, 2010)[Theorem 7.1]:

$$\mathcal{L}\left\{D_t^\beta f(t)\right\} = s^\beta F(s) - s^{\beta-1} f(0). \tag{3}$$

Comparing the Laplace transforms of the traditional and Caputo fractional derivatives, as depicted in (1) and (3), it is evident that the Caputo derivative serves as a generalization of the traditional one. The alteration in the exponent of $s$ introduces memory-dependent properties, as observed in (2), enabling the development of enhanced GNN models. As $\beta \to 1$, the Laplace transform of the Caputo fractional derivative converges to that of the traditional first-order derivative. Thus, when $\beta = 1$, $D_t^1 f = f'$ is uniquely determined through the inverse Laplace transform (Cohen, 2007). In summary, the Caputo fractional derivative and its Laplace transform can be seen as a natural extension of the traditional first-order derivative from the frequency domain using the Laplace transform. For a vector-valued function, the Caputo fractional derivative is defined component-wise for each dimension, similar to the first-order derivative.

### 2.2 GRAPH NEURAL ODE MODELS

We denote an undirected graph as $G = (\mathbf{X}, \mathbf{W})$, where $\mathbf{X} = \left(\left[\mathbf{x}^{(1)}\right]^\mathsf{T}, \cdots, \left[\mathbf{x}^{(N)}\right]^\mathsf{T}\right)^\mathsf{T} \in \mathbb{R}^{N \times d}$ consists of rows $\mathbf{x}^{(i)} \in \mathbb{R}^d$ as node feature vectors and $i$ is the node index. The $N \times N$ matrix $\mathbf{W} := (W_{ij})$ has elements $W_{ij}$ indicating the edge weight between the $i$-th and $j$-th feature vectors with $W_{ij} = W_{ji}$. The subsequent feature updating process leverages ODEs to facilitate information propagation amongst graph nodes, modifying the node features $\mathbf{X}$. We present prevalent graph neural ODE models as follows.

**GRAND:** Inspired by the heat diffusion equation, GRAND (Chamberlain et al., 2021c) utilizes the following nonlinear autonomous dynamical system:

$$\frac{\mathrm{d}\mathbf{X}(t)}{\mathrm{d}t} = (\mathbf{A}(\mathbf{X}(t)) - \mathbf{I})\mathbf{X}(t). \tag{4}$$

where $\mathbf{A}(\mathbf{X}(t))$ is a learnable, time-variant attention matrix, calculated using the features $\mathbf{X}(t)$, and $\mathbf{I}$ denotes the identity matrix. The feature update outlined in (4) is referred to as the **GRAND-nl** version (due to the nonlinearity in $\mathbf{A}(\mathbf{X}(t))$). We define $d_i = \sum_{j=1}^n W_{ij}$ and let $\mathbf{D}$ be a diagonal

matrix with $D_{ii} = d_i$. The *random walk Laplacian* is then represented as $\mathbf{L} = \mathbf{I} - \mathbf{W}\mathbf{D}^{-1}$. In a simplified context, we employ the following linear dynamical system:

$$\frac{d\mathbf{X}(t)}{dt} = (\mathbf{W}\mathbf{D}^{-1} - \mathbf{I})\mathbf{X}(t) = -\mathbf{L}\mathbf{X}(t). \qquad (5)$$

The feature update process in (5) is the **GRAND-l** version. For implementations of (5), one may set $\mathbf{W} = \mathbf{A}(\mathbf{X}(0))$, rather than using a plain weight. Notably, in this time-invariant setting, the attention weight matrix, reliant on the initial node features, stays unchanged throughout the feature evolution period, and $\mathbf{D} = \mathbf{I}$ if the attention matrix is chosen to be row-stochastic.

**GRAND++:** The work by (Thorpe et al., 2022) introduces graph neural diffusion with a source term, aimed at graph learning in scenarios with a limited number of labeled nodes.

**GraphCON:** Inspired by oscillator dynamical systems, GraphCON (Rusch et al., 2022) is defined through the employment of second-order ODEs. It is crucial to highlight that, the second-order ODE is equivalent to two first-order ODEs.

**CDE:** To navigate the challenges presented by heterophilic graphs, Zhao et al. (2023) incorporates convection-diffusion equations (CDE) into GNNs, leading to the proposal of the neural CDE model.

**GREAD:** To address the challenges posed by heterophilic graphs, Choi et al. (2023) presents the GREAD model. This model enhances the GRAND model by incorporating a reaction term, thereby formulating a diffusion-reaction equation within GNNs.

We do not present the detailed formulations for each graph ODE model but refer the interested reader to their respective primary papers and Appendix E.1. Broadly, the models diverge in their approaches to feature updating dynamics, and transformations on $\mathbf{X}$ may be performed preceding the ODE module.

## 3 FRACTIONAL-ORDER GRAPH NEURAL DYNAMICAL NETWORK

In this section, we introduce the FROND framework, a novel approach that augments traditional graph neural ODE models by incorporating fractional calculus. We elucidate the fractional counterparts of several well-established graph ODE models, including GRAND, GRAND++, GraphCON, CDE, and GREAD, as referenced in Section 2.2. We provide a detailed study of the fractional extension of GRAND, and present insights into the inherent memory mechanisms of fractional calculus through a random walk interpretation. Our theoretical findings suggest a potential mitigation of over-smoothness due to the model's algebraic convergence to stationarity. Subsequently, we outline techniques for the numerical FDE solver pertinent to FROND.

### 3.1 FRAMEWORK

Consider a graph $\mathcal{G} = (\mathcal{V}, \mathbf{W})$ composed of $|\mathcal{V}| = N$ nodes and $\mathbf{W}$ the set of edge weights as defined in Section 2.2. Analogous to the implementation in traditional graph neural ODE models, a preliminary learnable encoder function $\varphi : \mathcal{V} \to \mathbb{R}^d$ that maps each node to a feature vector can be applied. Stacking all the feature vectors together, we obtain $\mathbf{X} \in \mathbb{R}^{N \times d}$. Employing the Caputo time fractional derivative outlined in Section 2.1, the information propagation and feature updating dynamics in FROND are characterized by the following graph neural FDE:

$$D_t^\beta \mathbf{X}(t) = \mathcal{F}(\mathbf{W}, \mathbf{X}(t)), \quad \beta > 0, \qquad (6)$$

where $\beta$ denotes the fractional order of the derivative, and $\mathcal{F}$ is a dynamic operator on the graph like the models presented in Section 2.2. The initial condition for (6) is set as $\mathbf{X}^{(\lceil\beta\rceil-1)}(0) = \ldots = \mathbf{X}(0) = \mathbf{X}$ consisting of the preliminary node features[2], where $\lceil\beta\rceil$ denotes the smallest integer greater than or equal to $\beta$, akin to the initial conditions seen in ODEs. In alignment with the graph neural ODE models (Chamberlain et al., 2021c; Thorpe et al., 2022; Rusch et al., 2022; Song et al., 2022; Choi et al., 2023; Zhao et al., 2023), we set an integration time parameter $T$ to yield $\mathbf{X}(T)$. The final node embedding for subsequent downstream tasks may be decoded as $\psi(\mathbf{X}(T))$ with $\psi$ being a learnable decoder function.

The GNN framework in (6) incorporates the fractional feature updating process, forming a novel message-passing mechanism for GNNs. When $\beta = 1$, (6) reverts to the traditional graph neural diffusion elaborated in Section 2.2, with the infinitesimal variation of features dependent on their present state. Conversely, when $\beta < 1$, the Caputo fractional derivative definition (2) illustrates that it is the entire history of the feature updating process that is implicated, not merely the features'

---

[2]In the main paper, we mainly consider $\beta \in (0, 1]$ and the initial condition is $\mathbf{X}(0) = \mathbf{X}$. See Appendix B.3.2.

instantaneous change rate. This insight induces *memory-dependent dynamics* for information propagation and feature updating. For further insights into memory dependence, readers are directed to Section 3.3, where time discretization enables numerical resolution of the system, showing how time persistently serves as an analog to the layer index in ODE models and how the non-local nature of fractional derivatives introduces nontrivial dense or skip connections between layers. In Section 3.2, when the dynamic operator $\mathcal{F}$ is designated as the diffusion process in (5), we offer a broader memory-dependent *non-Markov* random walk interpretation of the fractional graph diffusion process. Here, as $\beta \to 1$, the non-Markov random walk increasingly detaches from the path history, becoming a Markov walk at $\beta = 1$, which is interpretable as the traditional diffusion process as shown in (Thorpe et al., 2022). The parameter $\beta$ provides flexibility to adjust the extent of memorized dynamics embedded in the framework. As clarified in Section 1, our methodology also adopts a *fractal geometric* interpretation. Within this perspective, the dynamics pertaining to information propagation can be more effectively represented using FDEs, particularly in fractal networks. The FROND framework may elegantly encapsulate the fractal attributes in graph datasets.

### 3.1.1 FRACTIONAL MODEL EXAMPLES

When the operator $\mathcal{F}$ in (5) is specified to the dynamics depicted in various notable graph neural ODE models, as illustrated in Section 2.2, we formulate fractional GNN variants such as F-GRAND, F-GRAND++, F-GREAD, F-CDE, and F-GraphCON. These serve as fractional counterparts to the graph ODE models.

**F-GRAND**: Mirroring the GRAND model, the fractional GRAND (F-GRAND) is divided into two versions. The F-GRAND-nl employs a time-variant FDE as follows:

$$D_t^\beta \mathbf{X}(t) = (\mathbf{A}(\mathbf{X}(t)) - \mathbf{I})\mathbf{X}(t), \quad 0 < \beta \le 1. \tag{7}$$

It is computed using $\mathbf{X}(t)$ and the attention mechanism derived from the Transformer model (Vaswani et al., 2017). The entries of $\mathbf{A}(\mathbf{X}(t))$ are given by:

$$a(\mathbf{x}_i, \mathbf{x}_j) = \mathrm{softmax}\left(\frac{(\mathbf{W}_K \mathbf{x}_i)^\top \mathbf{W}_Q \mathbf{x}_j}{d_k}\right). \tag{8}$$

In this formulation, $\mathbf{W}_K$ and $\mathbf{W}_Q$ are the learned matrices, and $d_k$ signifies a hyperparameter defining the dimensionality of $W_K$. In parallel, the F-GRAND-l version stands as the fractional equivalent of (5):

$$D_t^\beta \mathbf{X}(t) = -\mathbf{L}\mathbf{X}(t), \quad 0 < \beta \le 1. \tag{9}$$

**F-GRAND++, F-GREAD, F-CDE, and F-GraphCON:** Due to space constraints, we direct the reader to Appendix E for detailed formulations. Succinctly, they represent the fractional extensions of GRAND++, GraphCON, CDE, and GREAD. To highlight FROND's compatibility and its potential to enhance the performance of existing graph ODE models across a variety of datasets, exhaustive experiments are provided in Section 4 and Appendix E.

### 3.2 RANDOM WALK PERSPECTIVE OF F-GRAND-L

The established Markov interpretation of GRAND-l (5), as outlined in (Thorpe et al., 2022), aligns with F-GRAND-l (9) when $\beta = 1$. We herein broaden this interpretation to encompass non-Markov random walks when $\beta$ is a non-integer, thereby elucidating the memory effects inherent in FDEs through a consideration of the walker's path history. In contrast to the Markovian walk, which converges exponentially to equilibrium, our strategy assures algebraic convergence, enhancing F-GRAND-l's efficacy in mitigating over-smoothing as evidenced in Section 4.3.

To begin, we discretize the time domain into time instants as $t_n = n\sigma, \sigma > 0, n = 0, 1, 2, \ldots$, where $\sigma$ is assumed to be small enough to ensure the validity of the approximation. Let $\mathbf{R}(t_n)$ be a random walk on the graph nodes $\{\mathbf{x}^{(j)}\}_{j=0}^N$ that is, in general, not a Markov process and $\mathbf{R}(t_{n+1})$ depends on the path history $(\mathbf{R}(t_0), \mathbf{R}(t_1), \ldots, \mathbf{R}(t_n))$ of the random walker. For convenience, we introduce the coefficients $c_k$ for $k \ge 1$ and $b_m$ for $m \ge 0$ from (Gorenflo et al., 2002), which are used later to define the random walk transition probability:

$$c_k(\beta) = (-1)^{k+1}\binom{\beta}{k} = \left|\binom{\beta}{k}\right|, \quad b_m(\beta) = \sum_{k=0}^m (-1)^k \binom{\beta}{k}, \tag{10}$$

where the generalized binomial coefficient $\binom{\beta}{k} = \frac{\Gamma(\beta+1)}{\Gamma(k+1)\Gamma(\beta-k+1)}$ and the gamma function $\Gamma$ are employed in the definition of the coefficients. The sequences $c_k$ and $b_m$ consist of positive numbers, not greater than 1, decreasing strictly monotonically to zero (see supplementary material for details) and satisfy $\sum_{k=1}^n c_k + b_n = 1$. Using these coefficients, we define the transition probabilities of the

random walk starting from $\mathbf{x}^{(j_0)}$ as

$$\mathbb{P}\Big(\mathbf{R}(t_{n+1}) = \mathbf{x}^{(j_{n+1})} \,\Big|\, \mathbf{R}(t_0) = \mathbf{x}^{(j_0)}, \mathbf{R}(t_1) = \mathbf{x}^{(j_1)}, \ldots, \mathbf{R}(t_n) = \mathbf{x}^{(j_n)}\Big)$$

$$= \begin{cases} c_1 - \sigma^\beta & \text{if staying at current location with } j_{n+1} = j_n, \\ \sigma^\beta \frac{W_{j_n j_{n+1}}}{d_{j_n}} & \text{if jumping to neighboring nodes with } j_{n+1} \neq j_n, \\ c_{n+1-k} & \text{if revisiting historical positions with } j_{n+1} = j_k, 1 \le k \le n-1, \\ b_n & \text{if revisiting historical positions with } j_{n+1} = j_0. \end{cases} \tag{11}$$

This formulation integrates memory effects, considering the walker's time, position, and path history. The transition mechanism of the memory-inclusive random walk between $t_n$ and $t_{n+1}$ is elucidated as follows: Suppose the walker is at node $j_n$ at time $t_n$, having a full path history $(j_0, j_1, \ldots, j_n)$. Generating a uniform random number $0 \le \rho < 1$, we divide the interval $[0, 1)$ into adjacent sub-intervals with lengths $c_1, c_2, \ldots, c_n, b_n$. We further subdivide the first interval (with length $c_1$) into sub-intervals of lengths $c_1 - \sigma^\beta$ and $\sigma^\beta$.

1. If $\rho$ is in the first interval with length $c_1$, the walker either moves to a neighbor $j_{n+1} = k$ with probability $\sigma^\beta \frac{W_{j_n k}}{d_{j_n}}$ or remains at the current position with probability $c_1 - \sigma^\beta$.

2. For $\rho$ in subsequent intervals, the walker jumps to a previously visited node in the history $(j_0, j_1, \ldots, j_{n-1})$, specifically, to $j_{n+1-k}$ if in $c_k$, or to $j_0$ if in $b_n$.

When $\beta < 1$, the random walk can, with positive probability, revisit its history, which prevents the walker from drifting too far away from its local region. Using the technique from (Gorenflo et al., 2002), we can prove the following:

**Theorem 1.** *When $\sigma \to 0$ and $n\sigma = t$, $\mathbb{E}_i \mathbf{R}(t_n)$ converges to $\mathbf{x}^{(i)}(t)$, the $i$-th component of the solution $\mathbf{X}(t)$ to (9). Here, $\mathbb{E}_i$ denotes the expectation over the random walk, defined by transition probabilities in (11), which begins at node $i$ with initial distribution $\mathbf{R}(0) = \mathbf{x}^{(i)}$, with probability 1.*

**Remark 1.** *Theorem 1 relates F-GAND-l (9) to the non-Markovian random walk in (11), illustrating memory dependence in FROND. As $\beta \to 1$, this process reverts to the Markov random walk found in GRAND-l (Thorpe et al., 2022) in (12). It underscores the FROND framework's capability to apprehend more complex dynamics than graph ODE models, potentially improving predictive performance.*

$$\mathbb{P}\Big(\mathbf{R}(t_{n+1}) = \mathbf{x}^{(j_{n+1})} \,\Big|\, \mathbf{R}(t_0) = \mathbf{x}^{(j_0)}, \mathbf{R}(t_1) = \mathbf{x}^{(j_1)}, \ldots, \mathbf{R}(t_n) = \mathbf{x}^{(j_n)}\Big) \tag{12}$$

$$= \mathbb{P}\Big(\mathbf{R}(t_{n+1}) = \mathbf{x}^{(j_{n+1})} \,\Big|\, \mathbf{R}(t_n) = \mathbf{x}^{(j_n)}\Big) = \begin{cases} 1 - \sigma & \text{if staying at current location with } j_{n+1} = j_n \\ \sigma \frac{W_{j_n j_{n+1}}}{d_{j_n}} & \text{if jumping to neighbors with } j_{n+1} \neq j_n \end{cases}$$

*since we have that all these coefficients vanishing except $c_1 = 1$, i.e.,*

$$c_1 = 1, \quad \lim_{\beta \to 1} c_k(\beta) = 0, \quad k \ge 2, \quad \lim_{\beta \to 1} b_m = 0, \quad m \ge 1. \tag{13}$$

*The approximation solution to (9) at $\beta = 1$ via the Markov random walk (12) is established in (Thorpe et al., 2022). Similarly, in the continuous domain, a solution to the heat equation can be represented by random Brownian motion from the positions (Durrett, 2019, Theorem 9.2.2).*

### 3.2.1 Over-smoothing Mitigation of F-GRAND-l Compared to GRAND-l

The stationary distribution for the Markov random walk, as given by (12), is recognized as $\boldsymbol{\pi} = (\frac{d_1}{\sum_{j=1}^N d_j}, \ldots, \frac{d_N}{\sum_{j=1}^N d_j})$. *The seminal research (Oono & Suzuki, 2020)[Corollary 3. and Remark 1] has incisively underscored that GNN over-smoothing is the exponential convergence to the stationary distribution when considering a GNN as a layered dynamical system.* More specifically, according to (Chung, 1997), we have the fast *exponential convergence* for GRAND as $\|\mathbb{P}(\mathbf{R}(t_n)) - \boldsymbol{\pi}^\mathsf{T}\|_2 \sim O(e^{-r'n})^3$, where $\mathbb{P}(\mathbf{R}(t_n))$ is the probability column vector, with its $j$-th element given as $\mathbb{P}\big(\mathbf{R}(t_n) = \mathbf{x}^{(j)}\big)$. Here, $r'$ is a positive value related to the eigenvalues of the matrix $\mathbf{L}$, and $\|\cdot\|_2$ denotes the $\ell^2$ norm. The continuous limit also shows analogous exponential convergence with $r > 0$:

$$\|\mathbb{P}(\mathbf{R}(t)) - \boldsymbol{\pi}^\mathsf{T}\|_2 \sim O(e^{-rt}). \tag{14}$$

In contrast, we next prove that the non-Markovian random walk with memory, as defined in (11), converges to the stationary distribution at a *slow algebraic rate*, thereby helping to mitigate over-smoothing. As $\beta \to 0$, the convergence is expected to be *arbitrarily slow*. In real-world scenarios

---

[3]We use the asymptotic order notations from (Notations, 2023) in this paper.

where we operate within a finite horizon, this slower rate of convergence may be sufficient to alleviate over-smoothing, particularly when it is imperative for a deep model to extract distinctive features instead of achieving exponentially fast convergence to a stationary distribution.

**Theorem 2.** *Under the assumption that the graph is strongly connected and aperiodic, the stationary probability for the non-Markov random walk* (11), *with* $0 < \beta < 1$, *is* $\boldsymbol{\pi} = (\frac{d_1}{\sum_{j=1}^N d_j}, \ldots, \frac{d_N}{\sum_{j=1}^N d_j})$, *which is unique. This mirrors the stationary probability of the Markovian random walk as defined by* (12) *when* $\beta = 1$. *Notably, when* $\beta < 1$, *the convergence of the distribution distinct from* $\boldsymbol{\pi}$ *to* $\boldsymbol{\pi}$ *is algebraic:*

$$\|\mathbb{P}(\mathbf{R}(t)) - \boldsymbol{\pi}^\mathsf{T}\|_2 \sim \Theta(t^{-\beta}). \tag{15}$$

**Remark 2.** *For clarity, Theorem 2 indicates that the feature* $\mathbf{x}^{(i)}(t)$, *for all node* $i$, *is converging to the same stationary feature equilibrium* $\mathbf{x_s} := \sum_k \frac{\mathbf{x}^{(k)} d_k}{\sum_{j=1}^N d_j}$ *at a slow algebraic rate. More specifically, we have:*

$$\left\|\mathbf{x}^{(i)}(t) - \mathbf{x_s}\right\|_2^2 = \|\sum_k \mathbf{x}^{(k)}[\mathbb{P}_i(\mathbf{R}(t))_k - \boldsymbol{\pi}_k]\|_2^2 = \Theta\left(t^{-2\beta}\right) \text{ for all node } i. \tag{16}$$

*where* $\mathbb{P}_i$ *refers to that we have the initial probability as a one-hot vector with the* $i$-*th component being 1. This is because Theorem 2 confirms* $\|\boldsymbol{\pi}_k - \mathbb{P}_i(\mathbf{R}(t))_k\| = \Theta\left(t^{-\beta}\right)$ *for some* $k$.

In (Rusch et al., 2022), the phenomenon of over-smoothness is defined through the exponential convergence of Dirichlet energy to zero. However, the following Corollary 1 establishes that the Dirichlet energy of F-GRAND-l converges algebraically to zero, mitigating over-smoothness issues as corroborated by the plots in Section 4.3 and Appendix D.7.

**Corollary 1.** *The Dirichlet energy,* $\mathbf{E}(\mathbf{X}(t))$, *with* $\mathbf{X}(t)$ *being the solution to* (9), *has the convergence rate* $\Theta(t^{-2\beta})$. *Here, Dirichlet energy* $\mathbf{E}(\mathbf{X}(t))$ *is formally defined as*

$$\mathbf{E}(\mathbf{X}(t)) := \sum_{i \in \mathcal{V}} \sum_{j \in \mathcal{V}} \left\|\mathbf{x}^{(i)}(t) - \mathbf{x}^{(j)}(t)\right\|_2^2 \tag{17}$$

### 3.3 SOLVING FROND

The studies by (Chen et al., 2018b; Quaglino et al., 2019; Yan et al., 2018) introduce numerical solvers specifically designed for neural ODE models when $\beta$ is an integer in the FROND framework. Our research, in contrast, engages with FDEs, entities inherently more intricate than ODEs. To address the scenario *where* $\beta$ *is non-integer*, we introduce the *fractional explicit Adams–Bashforth–Moulton method*, incorporating three variants employed in this study: the **basic predictor** discussed in Appendix C.1, the **predictor-corrector** elaborated in Appendix C.2, and the **short memory principle** detailed in Appendix C.3. Additionally, we present one **implicit L1** solver in Appendix C.4. These methods exemplify how time persistently acts as a continuous analog to the layer index and elucidate how resultant memory dependence manifests as nontrivial dense or skip connections between layers (see Figs. 2 and 3), stemming from the non-local properties of fractional derivatives.

## 4 EXPERIMENTS

We execute a series of experiments to illustrate that graph neural ODE models, structured under the FROND framework and utilizing $D_t^\beta$, achieve superior performance compared to the traditional models reliant on the $\frac{\mathrm{d}}{\mathrm{d}t}$ approach. **Importantly, our primary aim is not to achieve state-of-the-art results, but rather to demonstrate the additional effectiveness of the FROND framework when applied to existing graph neural ODE models.** In the main paper, we detail the impressive results achieved by F-GRAND, particularly emphasizing its efficacy on tree-structured data, and F-CDE, highlighting its proficiency in managing large heterophilic datasets. We also further validate the slow algebraic convergence, as discussed in Theorem 2, by constructing deeper GNNs with non-integer $\beta < 1$. To maintain consistency in the experiments presented in the main paper, the basic predictor solver is used instead of other solvers when $\beta < 1$.

**More Experiments In the Appendix:** The Appendix D section provides additional details covering various aspects such as experimental settings, described in Appendices D.1 to D.3, the performance of different solver variants in Appendix D.5, the computational complexity of F-GRAND in Appendix D.6, and analysis of F-GRAND's robustness against adversarial attacks in Appendix D.9. Furthermore, results related to other FROND-based graph neural ODE models are extensively presented in the Appendix E. In the main text, we utilize the basic predictor, as delineated in (33), while the exploration of its variants is reserved for the Appendix D.5. The fractal dimensions of some datasets are computed using the Compact-Box-Burning algorithm (Song et al., 2007). The *correlation between fractional dimension and the optimal fractional-derivative order* $\beta$, steering the extent of memorized dynamics over graph datasets, is delineated in Appendix D.11.

## 4.1 Node Classification of F-GRAND

**Datasets and splitting.** We utilize datasets with varied topologies, including citation networks (Cora (McCallum et al., 2004), Citeseer (Sen et al., 2008), Pubmed (Namata et al., 2012)), tree-structured datasets (Disease and Airport (Chami et al., 2019)), coauthor and co-purchasing graphs (CoauthorCS (Shchur et al., 2018), Computer and Photo (McAuley et al., 2015)), and the ogbn-arxiv dataset (Hu et al., 2020). We follow the same data splitting and pre-processing in (Chami et al., 2019) for Disease and Airport datasets. Consistent with experiment settings in GRAND (Chamberlain et al., 2021c), we use random splits for the largest connected component of each other dataset. We also incorporate the large-scale Ogbn-Products dataset (Hu et al., 2021) to demonstrate the scalability of the FROND framework, with the results displayed in Table 7.

**Methods.** For a comprehensive performance comparison, we select several prominent GNN models as baselines, including GCN (Kipf & Welling, 2017), and GAT (Veličković et al., 2018). Given the inclusion of tree-structured datasets, we also incorporate well-suited baselines: HGCN(Chami et al., 2019) and GIL (Zhu et al., 2020). To highlight the benefits of memorized dynamics in FROND, we include GRAND (Chamberlain et al., 2021c) as a special case of F-GRAND with $\beta = 1$. In line with (Chamberlain et al., 2021c), we examine two F-GRAND variants: F-GRAND-l (7) and F-GRAND-nl (9). Graph rewiring is not explored in this study. Where available, results from the paper (Chamberlain et al., 2021c) are used.

**Performance.** The results for graph node classification are summarized in Table 1, which also report the optimal $\beta$ obtained via hyperparameter tuning. Consistent with our expectations, F-GRAND surpasses GRAND across nearly all datasets, given that GRAND represents a special case of FROND with $\beta = 1$. This underscores the consistent performance enhancement offered by the integration of memorized dynamics. This advantage is particularly noticeable on tree-structured datasets such as Airports and Disease, where F-GRAND markedly outperforms the baselines. For instance, F-GRAND-l outperforms both GRAND and GIL by approximately 7% on the Airport dataset. Interestingly, our experiments indicate a smaller $\beta$ (signifying greater dynamic memory) is preferable for such fractal-structured datasets, aligning with previous studies on fractional differential equations in biological and chemical systems (Nigmatullin, 1986; Mandelbrot & Mandelbrot, 1982; Ionescu et al., 2017). We refer readers to Section 4.4 for more analysis of $\beta$. Supporting our intuition with evidence, we evaluated graph datasets' fractal dimensions using Compact-Box-Burning (Song et al., 2007), and compared it to the optimal $\beta$, fractal dimension, and $\delta$-hyperbolicity (as referenced in (Chami et al., 2019) for assessing tree-like structures—with lower values suggesting more tree-like graphs) as outlined in Table 18. Notably, we discerned a trend: a larger fractal dimension typically corresponds to a smaller optimal $\beta$. This observation strengthens our initial hypothesis in Section 1 that there exists some relationship between the fractal dimension and the order of the fractional dynamics.

Table 1: Node classification results(%) for random train-val-test splits. The best and the second-best result are highlighted in **red** and **blue**, respectively.

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

## 5 CONCLUSIONS

We introduced FROND, a novel graph learning framework that incorporates time-fractional Caputo derivatives to capture long-term memory in the graph feature updating dynamics. This approach has demonstrated improved performance over various traditional graph neural ODE models. The resulting framework paves the way for a new class of GNNs capable of addressing key challenges in the field, such as over-smoothing. Our results signify a promising step towards more effective graph representation learning by capitalizing on the power of fractional calculus.

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

representation learning. The work (Di Giovanni et al., 2023) suggests that gradient-flow message passing neural networks may be able to deal with heterophilic graphs provided that a residual connection is available. The paper (Gutteridge et al., 2023) proposes a spatial domain rewiring and focuses on long-range interactions. DRew in (Gutteridge et al., 2023) does not adhere to any ODE evolutionary structure. Its numerical experiments are also done on the long-range graph benchmark, instead of the usual GNN benchmark datasets we have used in our paper.

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

 3.5.]. We believe the above explanation facilitates understanding the relation between the Caputo derivative and its generalization of the integer-order derivative.

## B.2 COMPARISON BETWEEN RIEMANN–LIOUVILLE AND CAPUTO DERIVATIVE

Another well-known fractional derivative is the Riemann–Liouville derivative, which, however, sees less use in practical applications (see the section "Reasons for Choosing Caputo Derivative" for more insights). In this section, we offer a succinct introduction to the Riemann–Liouville derivative and compare it with Caputo's definition. The Riemann–Liouville fractional derivative is given as

$$
\widehat{D}_t^\beta f(t) := \frac{1}{\Gamma(\lceil \beta \rceil - \beta)} \frac{\mathrm{d}^{\lceil \beta \rceil}}{\mathrm{d}t^{\lceil \beta \rceil}} \int_0^t (t-\tau)^{\lceil \beta \rceil - \beta - 1} f(\tau) \mathrm{d}\tau \tag{23}
$$

Here again, we make the assumption that sufficient conditions are satisfied to ensure well-definiteness (refer to (Diethelm, 2010)[section 2.2] for details).

Next, we compare the Taylor expansion for the two definitions of fractional derivatives and the conventional integer-order derivative. This comparison clearly highlights the differences in the differential equations under the three definitions.

- **Classical Integer-order Taylor Expansion:** (Diethelm, 2010)[Theorem 2.C] Assume $f$ has absolutely continuous $(m-1)$-st derivative, we have that for $t \in [0, b]$,

$$
f(t) = \sum_{k=0}^{m-1} \frac{t^k}{k!} \frac{\mathrm{d}^k}{\mathrm{d}t^k} f(0) + J^m \frac{\mathrm{d}^m}{\mathrm{d}t^m} f(t) \tag{24}
$$

where $J^n f(x) := \frac{1}{\Gamma(n)} \int_0^t (t-\tau)^{n-1} f(\tau) \mathrm{d}\tau$ and note that here $k$ is a integer.

- **Riemann-Liouville Fractional Taylor Expansion:** (Diethelm, 2010)[Theorem 2.24] Let $n > 0$ and $m = \lceil n \rceil$. Assume that $f$ is s.t. $J^{m-n} f$ has absolutely continuous $(m-1)$-st derivative. Then,

$$
f(t) = \frac{t^{n-m}}{\Gamma(n-m+1)} J^{m-n} f(0) + \sum_{k=1}^{m-1} \frac{t^{k+n-m}}{\Gamma(k+n-m+1)} \widehat{D}_t^{k+n-m} f(0) + J^n \widehat{D}_t^n f(t). \tag{25}
$$

Note that in the case $n \in \mathbb{N}$ we have $m = n + 1$ and $\Gamma(n - m + 1) = \Gamma(0) = \infty$, the first term outside the sum vanishes. Hence, we can retrieve the classical result. For general $n$, *the order in $\widehat{D}_t^{k+n-m}$ is not a integer.*

• **Caputo Fractional Taylor Expansion:** (Diethelm, 2010)[Theorem 3.8.] Assume that $n \geq 0, m = \lceil n \rceil$, and $f$ has absolutely continuous $(m-1)$-st derivative. Then

$$f(t) = \sum_{k=0}^{m-1} \frac{D_t^k f(0)}{k!} t^k + J^n D_t^n f(t). \tag{26}$$

Note *the order in $D_t^k$ is still an integer.* If we compare (24) to (26), *it becomes evident that the Caputo derivative closely resembles the classical integer-order derivative in terms of Taylor expansion.* This fact will influence the initial conditions for differential equations, as introduced in the following section.

### B.3 (Caputo) Fractional Differential Equation

In this section, we first loosely compare the initial conditions for fractional differential equations under the Riemann-Liouville and Caputo definitions. Following this, we present the precise conditions for the existence and uniqueness of the solution to the fractional differential equation. As we will see, these conditions closely align with those of ordinary differential equations, conditions which are widely assumed by all graph neural ODE works such as the recent contributions like GRAND (Chamberlain et al., 2021c), GraphCON (Rusch et al., 2022), GRAND++ (Thorpe et al., 2022), GREAD (Choi et al., 2023) and CDE (Zhao et al., 2023). In short, all these graph neural ODE works can be seamlessly extended into our FROND framework with fractional dynamics!

#### B.3.1 Riemann-Liouville Case

Drawing from Riemann-Liouville fractional Taylor expansion, let's assume that $e$ is a given function with the property that there exists some function $g$ such that $g = \widehat{D}_t^\beta e$. The solution of the Riemann-Liouville differential equation is the form

$$\widehat{D}_t^\beta f = g \tag{27}$$

is given by

$$f(x) = e(x) + \sum_{j=1}^{\lceil \beta \rceil} c_j (x - a)^{n-j} \tag{28}$$

where $c_j$ are arbitrary constants. In other words, to uniquely determine the solution from (25), we should know the value of $\widehat{D}_t^{k+n-m} f(0)$. This is akin to the $k$ order ordinary differential equation where the initial conditions are assumed as $\frac{d^k}{dt^k} f(0)$, *with the distinction that the order in $\widehat{D}_t^{k+n-m}$ is not an integer.*

#### B.3.2 Caputo Case

Similarly, if $e$ is a given function with the property that $e = D_t^\beta g$ and if we intend to solve

$$D_t^\beta f = g \tag{29}$$

then we find

$$f(x) = e(x) + \sum_{j=1}^{\lceil \beta \rceil} c_j (x - a)^{\lceil \beta \rceil - j} \tag{30}$$

once more, with $c_j^*$ as arbitrary constants. Thus, to obtain a unique solution, it is most logical to prescribe the values of *integer order derivatives* $f(0), D_t^1 f(0), \ldots, D_t^{\lceil \beta \rceil - 1} f(0)$ in the Caputo setting, *irroring the traditional ordinary differential equation.* Whereas in the Riemann-Liouville case, one would more likely prescribe the fractional derivatives of $f$ at 0.

### B.3.3 EXISTENCE AND UNIQUENESS OF THE (CAPUTO) SOLUTION

Next, we delve into a general Caputo fractional differential equation, presented as follows:

$$D_t^\beta y(t) = g(t, y(t)) \tag{31}$$

conjoined with suitable initial conditions. As hinted in (29) and (30), the initial conditions take the form:

$$D_t^k y(0) = y_0^{(k)}, \quad k = 0, 1, \dots, \lceil \beta \rceil - 1. \tag{32}$$

- **Caputo existence and uniqueness theorem:** (Diethelm, 2010)[Theorem 6.8] Let $y_0^{(0)}, \dots, y_0^{(m-1)} \in \mathbb{R}$ and $h^* > 0$. Define the set $G := [0, h^*] \times \mathbb{R}$ and let the function $g : G \to \mathbb{R}$ be continuous and fulfill a *Lipschitz condition* with respect to the second variable, i.e.

$$|g(x, y_1) - g(x, y_2)| \le L |y_1 - y_2|$$

  with some constant $L > 0$ independent of $x, y_1$, and $y_2$. Then there *uniquely exists* function $y \in C[0, h^*]$ solving the initial value problem (31) and (32).

  For a point of reference, we also provide the well-known Picard–Lindelöf uniqueness theorem for ordinary differential equations.

- **Picard–Lindelöf theorem** (Hartman, 2002)[Page 8] Let $D \subseteq \mathbb{R} \times \mathbb{R}^n$ be a closed rectangle with $(t_0, y_0) \in \text{int } D$, the interior of $D$. Let $g : D \to \mathbb{R}^n$ be a function that is continuous in $t$ and *Lipschitz continuous* in $y$. Then, there exists some $\varepsilon > 0$ such that the initial value problem

$$y'(t) = g(t, y(t)), \quad y(t_0) = y_0.$$

  has a *unique solution* $y(t)$ on the interval $[t_0, t_0 + \varepsilon]$.

This allows us to draw parallels between the existence and uniqueness theorem of the Caputo fractional differential equation and its integer-order ordinary differential equation equivalent. We also remind readers that standard neural networks, as compositions of linear maps and pointwise non-linear activation functions with bounded derivatives (such as fully-connected and convolutional networks), satisfy global Lipschitz continuity with respect to the input. For attention neural networks, which are compositions of softmax and matrix multiplication, we observe local Lipschitz continuity. To see this, suppose $\mathbf{v} = \text{softmax}(\mathbf{u}) \in \mathbb{R}^{n \times 1}$. Then

$$\frac{d\mathbf{v}}{\partial \mathbf{u}} = \text{diag}(\mathbf{v}) - \mathbf{v}\mathbf{v}^\top = \begin{bmatrix} v_1(1 - v_1) & -v_1 v_2 & \dots & -v_1 v_n \\ -v_2 v_1 & v_2(1 - v_2) & \dots & -v_2 v_n \\ \vdots & \vdots & \ddots & \vdots \\ -v_n v_1 & -v_n v_2 & \dots & v_n(1 - v_n) \end{bmatrix}$$

For bounded input, we always have a bounded Jacobian. All the graph neural ODE works, such as recent contributions like GRAND (Chamberlain et al., 2021c), GraphCON (Rusch et al., 2022), GRAND++ (Thorpe et al., 2022), GREAD (Choi et al., 2023) and CDE (Zhao et al., 2023) safely assume the uniqueness of the solution to ODEs. *This means that all the graph neural ODE works can be securely extended into our FROND framework with fractional dynamics!*

### B.4 REASONS FOR CHOOSING CAPUTO DERIVATIVE

We now explain the reasons behind our preference for the Caputo fractional derivative:

1. As previously discussed, Caputo fractional differential equations align with ordinary differential equations concerning initial conditions.

2. The Caputo fractional derivative maintains a more intuitive resemblance to the integer-order derivative and satisfies the significant property of equating to zero when applied to a constant. This property is not satisfied by the Riemann-Liouville fractional derivative. Refer to (Diethelm, 2010)[Example 2.4. and Example 3.1.] for further clarification.

3. Given its widespread application in academic literature for practical use cases, numerical methods for solving Caputo fractional differential equations have been meticulously developed and exhaustively analyzed (Diethelm, 2010; Diethelm et al., 2004; Deng, 2007).

We remind readers that numerous methods for training neural ODEs, and consequently updating the weights $\theta$ in the neural network have been proposed. These include the autodifferentiation technique in PyTorch (Yan et al., 2018; Paszke et al., 2017), the adjoint sensitivity method (Chen et al., 2018b), and Snode (Quaglino et al., 2019). In our work, we employ the most straightforward autodifferentiation technique for training FROND with fractional neural differential equations, leveraging the numerical solvers outlined in (Diethelm, 2010; Diethelm et al., 2004; Deng, 2007). While we plan to investigate more sophisticated techniques for training FROND in future work, we have open-sourced our current solver implementations. We believe these will serve as valuable tools for the GNN community, encouraging the advancement of a unique class of GNNs that incorporate memory effects (fractional dynamics).

## C  Numerical Solvers for FROND

In the traditional graph ODE models outlined in (Chamberlain et al., 2021c; Thorpe et al., 2022; Rusch et al., 2022; Song et al., 2022; Choi et al., 2023; Zhao et al., 2023), the time parameter $t$ is a continuous counterpart to GNN layers, mirroring the concept of neural ODEs (Chen et al., 2018b) as continuous residual networks. In many numerical solvers for neural ODEs, time discretization is crucial. For instance, in the explicit Euler scheme, neural ODEs reduce to residual networks (with shared hidden layers) (Chen et al., 2018b). With more sophisticated discretization, like adaptive step size solvers (Atkinson et al., 2011), neural ODE solutions are accurate but demand more computational resources. Unlike prior studies, our work involves fractional-order differential equations, which are more complex than ODEs when $\beta$ takes non-integer values in FROND. We present the *fractional Adams–Bashforth–Moulton method* with three variants utilized in this work, demonstrating how time continues to serve as a continuous analog to the layer index and how the non-local nature of fractional derivatives leads to nontrivial dense or skip connections between layers. Additionally, we also present one implicit L1 solver for solving FROND when $\beta$ is not an integer. It is worth noting that various neural ODE solvers remain applicable for FROND when $\beta$ is an integer.

We first recall the FROND framework

$$D_t^\beta \mathbf{X}(t) = \mathcal{F}(\mathbf{W}, \mathbf{X}(t)), \quad \beta > 0,$$

where $\beta$ denotes the fractional order of the derivative, and $\mathcal{F}$ is a dynamic operator on the graph like the models presented in Section 2.2. The initial condition is set as $\mathbf{X}^{(\lceil \beta \rceil - 1)}(0) = \ldots = \mathbf{X}(0) = \mathbf{X}$ consisting of the preliminary node features, where $\lceil \beta \rceil$ denotes the smallest integer greater than or equal to $\beta$, akin to the initial conditions seen in ODEs.

### C.1  Basic predictor

Referencing (Diethelm et al., 2004), we first employ a preliminary numerical solver called "predictor" through time discretisation $t_j = jh$, where the discretisation parameter $h$ is a small positive value:

$$\mathbf{X}^{\mathrm{P}}(t_n) = \sum_{j=0}^{\lceil \beta \rceil - 1} \frac{t_n^j}{j!} \mathbf{X}^{(k)}(0) + \frac{1}{\Gamma(\beta)} \sum_{j=0}^{n-1} \mu_{j,n} \mathcal{F}(\mathbf{W}, \mathbf{X}(t_j)), \tag{33}$$

where $\mu_{j,n} = \frac{h^\beta}{\beta} \left( (n-j)^\beta - (n-1-j)^\beta \right)$ and $h = t_n - t_{n-1}$ represents the temporal step size. When $\beta = 1$, this method simplifies to the Euler solver in (Chen et al., 2018b; Chamberlain et al., 2021c) as $\mu_{j,n} \equiv h$, yielding $\mathbf{X}^{\mathrm{P}}(t_n) = \mathbf{X}^{\mathrm{P}}(t_{n-1}) + h\mathcal{F}(\mathbf{W}, \mathbf{X}(t_{n-1}))$. Thus, our basic predictor can be considered as the fractional Euler method or fractional Adams–Bashforth method, which is a generalization of the Euler method used in (Chen et al., 2018b; Chamberlain et al., 2021c). However, when $\beta < 1$, we need to utilize the full memory $\{\mathcal{F}(\mathbf{W}, \mathbf{X}(t_j))\}_{j=0}^{n-1}$.

The block diagram of this basic predictor, shown in Fig. 2, reveals that our framework introduces nontrivial dense or skip connections between layers. A more refined visualization is conveyed in

Fig. 3, elucidating the manner in which information propagates through layers and the graph's spatial domain.

## C.2 PREDICTOR-CORRECTOR

The corrector formula from (Diethelm et al., 2004), a fractional variant of the one-step Adams-Moulton method, refines the initial approximation using the predictor $\mathbf{X}(t_n)^{\mathrm{P}}$ as follows:

$$\mathbf{X}(t_n) = \sum_{j=0}^{\lceil\beta\rceil-1} \frac{t_n^j}{j!} \mathbf{X}^{(k)}(0) + \frac{1}{\Gamma(\beta)} \sum_{j=0}^{n-1} \eta_{j,n} \mathcal{F}(\mathbf{W}, \mathbf{X}(t_j)) + \frac{1}{\Gamma(\beta)} \eta_{n,n} \mathcal{F}(\mathbf{W}, \mathbf{X}^{\mathrm{P}}(t_n)), \quad (34)$$

Here we show the coefficients $\eta_{j,n}$ in the predictor-corrector variant (34) from (Diethelm et al., 2004):

$$\eta_{j,n}(\beta) = \frac{h^\beta}{\beta(\beta+1)} \times \begin{cases} (n-1)^{\beta+1} - (n-1-\beta)n^\beta & \text{if } j = 0 \\ (n-j+1)^{\beta+1} + (n-1-j)^{\beta+1} - 2(n-j)^{\beta+1} & \text{if } 1 \le j \le n-1 \\ 1 & \text{if } j = n \end{cases}$$

$$(35)$$

## C.3 SHORT MEMORY PRINCIPLE

When $T$ is large, computational time complexity becomes a challenge due to the non-local nature of fractional derivatives. To mitigate this, (Deng, 2007; Podlubny, 1999) suggest leveraging the short memory principle to modify the summation in (33) and (34) to $\sum_{j=n-K}^{n-1}$. This corresponds to employing a shifting memory window with a fixed width $K$. The block diagram is depicted in Fig. 2.

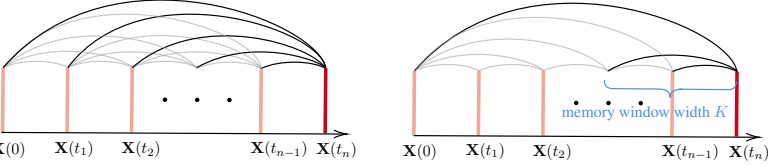

Figure 2: Diagrams of fractional Adams–Bashforth–Moulton method with full (left) and short (right) memory.

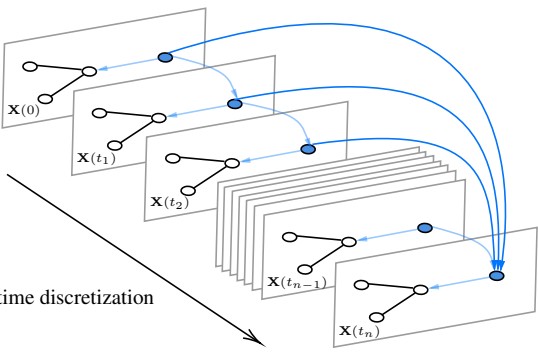

Figure 3: Model discretization in FROND with the basic predictor solver. Unlike the Euler discretization in ODEs, FDEs incorporate connections to historical times, introducing memory effects. Specifically, the dark blue connections observed in FDEs are absent in ODEs. The weight of these skip connections correlates with $\eta_{j,n}(\beta)$ as detailed in (35).

## C.4 L1 SOLVER

The L1 scheme is one of the most popular methods to approximate the Caputo fractional derivative in time. It utilizes a backward differencing method for effective approximation of derivatives. Referencing to (Gao & Sun, 2011; Sun & Wu, 2006), we have the L1 approximation of Caputo fractional

derivative as follows:

$$D_t^\beta \mathbf{X}(t_k) \approx \mu \sum_{j=0}^{k-1} R_{k,j}^\beta (\mathbf{X}(t_{j+1}) - \mathbf{X}(t_j))$$

where $h$ is the temporal step size,

$$\mu = \frac{1}{h^\beta \Gamma(2-\beta)}, \qquad R_{k,j}^\beta = (k-j)^{1-\beta} - (k-j-1)^{1-\beta}, \qquad 0 \le j \le k-1.$$

Applying L1 solver for our problem, we obtain

$$\mu \sum_{j=0}^{k-1} R_{k,j}^\beta (\mathbf{X}(t_{j+1}) - \mathbf{X}(t_j)) = (\mathbf{A}(\mathbf{X}(t_k)) - \mathbf{I})\mathbf{X}(t_k)$$

Manipulating the above equation, we obtain

$$\mathbf{X}(t_k) - \frac{1}{\mu}(\mathbf{A}(\mathbf{X}(t_k)) - \mathbf{I})\mathbf{X}(t_k) = \mathbf{X}(t_{k-1}) - \sum_{j=0}^{k-2} R_{k,j}^\beta (\mathbf{X}(t_{j+1}) - \mathbf{X}(t_j))$$

The above formula is an implicit nonlinear scheme. To solve it without calculating the inversion of a matrix, we propose the following iteration method:

(1) we can get a basic approximation of $\mathbf{X}(t_k)$ with the following formula:

$$\mathbf{X}^{\mathrm{P}}(t_k) - \frac{1}{\mu}(\mathbf{A}(\mathbf{X}(t_{k-1})) - \mathbf{I})\mathbf{X}(t_{k-1}) = \mathbf{X}(t_{k-1}) - \sum_{j=0}^{k-2} R_{k,j}^\beta (\mathbf{X}(t_{j+1}) - \mathbf{X}(t_j))$$

(2) After that, we can substitute the above $\mathbf{X}^{\mathrm{P}}(t_k)$ into the implicit scheme to update $\mathbf{X}(t_k)$:

$$\mathbf{X}(t_k) - \frac{1}{\mu}(\mathbf{A}(\mathbf{X}^{\mathrm{P}}(t_k)) - \mathbf{I})\mathbf{X}^{\mathrm{P}}(t_k) = \mathbf{X}(t_{k-1}) - \sum_{j=0}^{k-2} R_{k,j}^\beta (\mathbf{X}(t_{j+1}) - \mathbf{X}(t_j)) \qquad (36)$$

The step (2) can be repeated multiple times to get an accurate approximation of $\mathbf{X}(t_k)$.

## D  DATASETS, SETTINGS AND MORE EXPERIMENTS FOR **F-GRAND** MODEL

### D.1  DATASETS

The statistics for the datasets used in Table 1 are reported in Table 5. Adhering to the experimental framework in (Chamberlain et al., 2021c), we applied the largest connected component from each dataset, with the exclusive exception of tree-like graph datasets, specifically, Airport and Disease. Note however, in the study of over-smoothness, we utilize the fixed data splitting over the full

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

 | 97.0±0.79 | 97.09±0.87 | 96.97±0.84 | 96.50±0.60 | 97.41±0.42 | 96.53±0.74 | 97.03±0.55 | 94.91±3.72 |

#### D.7.2 DIRICHLET ENERGY

The Dirichlet Energy defined on the graph is represented as:

$$\mathbf{E}(\mathbf{X}(t)) := \sum_{i \in \mathcal{V}} \sum_{j \in \mathcal{V}} \left\| \mathbf{x}^{(i)}(t) - \mathbf{x}^{(j)}(t) \right\|_2^2 \tag{37}$$

Dirichlet Energy provides quantitative insights into the variability of features across nodes and their neighbors. Higher Dirichlet Energy implies greater diversity in node features, suggesting lower over-smoothing levels, while lower energy points to the contrary, indicating a possible risk of information loss through excessive smoothing.

We visualize the Dirichlet Energy of both Cora and Airport datasets across different models in Figures Fig. 4 and Fig. 5, respectively. The term "number of layers" for both the GRAND and F-GRAND models refers to the time $T$ of integration, calculated using the Euler solver and the basic predictor solver, respectively. This interpretation of layers is pivotal as it extends the discrete layer concept in traditional models to a continuous-time framework. Observations indicate that our F-GRAND model exhibits slower convergence compared to GRAND on the Cora dataset, while maintaining nearly consistent Dirichlet Energy values on the Airport dataset up to 120 layers. This consistency underscores its competence in mitigating the over-smoothing problem. As we have proven in Corollary 1, the Dirichlet Energy will *asymptotically* approach 0 at a slow algebraic rate. The sustained plot on the Airport dataset could arise from inadequacies in the layer count and the numerical precision of the solvers. Nonetheless, the depiction of Dirichlet Energy provides

Table 15: Node classification accuracy(%) under different value of $\beta$ when time $T = 8$.

| $\beta$ | 0.1 | 0.2 | 0.3 | 0.4 | 0.5 | 0.6 | 0.7 | 0.8 | 0.9 | 1.0 |
|---|---|---|---|---|---|---|---|---|---|---|
| Cora | 74.80±0.42 | 76.10±0.34 | 77.0±0.98 | 77.80±0.75 | 79.60±0.91 | 80.79±0.58 | 81.56±0.30 | 82.44±0.51 | 82.68±0.64 | 82.37±0.59 |
| Airport | 97.09±0.87 | 96.67±0.91 | 95.80±2.03 | 94.04±3.62 | 91.66±6.34 | 89.24±7.87 | 84.36±8.04 | 79.29±6.01 | 78.73±6.33 | 78.88±9.67 |

substantial evidence of FROND's potential in alleviating over-smoothness. It is worth highlighting that, particularly on tree-structured datasets, F-GRAND stands out as the sole model capable of alleviating over-smoothness. This observation is consistent with the findings presented in Fig. 1 of the main paper.

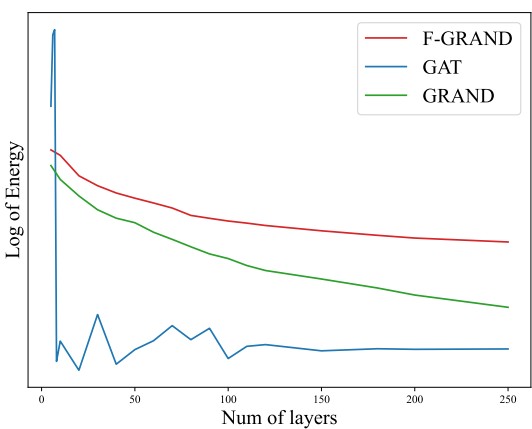

Figure 4: Dirichlet Energy of Cora dataset

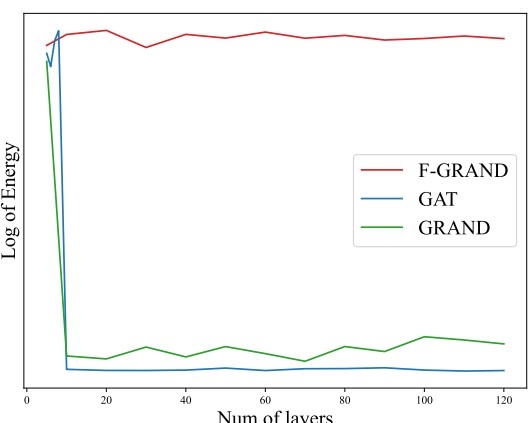

Figure 5: Dirichlet Energy of Airport dataset.

### D.8   ABLATION STUDY: SELECTION OF $\beta$ CONTINUED

In the main paper, we explore the impact of the fractional order parameter $\beta$ across a variety of graph datasets, with the results of these investigations presented in Table 3. More comprehensive details concerning the variations in $\beta$ can be found in Table 15.

### D.9   ROBUSTNESS AGAINST ADVERSARIAL ATTACKS

Despite the significant advancements GNNs have made in inference tasks on graph-structured data, they are recognized as being susceptible to adversarial attacks (Zügner et al., 2018). Adversaries, aiming to deceive a trained GNN, can either introduce new nodes into the graph during the inference phase, known as an injection attack (Wang et al., 2020; Zheng et al., 2022; Zou et al., 2021; Hussain et al., 2022), or manipulate the graph's topology by adding or removing edges, termed as a modification attack (Chen et al., 2018a; Waniek et al., 2018; Du et al., 2017). In this section, we present preliminary experiments assessing the robustness of our model against adversarial attacks. Specifically, we carry out graph modification adversarial attacks using the Metattack method (Zügner & Günnemann, 2019). Our approach adheres to the attack setting described in Pro-GNN (Jin et al., 2020), and we utilize the perturbed graph provided by the DeepRobust library (Li et al., 2020b) to ensure a fair comparison. The perturbation rate, indicating the proportion of altered edges, is incrementally adjusted in 5% steps from 0% to 25%.

The results of these experiments are presented in Table 16. It should be noted that the impact of Meta-attacks with higher strengths detrimentally affects the performance of all models under test. However, our FROND-nl model consistently demonstrates enhanced resilience against adversarial attacks compared to the baselines, including GRAND-nl. For instance, at a perturbation rate of 25%, F-GRAND-nl outshines the baselines by an estimated margin of 10 to 15% on the Cora dataset.

Comprehensive testing against a variety of adversarial attack methods constitutes an important direction for our future work.

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

$$= b_n\mathbb{P}\Big(\mathbf{R}(t_0) = \mathbf{x}^{(h)}(0)\Big) + c_n\mathbb{P}\Big(\mathbf{R}(t_1) = \mathbf{x}^{(h)}(0)\Big) + \ldots + c_2\mathbb{P}\Big(\mathbf{R}(t_{n-1}) = \mathbf{x}^{(h)}(0)\Big) +$$

$$+ (c_1 - \sigma^\beta)\mathbb{P}\Big(\mathbf{R}(t_n) = \mathbf{x}^{(h)}(0)\Big) + \sum_{j=1}^{n}\sigma^\beta\frac{W_{jh}}{d_j}\mathbb{P}\Big(\mathbf{R}(t_n) = \mathbf{x}^{(j)}(0)\Big)$$

$$= b_n\mathbb{P}\Big(\mathbf{R}(t_0) = \mathbf{x}^{(h)}(0)\Big) + c_n\mathbb{P}\Big(\mathbf{R}(t_1) = \mathbf{x}^{(h)}(0)\Big) + \ldots + c_2\mathbb{P}\Big(\mathbf{R}(t_{n-1}) = \mathbf{x}^{(h)}(0)\Big) +$$

$$+ c_1\mathbb{P}\Big(\mathbf{R}(t_n) = \mathbf{x}^{(h)}(0)\Big) - \sigma^\beta\mathbb{P}\Big(\mathbf{R}(t_n) = \mathbf{x}^{(h)}(0)\Big) + \sum_{j=1}^{N}\sigma^\beta\frac{W_{jh}}{d_j}\mathbb{P}\Big(\mathbf{R}(t_n) = \mathbf{x}^{(j)}(0)\Big)$$

$$(47)$$

By rearranging, we have that

$$\mathbb{P}\Big(\mathbf{R}(t_{n+1}) = \mathbf{x}^{(h)}(0)\Big) - \sum_{k=1}^{n} c_k\mathbb{P}\Big(\mathbf{R}(t_{n+1-k}) = \mathbf{x}^{(h)}(0)\Big) - b_n\mathbb{P}\Big(\mathbf{R}(t_0) = \mathbf{x}^{(h)}(0)\Big)$$

$$= (-1)^0\binom{\beta}{0}\mathbb{P}\Big(\mathbf{R}(t_{n+1}) = \mathbf{x}^{(h)}(0)\Big) - \sum_{k=1}^{n}(-1)^{k+1}\binom{\beta}{k}\mathbb{P}\Big(\mathbf{R}(t_{n+1-k}) = \mathbf{x}^{(h)}(0)\Big) - \sum_{k=0}^{n}(-1)^k\binom{\beta}{k}\mathbb{P}\Big(\mathbf{R}(0) = \mathbf{x}^{(h)}(0)\Big)$$

$$= \sum_{k=0}^{n}(-1)^k\binom{\beta}{k}\mathbb{P}\Big(\mathbf{R}(t_{n+1-k}) = \mathbf{x}^{(h)}(0)\Big) - \sum_{k=0}^{n}(-1)^k\binom{\beta}{k}\mathbb{P}\Big(\mathbf{R}(0) = \mathbf{x}^{(h)}(0)\Big)$$

$$= \sum_{k=0}^{n}(-1)^k\binom{\beta}{k}\Big[\mathbb{P}\Big(\mathbf{R}(t_{n+1-k}) = \mathbf{x}^{(h)}(0)\Big) - \mathbb{P}\Big(\mathbf{R}(0) = \mathbf{x}^{(h)}(0)\Big)\Big]$$

$$= -\sigma^\beta\mathbb{P}\Big(\mathbf{R}(t_n) = \mathbf{x}^{(h)}(0)\Big) + \sum_{j=1}^{n}\sigma^\beta\frac{W_{jh}}{d_j}\mathbb{P}\Big(\mathbf{R}(t_n) = \mathbf{x}^{(j)}(0)\Big)$$

Dividing both sides of the final equality by $\sigma^\beta$, it follows that

$$\sum_{k=0}^{n}(-1)^k\binom{\beta}{k}\frac{\mathbb{P}\big(\mathbf{R}(t_{n+1-k}) = \mathbf{x}^{(h)}(0)\big) - \mathbb{P}\big(\mathbf{R}(0) = \mathbf{x}^{(h)}(0)\big)}{\sigma^\beta}$$

$$= -\mathbb{P}\Big(\mathbf{R}(t_n) = \mathbf{x}^{(h)}(0)\Big) + \sum_{j=1}^{N}\frac{W_{jh}}{d_j}\mathbb{P}\Big(\mathbf{R}(t_n) = \mathbf{x}^{(j)}(0)\Big) \qquad (48)$$

From the Grünwald-Letnikov fractional derivatives formulation (Podlubny, 1999)[eq. (2.54)], the limit of LHS of (48) is

$$\lim_{\substack{\sigma \to 0 \\ n\sigma=t}}\sum_{k=0}^{n}(-1)^k\binom{\beta}{k}\frac{\mathbb{P}\big(\mathbf{R}(t_{n+1-k}) = \mathbf{x}^{(h)}(0)\big) - \mathbb{P}\big(\mathbf{R}(0) = \mathbf{x}^{(h)}(0)\big)}{\sigma^\beta} = D_t^\beta\mathbb{P}\Big(\mathbf{R}(t) = \mathbf{x}^{(h)}(0)\Big)$$

$$(49)$$

On the other hand, the RHS of (48) is

$$-\mathbb{P}\Big(\mathbf{R}(t_n) = \mathbf{x}^{(h)}(0)\Big) + \sum_{j=1}^{N}\frac{W_{jh}}{d_j}\mathbb{P}\Big(\mathbf{R}(t_n) = \mathbf{x}^{(j)}(0)\Big) = [-\mathbf{L}\mathbb{P}(\mathbf{R}(t_n))]_h \qquad (50)$$

where $\mathbb{P}(\mathbf{R}(t_n))$ is the probability (column) vector with $j$-th element being $\mathbb{P}\big(\mathbf{R}(t_n) = \mathbf{x}^{(j)}(0)\big)$, and $[-\mathbf{L}\mathbb{P}(\mathbf{R}(t_n))]_h$ denotes the $h$-th element of the vector $-\mathbf{L}\mathbb{P}(\mathbf{R}(t_n))$.

Putting them together, we have

$$D_t^\beta\mathbb{P}(\mathbf{R}(t)) = -\mathbf{L}\mathbb{P}(\mathbf{R}(t)) \qquad (51)$$

since we assume $t_n = t$ in the limit. Finally, from the linearity of the operator $D_t^\beta$ and $\mathbf{L}$, we have

$$D_t^\beta \mathbb{P}(\mathbf{R}(t))\mathbf{X}(0) = -\mathbf{L}\mathbb{P}(\mathbf{R}(t))\mathbf{X}(0), \tag{52}$$

which states that $D_t^\beta \mathbb{E}\mathbf{R}(t) = -\mathbf{L}\mathbb{E}\mathbf{R}(t)$ for any probability distribution $\mathbb{P}(\mathbf{R}(0))$. For each $i$, if $\mathbf{R}(t_0) = \mathbf{x}^{(i)}$ with probability one, the initial condition of (9) is satisfied. The proof of Theorem 1 is now complete. $\qquad\square$

### F.2 Proof of Theorem 2

Before presenting the formal proof, we aim to provide additional insights and intuition regarding the algebraic convergence from two perspectives.

- Fractional Random Walk Perspective: In a standard random walk, a walker moves to a new position at each time step without delay. However, in a fractional random walk, which is more reflective of our model's behavior, the walker has a probability of revisiting past positions. This revisitation is not arbitrary; it's governed by a waiting time that follows a power-law distribution with a long tail. This characteristic fundamentally changes the walk's dynamics, introducing a memory component and leading to a slower, algebraic rate of convergence. This behavior is intrinsically different from normal random walks, where the absence of waiting times facilitates a quicker, often exponential, convergence.

- Analytic Perspective: From an analytic perspective, the essential slow algebraic rate primarily stems from the slow convergence of the Mittag-Leffler function towards zero. To elucidate this, let's consider the scalar scenario. Recall that the Mittag-Leffler function $E_\beta$ is defined as:

$$E_\beta(z) := \sum_{j=0}^{\infty} \frac{z^j}{\Gamma(j\beta + 1)}$$

  for values of $z$ where the series converges. Specifically, when $\beta = 1$,

$$E_1(z) = \sum_{j=0}^{\infty} \frac{z^j}{\Gamma(j+1)} = \sum_{j=0}^{\infty} \frac{z^j}{j!} = \exp(z)$$

  corresponds to the well-known exponential function. According to [A1, Theorem 4.3.], the eigenfunctions of the Caputo derivative are expressed through the Mittag-Leffler function. In more precise terms, if we define $y(t)$ as

$$y(t) := E_\beta\left(-\lambda t^n\right), \quad t \geq 0,$$

  it follows that

$$D_t^\beta y(t) = -\lambda y(t)$$

  Notably, when $\beta = 1$, this reduces to $\frac{\mathrm{d}\exp(-\lambda t)}{\mathrm{d}t} = -\lambda \exp(-\lambda t)$.
  Further, we examine the behavior of $E_\beta\left(-\lambda t^n\right)$. As per [A1, Theorem 7.3.], it is noted that:
  (a) The function $y(t)$ is completely monotonic on $(0, \infty)$.
  (b) As $x \to \infty$,

$$y(t) = \frac{t^{-\beta}}{\lambda\Gamma(1-\beta)}(1 + o(1)).$$

  Thus, the function $E_\beta\left(-\lambda t^\beta\right)$ converges to zero at a rate of $\Theta\left(t^{-\beta}\right)$. Our paper extends this to the general high-dimensional case by replacing the scalar $\lambda$ with the Laplacian matrix $\mathbf{L}$, wherein the eigenvalues of $\mathbf{L}$ play a critical role analogous to $\lambda$ in the scalar case.
  For a diagonalizable Laplacian matrix $\mathbf{L}$, the proof essentially reverts to the scalar case as outlined above (refer to (55) in our paper). However, in scenarios where $\mathbf{L}$ is non-diagonalizable and has a general Jordan normal form, it becomes necessary to employ the Laplace transform technique to demonstrate that the algebraic rate remains valid (refer to the context between (55) and (56) in our paper).

*Proof.* We first prove the stationary probability $\boldsymbol{\pi} = \left( \frac{d_1}{\sum_{j=1}^N d_j}, \ldots, \frac{d_N}{\sum_{j=1}^N d_j} \right)$ by induction. Assume that for $i = 1, \ldots, n$, the probability distribution $\mathbb{P}(\mathbf{R}(t_n))$ always equals $\boldsymbol{\pi}^\intercal$. For $i = n + 1$, from (47), it follows that

$$
\begin{aligned}
{[\mathbb{P}(\mathbf{R}(t_{n+1}))]_h} &= \mathbb{P}\left( \mathbf{R}(t_{n+1}) = \mathbf{x}^{(h)}(0) \right) \\
&= b_n \mathbb{P}\left( \mathbf{R}(t_0) = \mathbf{x}^{(i)}(0) \right) + \sum_k c_k \mathbb{P}\left( \mathbf{R}(t_{n+1-k}) = \mathbf{x}^{(h)}(0) \right) \\
&\quad - \sigma^\beta \mathbb{P}\left( \mathbf{R}(t_n) = \mathbf{x}^{(h)}(0) \right) + \sum_{j=1}^N \sigma^\beta \frac{W_{jh}}{d_j} \mathbb{P}\left( \mathbf{R}(t_n) = \mathbf{x}^{(j)}(0) \right) \\
&= \boldsymbol{\pi}_h b_n + \sum_{k=1}^n \boldsymbol{\pi}_h c_k - \boldsymbol{\pi}_h \sigma^\beta + \sum_{j=1, j \neq h}^N \boldsymbol{\pi}_j \sigma^\beta \frac{W_{jh}}{d_j} \\
&= \boldsymbol{\pi}_h (b_n + \sum_{k=1}^n c_k) - \boldsymbol{\pi}_h \sigma^\beta + \sum_{j=1, j \neq h}^N \frac{d_j}{\sum_{j=1}^N d_j} \sigma^\beta \frac{W_{jh}}{d_j} \\
&= \boldsymbol{\pi}_h - \boldsymbol{\pi}_h \sigma^\beta + \sigma^\beta \sum_{j=1, j \neq h}^N \frac{W_{jh}}{\sum_{j=1}^N d_j} \\
&= \boldsymbol{\pi}_h - \boldsymbol{\pi}_h \sigma^\beta + \sigma^\beta \frac{d_h}{\sum_{j=1}^N d_j} \\
&= \boldsymbol{\pi}_h.
\end{aligned}
$$

This proves the the existence of stationary probability. The uniqueness follows from if $\mathbb{P}(\mathbf{R}(t_1)) = \boldsymbol{\pi}' \neq \boldsymbol{\pi}$, we do not have $\mathbb{P}(\mathbf{R}(t_2)) = \mathbb{P}(\mathbf{R}(t_1))$ since otherwise it indicate that the Markov chain defined by

$$
\begin{aligned}
&\mathbb{P}\left( \mathbf{R}(t_{n+1}) = \mathbf{x}^{(j_{n+1})}(0) \mid \mathbf{R}(t_n) = \mathbf{x}^{(j_n)}(0) \right) = \mathbb{P}\left( \mathbf{R}(t_{n+1}) = \mathbf{x}^{(j_2)}(0) \mid \mathbf{R}(t_n) = \mathbf{x}^{(j_1)}(0) \right) \\
&= \begin{cases} c_1 - \sigma^\beta + b_1 & \text{if staying at current location with } j_{n+1} = j_n \\ \sigma^\beta \frac{W_{j_n j_{n+1}}}{d_{j_n}} & \text{if jumping to neighboring nodes with } j_{n+1} \neq j_n \end{cases}
\end{aligned} \tag{53}
$$

has stationary distribution other than $\boldsymbol{\pi}$ which contradicts the assumption of a strongly connected and aperiodic graph.

We now establish the algebraic convergence. Consider $\mathbf{L} = \mathbf{S}\mathbf{J}\mathbf{S}^{-1}$ as the Jordan canonical form of $\mathbf{L}$. It is evident that for the matrix $\mathbf{W}\mathbf{D}^{-1}$, since $\mathbf{W}\mathbf{D}^{-1}$ is left stochastic and the graph is strongly connected and aperiodic, the Perron-Frobenius theorem (Horn & Johnson, 2012)[Lemma 8.4.3., Theorem 8.4.4] confirms that the value 1 is the sole eigenvalue of it that equals the spectral radius 1. Hence, we have that $\mathbf{L} = \mathbf{I} - \mathbf{W}\mathbf{D}^{-1}$ possesses an eigenvalue of 0, and all the remaining eigenvalues have positive real parts. Consequently, $\mathbf{J}$ contains a block that consists of only a single 0. We can rewrite (51) as

$$
D_t^\beta \mathbf{Y}(t) = -\mathbf{J}\mathbf{Y}(t) \tag{54}
$$

where $\mathbf{S}^{-1}\mathbb{P}(\mathbf{R}(t)) = \mathbf{Y}(t) \in \mathbb{R}^N$ and the inital condition is $\mathbf{S}^{-1}\mathbb{P}(\mathbf{R}(0)) = \mathbf{Y}(0)$.

If $\mathbf{L}$ is diagonalizable, then $\mathbf{J}$ is a diagonal matrix with the diagonal elements being the eigenvalues. We have an uncoupled set of equations in the form $D_t^\beta \mathbf{Y}_k(t) = -\lambda_k \mathbf{Y}_k(t)$, where $\mathbf{Y}_k$ is the $k$-th component of $\mathbf{Y}$. According to (Podlubny, 1999), the solution is given by

$$
\mathbf{Y}_k(t) = \mathbf{Y}_k(0) E_\beta(-\lambda_k t^\beta) \tag{55}
$$

where $E_\beta(\cdot)$ is the Mittag-Leffler function define as $E_\beta(z) = \sum_{j=0}^\infty \frac{z^j}{\Gamma(\beta j + \beta)}$ and $\Gamma(\cdot)$ is the gamma function. For the index $w$ s.t. the eigenvalue $\lambda_w = 0$, we have the solution $\mathbf{Y}_k(t) = \mathbf{Y}_k(0)$ which corresponds to the stationary probability vector if we transform it back to $\mathbb{P}(\mathbf{R}(t))$. From (Mainardi,

2014), we have that for $k \neq w$, the convergence to 0 is in the following order

$$\mathbf{Y}_k(t) = \Theta(t^{-\beta}).$$

If $\mathbf{J}$ is not diagonal, the entries of $\mathbf{Y}(t)$ that correspond to distinct Jordan blocks in $\mathbf{J}$ are not coupled. W.L.O.G, we assume the first Jordan block is associated to eigenvalue $\lambda_1 = 0$, while all eigenvalues $\lambda_k > 0$, for $k = 2, \ldots, N$. A consideration of the Jordan block corresponding to one $\lambda_k$, $k = 2, \ldots, N$, is adequate. We assume the Jordan block $\mathbf{J}(\lambda_k)$ corresponding to $\lambda_k$ has size $m$. It follows that for this Jordan block we have

$$D_t^\beta \mathbf{Y}_1(t) = \lambda_k \mathbf{Y}_1(t) + \mathbf{Y}_2(t)$$

$$\vdots \quad \vdots$$

$$D_t^\beta \mathbf{Y}_{m-1}(t) = \lambda_k \mathbf{Y}_{m-1}(t) + \mathbf{Y}_m(t)$$

$$D_t^\beta \mathbf{Y}_m(t) = \lambda_k \mathbf{Y}_m(t)$$

which can be solved from the bottom up. Starting with the last equation, we have that

$$\mathbf{Y}_m(t) = \mathbf{Y}_m(0)E_\beta(-\lambda_k t^\beta) = \Theta(t^{-\beta}).$$

Furthermore, we have

$$D_t^\beta \mathbf{Y}_{m-1}(t) = \lambda_k \mathbf{Y}_{m-1}(t) + \mathbf{Y}_m(0)E_\beta(-\lambda_k t^\beta)$$

Take the Laplace transform, we have

$$\mathcal{L}\left\{D_t^\beta \mathbf{Y}_{m-1}(t)\right\} = s^\beta Y_{m-1}(s) - s^{\beta-1}\mathbf{Y}_{m-1}(0)$$

where $Y_{m-1}(s)$ is the Laplace transform of $\mathbf{Y}_{m-1}(t)$ according to (3). Now, for the right hand side, we have $L\left\{\lambda \mathbf{Y}_{m-1}(t)\right\} = \lambda_k Y_{m-1}(s)$ and we know that the Laplace transform of $E_\beta\left(-\lambda_k t^\beta\right)$ is $\frac{s^{\beta-1}}{(s^\beta + \lambda_k)}$. Therefore, the equation in the Laplace domain becomes:

$$s^\beta Y_{m-1}(s) - s^{\beta-1}\mathbf{Y}_{m-1}(0) = \lambda_k Y_{m-1}(s) + \mathbf{Y}_m(0)\frac{s^{\beta-1}}{s^\beta + \lambda_k}$$

Rearranging this equation to solve for $Y_{m-1}(s)$ gives:

$$Y_{m-1}(s) = \frac{s^{\beta-1}\mathbf{Y}_{m-1}(0) + \mathbf{Y}_m(0)\frac{s^{\beta-1}}{s^\beta + \lambda_k}}{s^\beta + \lambda_k}$$

It follows that $Y_{m-1}(s) \sim Cs^{\beta-1}$ when $s \to 0$. We can repeat the above process to show $Y_i(s) \sim Cs^{\beta-1}$ when $s \to 0$ for all $i = 1, \ldots, m-2$. According to the Hardy–Littlewood Tauberian theorem (theorem, 2023), we have that, for all $i = 1, \ldots, m$,

$$\mathbf{Y}_i(t) = \Theta(t^{-\beta}). \tag{56}$$

The proof is now complete. $\qquad\square$

### F.3 Proof of Corollary 1

We are unable to directly invoke (15) to infer $\mathbf{E}(\mathbf{X}(t)) = \Theta(t^{-2\beta})$ in (17) since it only yields an upper bound, as presented below:

$$\left\|\mathbf{x}^{(i)}(t) - \mathbf{x}^{(j)}(t)\right\|_2^2 \leq \left(\|\mathbf{x}^{(i)}(t) - \mathbf{x_s}\|_2 + \|\mathbf{x}^{(j)}(t) - \mathbf{x_s}\|_2\right)^2 \tag{57}$$

Consequently, we directly refer to (54) for resolution. We use notation $\mathbf{e}_i$ to denote the one-hot vector where the $i$-th component stands at 1. Recall that we have solution $\mathbb{P}_i(\mathbf{R}(t))$ with initial condition $\mathbf{e}_i$ for each $i$. The set $\{\mathbf{e}_i\}_{i=1}^N$ is linearly independent and span the full $\mathbb{R}^N$ space. It is equivalent to getting the transformed solution $\mathbf{Y}_{(i)}(t)$ with the initial condition $\mathbf{S}^{-1}\mathbf{e}_i$ in (54). The entries of $\mathbf{Y}(t)$ that correspond to distinct Jordan blocks in $\mathbf{J}$ are not coupled. We denote the solution to (54) with initial condition $\mathbf{e}_k$ as $\bar{\mathbf{Y}}_{(k)}(t)$. Note, according to the proof of theorem 2, we have that the solution corresponds to the unique eigenvalue 0 to matrix $\mathbf{J}$ keep a constant. If we assume eigenvalue 0 is

the first Jodan block, we have $\bar{\mathbf{Y}}_{(1)}(t) = \bar{\mathbf{Y}}_{(1)}(0)$ for all time $t \geq 0$. While all the other solutions $\bar{\mathbf{Y}}_{(k)}(t)$, $k = 2, ..., N$, corresponding to the other Jordan blocks, converge to 0 in $\Theta\left(t^{-\beta}\right)$ rate. From the linearity, we then have $\mathbf{Y}_{(i)}(t)$ are the linear combination of the $N$ independent solutions $\{\bar{\mathbf{Y}}_{(k)}(t)\}_{i=1}^{N}$. More specifically, we have that

$$\mathbf{Y}_{(i)}(t) = [\mathbf{S}^{-1}\mathbf{e}_i]_0\bar{\mathbf{Y}}_1(0) + \sum_{k=2}^{N}[\mathbf{S}^{-1}\mathbf{e}_i]_k\Theta\left(t^{-\beta}\right)$$

where $[\mathbf{S}^{-1}\mathbf{e}_i]_k$ is the $k$-component of matrix $\mathbf{S}^{-1}\mathbf{e}_i$. We can prove that the first row of $\mathbf{S}^{-1}$ is $a\mathbf{1}^\top$ with $a$ being a scalar and $\mathbf{1}$ is an all-ones vector (it is based on Horn & Johnson (2012)[Theorem 3.2.5.2.], see Lemma 1). It follows that $[\mathbf{S}^{-1}\mathbf{e}_i]_0$ is the same for all $i$. We therefore have that for some $i$ and $j$

$$\left\|\mathbf{x}^{(i)}(t) - \mathbf{x}^{(j)}(t)\right\|_2^2 = \left\|\mathbf{S}\mathbf{Y}_{(i)}(t) - \mathbf{S}\mathbf{Y}_{(j)}(t)\right\|^2 = \Theta\left(t^{-2\beta}\right)$$

The proof is now complete.

**Lemma 1.** *The first row of $\mathbf{S}^{-1}$ is $a\mathbf{1}^\top$ with $a$ being a scalar and $\mathbf{1}$ is an all-ones vector.*

*Proof.* The Jordan canonical form of $\mathbf{W}\mathbf{D}^{-1}$ is represented as $\mathbf{S}\bar{\mathbf{J}}\mathbf{S}^{-1}$ where $\bar{\mathbf{J}} = \mathbf{J} + \mathbf{I}$ with the first Jordan block being 1 and the rest having eigenvalues *strictly less than* 1. Based on (Horn & Johnson, 2012)[Theorem 3.2.5.2.], we observe that $\lim_{k\to\infty}(\mathbf{W}\mathbf{D}^{-1})^k = \lim_{k\to\infty}\mathbf{S}\bar{\mathbf{J}}^k\mathbf{S}^{-1} = \mathbf{S}\mathbf{\Lambda}\mathbf{S}^{-1}$, where $\mathbf{\Lambda}$ is a diagonal matrix with the first element as 1 and all the others as 0:

$$\mathbf{\Lambda} = \begin{pmatrix} 1 & & & \\ & 0 & & \\ & & \ddots & \\ & & & 0 \end{pmatrix}$$

Since $\lim_{k\to\infty}(\mathbf{W}\mathbf{D}^{-1})^k$ maintains its column stochasticity and the rank of $\mathbf{S}\mathbf{\Lambda}\mathbf{S}^{-1}$ is 1, we deduce that the first row of $\mathbf{S}^{-1}$ is $a\mathbf{1}^\top$ with $a$ being a scalar and $\mathbf{1}$ an all-ones vector. $\square$

## LIMITATIONS

Our research proposes an advanced graph diffusion framework that integrates *time-fractional derivatives*, effectively encompassing many GNNs. Nonetheless, it presents certain limitations. A crucial element we have overlooked is the application of the *fractional derivative in the spatial domain*. In fractional diffusion equations, this implies substituting the standard second-order spatial derivative with a Riesz-Feller derivative (Gorenflo & Mainardi, 2003), thus modeling a random walk with space-based long-range jumps. Incorporating such a space-fractional diffusion equation within GNNs could potentially alleviate issues like the bottleneck and over-squashing highlighted in (Alon & Yahav, 2021). This represents a current limitation of our work and suggests a compelling future research trajectory that merges both time and space fractional derivatives in GNNs.

## BROADER IMPACT

The introduction of FROND holds significant potential for applications such as sensor networks, transportation, and manufacturing. FROND's ability to encapsulate long-term memory in neural dynamical processes can enhance the representation of complex interconnections, improving predictive modeling and efficiency. This could lead to more responsive sensor networks, optimized routing in transportation, and improved visibility into manufacturing process networks. However, the advent of FROND and similar models may also have mixed labor implications. While these technologies might render certain repetitive tasks obsolete, potentially displacing jobs, they may also generate new opportunities focused on developing and maintaining such advanced systems. Moreover, the shift from mundane tasks could enable workers to focus more on strategic and creative roles, enhancing job satisfaction and productivity. It's paramount that the deployment of FROND is done ethically, with ample support for reskilling those whose roles may be affected. This helps ensure that the broader impact of this technology is beneficial to society as a whole.