# OpenReview forum: "Unleashing the Potential of Fractional Calculus in Graph Neural Networks with FROND"
_ICLR.cc/2024/Conference — ICLR 2024 spotlight_

### Official Review · Reviewer_7EpT · 2023-10-31

**Soundness:** 3 good
**Presentation:** 3 good
**Contribution:** 3 good
**Rating:** 8
**Confidence:** 3

**Summary:**

The paper introduces the Fractional-Order graph Neural Dynamical network (FROND), a novel learning framework that enhances traditional graph neural ordinary differential equation (ODE) models by integrating the time-fractional Caputo derivative. This incorporation allows FROND to capture long-term memories in feature updating due to the non-local nature of fractional calculus, addressing the limitation of Markovian updates in existing graph neural ODE models and promising improved graph representation learning.

**Strengths:**

1. The paper is well-written, providing a clear and straightforward presentation of the content, which enhances the overall readability.

2. The innovative integration of time-fractional derivatives into traditional graph ODEs is a novel approach that effectively addresses key issues like non-local interactions and over-smoothing.

3. The proposal is supported by theoretical motivations.

4. An extensive evaluation of the framework is presented, demonstrating its effectiveness and versatility across various settings and providing substantial empirical evidence of its performance.

**Weaknesses:**

The correlation between beta and fractal dimemsion is not clear.  For instance, despite Pubmed having a higher fractal dimension of 2.25 compared to Airport, the optimal beta for it is set at 0.9. This observation raises curiosity about the specific conditions or types of datasets under which FROND demonstrates significant performance improvements. Clarification on this matter would greatly enhance the reader’s understanding and application of FROND in various contexts.

To further highlight the strengths of FROND and to provide clearer guidance on its optimal application scenarios, I would recommend conducting additional evaluations on datasets that necessitate long-range interactions[1].

The content in section 3.3 offers valuable insights, and I believe it could be enriched with additional technical details and formulations related to the graph layer. This enhancement would aid readers in developing a more comprehensive and profound understanding of the model.


Drawing a more explicit connection between fractal characteristics and FROND’s efficacy, particularly in handling tree-like data, would contribute to a more coherent narrative and justification for the framework. I kindly suggest expanding on this aspect.


[1]Dwivedi, Vijay Prakash, et al. "Long range graph benchmark." Advances in Neural Information Processing Systems 35 (2022): 22326-22340.

**Questions:**

What is the computational complexity of FROND? What is the T chosen for each experiment? and how is the short memory principle applied?

---

> ### Author Response · Authors · 2023-11-16
>
> Dear Reviewer 7EpT,
> we sincerely appreciate the time and effort you have dedicated to reviewing our work. Your insightful comments are invaluable to us. We are still in the process of carefully considering and addressing each of your points. Our detailed response will be ready and submitted within the next 1-2 days. We thank you for your understanding and patience during this period.

---

> ### Author Response · Authors · 2023-11-17
> **Weakness 1&4: Correlation between beta and fractal dimemsion**
>
> >Weakness 1: The correlation between beta and fractal dimemsion is not clear. For instance, despite Pubmed having a higher fractal dimension of 2.25 compared to Airport, the optimal beta for it is set at 0.9. This observation raises curiosity about the specific conditions or types of datasets under which FROND demonstrates significant performance improvements. Clarification on this matter would greatly enhance the reader’s understanding and application of FROND in various contexts.
>
> >Weakness 4: Drawing a more explicit connection between fractal characteristics and FROND’s efficacy, particularly in handling tree-like data, would contribute to a more coherent narrative and justification for the framework. I kindly suggest expanding on this aspect.
>
>
> **Response:**  It is crucial to highlight that the fractal dimension computation is solely based on the graph topology, similar the concept of $\delta$-hyperbolicity. Thus, it can only depict a trend between the fractal dimension and the optimal $\beta$ value derived from our experiments. For F-GRAND-l, the optimal $\beta$ is identified as 0.9, whereas for F-GRAND-nl, it is 0.4. This variation is notable even among the same model variants.  Developing a measure that integrates both feature similarity and graph topology remains a significant challenge. However, we propose that the optimal $\beta$ could act as an indicator of the dataset's fractal nature.
>
>
> Dynamical processes exhibiting self-similarity in fractal media are known to be more accurately described by Fractional Differential Equations (FDEs). For instance, in scenarios where heat or mass disperses across such structures, the concentration is best modeled using fractional diffusion equations.
> _In existing literature, the relationship between fractional order and the fractal properties of media has been extensively studied._ Theoretical analyses have been conducted under rigorous assumptions of the media [F1, F2, F3], encompassing structures like comb- and brush-like graphs [F4] and fractal meshes [F5]. Additionally, empirical investigations, such as those presented in [F6], have further explored this relationship.
> _They have shown that decreasing the fractal dimension leads to increasing the anomalous diffusion exponent and equivalently the fractional order $\beta$._ The fractal dimension effectively quantifies the complexity or the degree of 'brokenness' of a fractal structure, essentially measuring how pattern details—specifically, the number of self-similar pieces—alter with the scale of observation. Conversely, the fractional order in differential equations introduces a 'memory' or 'history dependence' within the system. It is hypothesized that a larger fractal dimension, indicative of a more complex structure, might be associated with a smaller fractional order in the differential equations modeling diffusion processes. This is because a higher fractal dimension suggests increased complexity and more obstacles in the medium, potentially slowing down processes like diffusion. Consequently, a smaller fractional order could more accurately represent the slower, more intricate diffusion process in such a medium. _The trend observed between the fractal dimension and the optimal $\beta$ value from our experiments aligns well with the anomalous diffusion model described in the aforementioned literature._
>
>
> [F1] Nigmatullin, R. R. "The realization of the generalized transfer equation in a medium with fractal geometry." Physica status solidi (b) 133.1 (1986): 425-430.
>
> [F2] Butera, Salvatore, and Mario Di Paola. "A physically based connection between fractional calculus and fractal geometry." Annals of Physics 350 (2014): 146-158.
>
> [F3] A. Iomin, Subdiffusion on a fractal comb, Phys. Rev. E - Stat. Nonlinear, Soft Matter Phys. 83 (2011) 52106. doi:10.1103/PhysRevE.83.052106.
>
> [F4] Plyukhin, Alex V., and Dan Plyukhin. "Random walks on uniform and non-uniform combs and brushes." Journal of Statistical Mechanics: Theory and Experiment 2017.7 (2017): 073204.
>
> [F5] Sandev, Trifce, Alexander Iomin, and Holger Kantz. "Anomalous diffusion on a fractal mesh." Physical Review E 95.5 (2017): 052107.
>
> [F6] Zhokh, Alexey, Andrey Trypolskyi, and Peter Strizhak. "Relationship between the anomalous diffusion and the fractal dimension of the environment." Chemical Physics 503 (2018): 71-76.

---

> ### Author Response · Authors · 2023-11-17
> **Weakness 2: More experiments on long-range interaction datasets**
>
> >Weakness 2: To further highlight the strengths of FROND and to provide clearer guidance on its optimal application scenarios, I would recommend conducting additional evaluations on datasets that necessitate long-range interactions[1].
>
> **Response:** Thank you for your suggestion. While over-squashing falls outside the purview of our current study, it is an important aspect worth considering in future work. The FROND framework is designed to integrate seamlessly with various graph neural ODE models. In our experiments using the dataset from [1], as detailed in **Tab. R2**, we observe that **FROND enhances the base GRAND model's performance by 5\% in graph classification tasks.** We plan to delve into the over-squashing issue in GNNs in our subsequent research endeavors.
>
>
>
> | Model | GCN | GCNII | GINE | GatedGCN | GatedGCN+RWSE | GRAND-l | F-GRAND-l |
> |-------|-----|-------|------|----------|---------------|---------|-----------|
> | Test AP | 0.5930±0.0023 | 0.5543±0.0078 | 0.5498±0.0079 | 0.5864±0.0077 | 0.6069±0.0035 | 0.5774±0.0063 | **0.6253±0.0015** |
>
>
>
>
>
> **Table: R2.** Graph classification on Peptides-func dataset. Performance metric is Average Precision (AP).

---

> > ### Comment · Reviewer_7EpT · 2023-11-21
> >
> > It seems there's been a misunderstanding. My suggestion is intended to enhance Section 4.3, focusing on strengthening it rather than addressing over-squashing issues. It would be beneficial to demonstrate how oversmoothing mitigation effectively works on datasets where depth is advantageous.

---

> ### Author Response · Authors · 2023-11-17
> **Weakness 3: More details for section 3.3.**
>
> >Weakness 3: The content in section 3.3 offers valuable insights, and I believe it could be enriched with additional technical details and formulations related to the graph layer. This enhancement would aid readers in developing a more comprehensive and profound understanding of the model.
>
> **Response:**  Due to space constraints, we have included the detailed discussion in the Appendix. We direct the reviewer to Appendix Section C of our paper, where we also offer illustrative diagrams in Figures 2 and 3. These figures draw parallels between the solvers and GNN layers, aiding in understanding.

---

> > ### Comment · Reviewer_7EpT · 2023-11-21
> >
> > During my initial review, I read Appendix C and found it both useful and relevant for the main paper. Therefore, I recommend integrating some of its content into Section 3.3 for enhanced clarity.

---

> ### Author Response · Authors · 2023-11-17
> **Question: Computation cost**
>
> > Question: What is the computational complexity of FROND? What is the T chosen for each experiment? and how is the short memory principle applied?
>
>
> **Response:** Thank you for your attention to this matter. We concur with the significance of examining model computation. In our initial submission, we have comprehensively addressed both the computational costs in Appendix Sections D.4, D.5, and D.6.
> The time complexity for the graph ODE framework is $\mathcal{O}(C E)$, with $C$ as the complexity for evaluating $\mathcal{F}(\mathbf{W}, \mathbf{X}(t))$, and $E=T / h$ for discretization steps. In contrast, FROND has a higher time complexity of $\mathcal{O}(C E \log (E))$ with fast convolution or $\mathcal{O}(C E^2)$ using direct calculation.
> To effectively address the computation cost, we have introduced the Short Memory solver in Section C.3 of our paper. This solver leverages a shifting memory window with a fixed width $K$, a strategy discussed in detail in Section D.5.1. For additional complexity analysis and tests, we kindly refer the reviewer to these specific sections in our Appendix.

---

> ### Comment · Area_Chair_n84J · 2023-11-20
> **Respond to authors' rebuttal**
>
> Please, confirm that you have read the author's response and the other reviewers' comments and indicate if you are willing to revise your rating.

---

> > ### Comment · Reviewer_7EpT · 2023-11-21
> >
> > I have read the response, and will keep my score for now.

---

> ### Author Response · Authors · 2023-11-21
> **New responses to clarify the misunderstanding**
>
> We are very delighted to have received feedback from Reviewer 7EpT. We acknowledge that some misunderstandings occurred in our earlier responses. Consequently, we have made the following updates to parts of our responses:
>
>
> > It seems there's been a misunderstanding. My suggestion is intended to enhance Section 4.3, focusing on strengthening it rather than addressing over-squashing issues. It would be beneficial to demonstrate how oversmoothing mitigation effectively works on datasets where depth is advantageous.
>
>
>
> We apologize for the previous misunderstanding. To illustrate the effectiveness of our oversmoothing mitigation, especially in contexts where depth is beneficial, we conducted additional experiments on the Peptides-func dataset. This dataset is recognized for its dependence on depth to facilitate interactions between long-range nodes.
> In our experiments, we exclusively focused on the impact of depth, avoiding any techniques like rewiring, to isolate its effect.
>
> From the preliminary results in Table R5, F-GRAND significantly enhances GRAND's performance on this dataset. More specifically, we observe that when we have 5 layers, GRAND and its fractional counterpart F-GRAND both have similar performances. As more layers are added, GRAND’s performance remains stable but begins to decline when exceeding 10 layers. In contrast, F-GRAND demonstrates improved performance with an increased number of layers. We will include more experiments on additional datasets in the camera-ready revision.
>
> We also direct the reviewer to our oversmoothness experiment detailed in Table R1 (reproduced in the following Table R6). In the Airport datasets, we note that FLODE achieves optimal performance at around 8 layers. In contrast, F-FLODE, with additional layers, shows increased performance, peaking at 80 layers.
>
> We hope these clarifications adequately demonstrate how mitigating oversmoothness can indeed bolster model performance.
>
>
> | Depth | 5 | 10 | 15 |
> |-------|---|----|----|
> | GRAND | **57.35±0.08** | 57.32±0.33 | 55.74±0.63 |
> | F-GRAND | 58.46±0.45 | 62.41±0.36 | **62.53±0.15** |
>
> **Table R5:** Graph classification on Peptides-func dataset. Performance metric is Average Precision (AP) (%)
>
>
>
>
>
>
> |       | Depth | 4 | 8 | 16 | 32 | 64 | 80 | 128 | 256 |
> |--------|-------|---|---|----|----|----|----|-----|-----|
> | FLODE  |       | 80.90±1.64 | **82.79±1.24** | 82.45±1.41 | 76.95±9.96 | 69.88±9.88 | 65.03±9.46 | 62.42±10.43 | 56.93±6.98 |
> | F-FLODE|       | 83.83±1.46 | 85.95±0.87 | 86.47±1.50 | 87.90±1.46 | 88.12±1.28 | **89.08±1.50** | 88.94±1.11 | 88.74±1.57 |
>
>
> **Table R6:** Node classification on Airport dataset
>
>
>
>
> > During my initial review, I read Appendix C and found it both useful and relevant for the main paper. Therefore, I recommend integrating some of its content into Section 3.3 for enhanced clarity.
>
>
> We appreciate the reviewer's suggestions and have accordingly moved some content from Appendix C to Section 3.3. We believe this adjustment enhances the readability of our paper and invite the reviewer to examine the updated Section 3.3. Thank you for your valuable suggestion!

---

> > ### Comment · Reviewer_7EpT · 2023-11-22
> >
> > Thank the authors for the reponses. I will increase my score.

---

### Official Review · Reviewer_1hAW · 2023-11-01

**Soundness:** 3 good
**Presentation:** 3 good
**Contribution:** 3 good
**Rating:** 8
**Confidence:** 3

**Summary:**

The paper extends the graph neural ODE framework by allowing fractional (Caputo) derivatives in the time variable. By leveraging on global information in the fractional derivatives, the authors prove a slow mixing theorem that prevents oversmoothing of node features. Experimentally, the authors demonstrate this fractional calculus framework over different graph ODE models and show they achieve good performances.

**Strengths:**

Novelty: To the best of my knowledge, this is the first approach to directly generalize graph neural ODE to fractional derivatives and demonstrate its applicability in real-world datasets.

Flexibility: The framework is general enough to be incorporated to a wide range of existing graph neural ODE in the literature, such as GRAND, GRAND++, GREAD, etc.

Experiments: Empirical study conducted is extensive and results are explained comprehensively. Showing competitiveness of the new framework over existing ones in many different dimensions.

**Weaknesses:**

I have not studied the appendix closely so it is possible that some of these questions are addressed there.

It is not immediately clear how this current approach quantitatively/qualitatively compares to existing approaches that exploit long-range memory in the modeling process, for instance Maskey et al. 2023 (see question 1).

Due to the long-range memory information, I believe it is expected that this approach is more computationally heavy than traditional neural graph ODE. It would be more complete to have a discussion of this increased cost, if there are any, as well as techniques used to overcome it. This is crucial in scaling the approach to larger datasets.

I am very willing to raise my score if these issues are addressed sufficiently.

**Questions:**

Is it possible to get a clearer distinction between the approach of this work, which is modeling node feature evolution through the layers as a FDE, versus Maskey et al. 2023, which proposes using fractional graph Laplacian in the usual ODE framework. It appears that Maskey et al. 2023 approach also tackles oversmoothing via long-range dependency of the dynamics, which is the main theoretical justification of the current work as well. More specifically, are there simple examples in which one framework strictly encapsulates another? Most importantly, what is the advantage of using this framework over Maskey et al. 2023 framework?

I understand fast mixing of graph random walk results of Chung 1997 and its dependence on various factors, such as eigenvalues of the adjacency/Laplacian. However, it is not immediately clear to me that the same fast rate carries over to graph neural ODE (which has some kind of skip connection across depths). Can this be explained more thoroughly?

Is it possible to give a proof sketch/intuition of Theorem 2, in particular, why should we expect the slow algebraic rate? It is also interesting that the rate is tight. Following the proof in the appendix seems to suggest that this come from a deeper result by Mainardi but there is not a lot of intuition there.

Minor issues:
M1: F(s) in equation (1) is not defined (I assume it is the Laplace transform of f). The variable s is also not defined (I assume it is the variable of the transformed function).
M2: There should also be conditions under which the Laplace transform exists (and the Laplace transform of the derivative)

---
After the rebuttal phase, the authors have addressed my concerns and I am raising my score to an 8.

---

> ### Author Response · Authors · 2023-11-16
> **Weakness 1 & Question 1: Compared to long-range interaction works**
>
> >Weakness 1: It is not immediately .... in the modeling process, for instance Maskey et al. 2023 (see question 1).
>
> >Question 1: Is it possible to get a clearer distinction between the approach of this work....what is the advantage of using this framework over Maskey et al. 2023 framework?
>
>
> **Response:** In our initial submission, we briefly discussed the differences between our FROND model and FLODE (Maskey et al. 2023) in the second paragraph on page 2 and the "Graph Neural ODE Models" section of the Related Work on page 17. In our revised version, we have included FLODE as a baseline and expanded the discussion in Section E.6 of the appendix. Fundamentally, FLODE differs from our work in the following key aspects:
>
> - FLODE employs the fractional (real-valued) power of $\mathbf{L}$, namely $\mathbf{L}^\alpha$.
> The feature evolution model used by FLODE is given by:
> \begin{align*}
> \frac{\mathrm{d}\mathbf{X}(t)}{\mathrm{d}t} = -\mathbf{L}^{\alpha}\mathbf{X}(t)\mathbf{\Phi}.  \quad  \quad  \quad   \quad \quad \text{[FLODE]}
> \end{align*}
> This is a graph spatial domain rewiring technique, as $\mathbf{L}^\alpha$ introduces dense connections compared to $\mathbf{L}$. As a result, FLODE introduces space-based long-range interactions during the feature updating process.
>
> - In contrast, our FROND model incorporates the time-fractional derivative $D_t^\beta$ to update graph node features in a memory-inclusive dynamical process. In this context, time acts as a continuous counterpart to the layer index, leading to significant dense skip connections between layers due to memory dependence. Thus, FROND induces time/layer-based long-range interactions in the feature update process. Note that FLODE does not utilize time-fractional derivatives.
> Our method is not only __compatible with various graph ODE modes, including FLODE (see F-FLODE below),__ but also extends them to graph fractional differential equation (FDE) models.
>
> In Section E.6, we introduce a new FROND model based on FLODE, called the F-FLODE model, which utilizes time-fractional derivatives for updating graph node features in FLODE:
> \begin{align*}
> D_t^\beta \mathbf{X}(t)=-\mathbf{L}^\alpha \mathbf{X}(t) \mathbf{\Phi},   \quad  \quad  \quad   \quad \quad \text{[F-FLODE]}
> \end{align*}
> Note the difference in the equations in [FLODE] and [F-FLODE], where the two are equivalent only when $\beta=1$.
> This example illustrates that the FROND framework encompasses the FLODE model as a special case when $\beta=1$. Our experimental results indicate that F-FLODE outperforms FLODE with the optimal $\beta\ne 1$ in general. For more detailed information, we refer the reviewer to the revised version of our paper.
>
>
> To better distinguish between FLODE and F-FLODE, we interpret them using Euler discretization, illustrating how skip/dense connections operate in each model and the flow of information across nodes and layers.
>
> In the FLODE model, as detailed in the paper, the feature update rule is represented by the explicit Euler scheme:
> \begin{align*}
> \mathbf{X}(t_{n})=\mathbf{X}(t_{n-1})-h \mathbf{L}^\alpha \mathbf{X}(t_{n-1}) \mathbf{\Phi},
> \end{align*}
> Here, the dense connections between nodes created by $\mathbf{L}^\alpha$ and the direct (i.e., memory-less) connection to the preceding layer $\mathbf{X}(t_{n-1})$ enable layer $n$ to assimilate information from distantly connected nodes in layer $n-1$.
>
> In contrast, in the F-FLODE model, following the fractional Euler scheme in Eq. (33), the feature update rule is:
> \begin{align*}
> \mathbf{X}\left(t_n\right)=\mathbf{X}(0)-\frac{1}{\Gamma(\beta)} \sum_{j=0}^{n-1} \mu_{j, n}\mathbf{L}^{\alpha}\mathbf{X}(t_j)\mathbf{\Phi},
> \end{align*}
> with $\mu_{j, n}=\frac{h^\beta}{\beta}\left((n-j)^\beta-(n-1-j)^\beta\right)$ and $h=t_n-t_{n-1}$ as the temporal step size.
> This model, through its dense connection between nodes by $\mathbf{L}^\alpha$ and historical links to all previous layers (${\mathbf{X}(0),\ldots,\mathbf{X}(t_{n-1})}$), facilitates layer $n$ in leveraging information from distant nodes across layers $0$ to $n-1$.
> Notably, when $\alpha=1$, FLODE reduces to local aggregation, with information exchange restricted to immediate neighbors from the preceding layer $\mathbf{X}(t_{n-1})$.
> However, in such cases with $\beta<1$, historical connections to previous layers in F-FLODE are maintained. Even though at layer $n$ both F-FLODE and FLODE have information exchange within the nodes' $n$-hop neighbors (where the shortest distance $\leq n$ ), F-FLODE enables more explicit information exchange since the past states have been stored in the previous layers. To see this, note that $\mathbf{X}\left(t_k\right)$ are the node states after the $k$-hop local aggregation. By using F-FLODE, we have more direct information exchange within the nodes' all $n$-hop neighbors, more than within the nodes with exactly distance $n$.
> For a visual representation of information flow in FROND, we direct the reviewer to Fig. 3 in our paper.

---

> ### Author Response · Authors · 2023-11-16
> **Weakness 2: Computation cost (had included in our initial submission Appendix)**
>
> > Weakness 2: Due to the long-range memory information, I believe it is expected that this approach is more computationally heavy than traditional neural graph ODE. It would be more complete to have a discussion of this increased cost, if there are any, as well as techniques used to overcome it. This is crucial in scaling the approach to larger datasets.
>
>
> **Response:**
> We refer the reviewer to the Appendix Sections D.4, D.5, and D.6, where we address both the computational costs and performance of FROND on the large Ogbn dataset.
> The time complexity for the graph ODE framework is $\mathcal{O}(C E)$, with $C$ as the complexity for evaluating $\mathcal{F}(\mathbf{W}, \mathbf{X}(t))$, and $E=T / h$ for discretization steps. In contrast, FROND has a higher time complexity of $\mathcal{O}(C E \log (E))$ with fast convolution or $\mathcal{O}(C E^2)$ using direct calculation.
> **To effectively mitigate the higher computation cost, we have introduced the Short Memory solver in Section C.3 of our paper.** This solver leverages a shifting memory window with a fixed width $K$, a strategy discussed in detail in Section D.5.1. For additional complexity analysis and tests, we kindly refer the reviewer to these specific sections in our Appendix.

---

> ### Author Response · Authors · 2023-11-16
> **Question 2: Why fast convergence rate of graph neural ODE model**
>
> >Question 2: I understand fast mixing of graph random walk results of Chung 1997 and its dependence on various factors, such as eigenvalues of the adjacency/Laplacian. However, it is not immediately clear to me that the same fast rate carries over to graph neural ODE (which has some kind of skip connection across depths). Can this be explained more thoroughly?
>
>
> **Response:** The speed of convergence for a memoryless random walk on a graph to the stationary distribution is closely related to the eigenvalues of the graph's adjacency or Laplacian matrix. Specifically, the second-largest eigenvalue (in absolute value) of the transition matrix (derived from the adjacency or Laplacian matrix) plays a crucial role.
> The rate of convergence increases as this eigenvalue decreases, in an exponential manner. Importantly, changes in this eigenvalue only modify the exponential convergence rate's exponent. Regardless of these changes, the convergence remains exponentially swift.
>
> In the context of continuous analogs to discrete random walks, the graph neural diffusion process, as demonstrated in GRAND++ and our research, serves as a pertinent example. In discrete solvers for graph neural ODE models, the implementation of skip connections is typically restricted, linking only to the immediately preceding layer and not altering the convergence rate. However, in our approach, the skip connections extend even to the initial layers, fundamentally altering the diffusion rate.
>
> From an analytical standpoint, examining scalar cases can be enlightening. Consider the Lyapunov asymptotically stable ODE:
> \begin{align*}
>   \frac{\mathrm{d} y(t)}{\mathrm{d} t}=-\lambda y(t)
> \end{align*}
> Here, $\lambda$ is a positive constant. The solution to this equation is $y(t)=\exp (-\lambda t)$, which demonstrates an exponential convergence to zero. Further insights are offered in our response to Question 3, where we discuss the emergence of a slow algebraic rate in our solution.

---

> ### Author Response · Authors · 2023-11-16
> **Question 3: More intuition about the algebraic convergence**
>
> >Question 3: Is it possible to give a proof sketch/intuition of Theorem 2, in particular, why should we expect the slow algebraic rate? It is also interesting that the rate is tight. Following the proof in the appendix seems to suggest that this come from a deeper result by Mainardi but there is not a lot of intuition there.
>
>
> **Response:**
>
> Thank you for your insightful query regarding the intuition behind the slow algebraic convergence rate in Theorem 2. To provide a clearer understanding, we approach this from three perspectives.
>
> - Fractional Random Walk Perspective: In a standard random walk, a walker moves to a new position at each time step without delay. However, in a fractional random walk, which is more reflective of our model's behavior, the walker has a probability of revisiting past positions. This revisitation is not arbitrary; it is governed by a waiting time that follows a power-law distribution with a long tail. This characteristic fundamentally changes the walk's dynamics, introducing a memory component and leading to a slower, algebraic rate of convergence. This behavior is intrinsically different from normal random walks, where the absence of waiting times facilitates a quicker exponential convergence.
>
> - Information Interaction Perspective: From the response to Weakness 1 , we clearly observe that all the previous layers $\left(\mathbf{X}(0), \ldots, \mathbf{X}\left(t_{n-1}\right)\right.$ ) have a direct impact on $\mathbf{X}\left(t_n\right)$. This means the initial features are less forgettable than only a skip connection to the preceding layer $\mathbf{X}\left(t_{n-1}\right)$. The asymptotic convergence rate with dense layer connections is therefore expected to be lower than the convergence rate with only skip layer connections.
>
> - Analytic Perspective: From an analytic perspective, the essential slow algebraic rate primarily stems from the slow convergence of the Mittag-Leffler function towards zero.
> To elucidate this, let's consider the scalar scenario. Recall that the Mittag-Leffler function $E_\beta$ is defined as:
> \begin{align*}
> E_\beta(z):=\sum_{j=0}^{\infty} \frac{z^j}{\Gamma(j \beta+1)}
> \end{align*}
> for values of $z$ where the series converges. Specifically, when $\beta=1$,
> \begin{align*}
> E_1(z)=\sum_{j=0}^{\infty} \frac{z^j}{\Gamma(j+1)}=\sum_{j=0}^{\infty} \frac{z^j}{j !}=\exp (z)
> \end{align*}
> corresponds to the well-known exponential function. According to [A1, Theorem 4.3.], the eigenfunctions of the Caputo derivative are expressed through the Mittag-Leffler function. In more precise terms, if we define $y(t)$ as
> \begin{align*}
> y(t):=E_\beta\left(-\lambda t^n\right), \quad t \geq 0,
> \end{align*}
> it follows that
> \begin{align*}
> D_t^\beta y(t)=-\lambda y(t)
> \end{align*}
> Notably, when $\beta=1$, this reduces to $\frac{\mathrm{d} \exp (-\lambda t)}{\mathrm{d} t}=-\lambda \exp (-\lambda t)$.
>
> Further, we examine the behavior of $E_\beta\left(-\lambda t^n\right)$. As per [A1, Theorem 7.3.], it is noted that:
>
> (a) The function $y(t)$ is completely monotonic on $(0, \infty)$.
>
> (b) As $x \rightarrow \infty$,
> \begin{align*}
> y(t)=\frac{t^{-\beta}}{\lambda \Gamma(1-\beta)}(1+o(1)) .
> \end{align*}
>
> Thus, the function $E_\beta\left(-\lambda t^\beta\right)$ converges to zero at a rate of $\Theta\left(t^{-\beta}\right)$. Our paper extends this to the general high-dimensional case by replacing the scalar $\lambda$ with the Laplacian matrix $\mathbf{L}$, wherein the eigenvalues of $\mathbf{L}$ play a critical role analogous to $\lambda$ in the scalar case.
>
> For a diagonalizable Laplacian matrix $\mathbf{L}$, the proof essentially reverts to the scalar case as outlined above (refer to Eq. (54) in our paper). However, in scenarios where $\mathbf{L}$ is non-diagonalizable and has a general Jordan normal form, it becomes necessary to employ the Laplace transform to demonstrate that the algebraic rate remains valid ((refer to the context between Eq. (55) and (56) in our paper)
>
> We have included the above discussion to in our revised version.
>
> [A1] Kai Diethelm. The analysis of fractional differential equations: an application-oriented exposition using differential operators of Caputo type, volume 2004. Springer, 2010.

---

> ### Author Response · Authors · 2023-11-16
> **Question 4: Minor issues**
>
> >Typos: Minor issues:
> M1: F(s) in equation (1) is not defined (I assume it is the Laplace transform of f). The variable s is also not defined (I assume it is the variable of the transformed function).\\
> M2: There should also be conditions under which the Laplace transform exists (and the Laplace transform of the derivative)}
>
> **Response:**  Thank you for pointing out these issues. We have made the necessary corrections in our revised version.

---

> ### Author Response · Authors · 2023-11-16
> **Weakness 1 & Question 1: Compared to long-range interaction works (Cont'd)**
>
> Additionally, to show more advantages, we conduct experiments to compare oversmoothness mitigation in **Tab R1**, observing F-FLODE's superior performance over FLODE with many layers. Furthermore, our experimental results in Tabs 23 and 24 in the paper indicate that F-FLODE outperforms FLODE with the optimal $\beta\ne1$ in general. For more detailed information, we refer the reviewer to the revised version of our paper.
>
> | Dataset | Model | 4 | 8 | 16 | 32 | 64 | 80 | 128 | 256 |
> |---------|-------|---|---|----|----|----|----|-----|-----|
> | Cora | GCN | 81.35±1.27 | 15.3±3.63 | 19.70±7.06 | 21.86±6.09 | 13.0±0.0 | 13.0±0.0 | 13.0±0.0 | 13.0±0.0 |
> | | GAT | 80.95±2.28 | 31.90±0.0 | 31.90±0.0 | 31.90±0.0 | 31.90±0.0 | 31.90±0.0 | 31.90±0.0 | 31.90±0.0 |
> | | GRAND-l | 81.29±0.43 | 82.95±0.52 | 82.48±0.46 | 81.72±0.35 | 81.33±0.22 | 81.07±0.44 | 80.09±0.43 | 73.37±0.59 |
> | | F-GRAND-l | 81.17±0.75 | 82.68±0.64 | 83.05±0.81 | 82.90±0.81 | 83.44±0.91 | 82.85±0.89 | 82.34±0.83 | 81.74±0.53 |
> | | FLODE | 67.85±6.01 | 64.49±7.11 | 62.51±9.03 | 56.40±12.15 | 46.80±14.08 | 42.25±13.54 | 41.54±12.72 | 39.01±9.69 |
> | | F-FLODE | 67.82±5.37 | 64.36±9.45 | 63.49±9.25 | 64.05±9.65 | 60.60±4.95 | 60.50±6.77 | 60.73±5.21 | 61.40±9.98 |
> | Airport | GCN | 84.77±1.45 | 74.43±8.19 | 62.56±2.16 | 15.27±0.0 | 15.27±0.0 | 15.27±0.0 | 15.27±0.0 | 15.27±0.0 |
> | | GAT | 83.59±1.51 | 67.02±4.70 | 46.56±0.0 | 46.56±0.0 | 46.56±0.0 | 46.56±0.0 | 46.56±0.0 | 46.56±0.0 |
> | | GRAND-l | 80.53±9.59 | 79.88±9.67 | 76.24±3.80 | 68.67±4.02 | 62.28±10.83 | 50.38±2.98 | 57.96±11.63 | 53.0±14.85 |
> | | F-GRAND-l | 97.0±0.79 | 97.09±0.87 | 96.97±0.84 | 96.50±0.60 | 97.41±0.42 | 96.53±0.74 | 97.03±0.55 | 94.91±3.72 |
> | | FLODE | 80.90±1.64 | 82.79±1.24 | 82.45±1.41 | 76.95±9.96 | 69.88±9.88 | 65.03±9.46 | 62.42±10.43 | 56.93±6.98 |
> | | F-FLODE | 83.83±1.46 | 85.95±0.87 | 86.47±1.50 | 87.90±1.46 | 88.12±1.28 | 89.08±1.50 | 88.94±1.11 | 88.74±1.57 |
>
>
>
> **Table: R1.** Over-smoothing mitigation under fixed data splitting without LCC

---

> ### Author Response · Authors · 2023-11-18
> **Thank you for your invaluable support!**
>
> > I am very willing to raise my score if these issues are addressed sufficiently.
>
> Dear Reviewer 1hAW,
>
> We are profoundly grateful for the time and expertise you dedicated to reviewing our paper. Your insightful feedback has been instrumental in guiding us to improve our work. We deeply value your support and guidance in this process.
>
> In light of the changes we have made in response to your comments, we would be immensely appreciative if you could take a moment to review our revisions. Your confirmation on whether our responses have adequately addressed your concerns would be greatly valued.
>
> Thank you once again for your invaluable contribution and for sharing your expertise with us!
>
> With sincere gratitude,
>
> Authors

---

> > ### Comment · Reviewer_1hAW · 2023-11-21
> >
> > I'd like to thank the authors for their thorough comments and am raising my recommendation to a 8.

---

> > > ### Author Response · Authors · 2023-11-22
> > >
> > > Thank you for your continued support of our paper! We sincerely appreciate your valuable insights, constructive feedback, and the time and effort you've invested in reviewing our work. Our discussions have been essential in improving the overall quality of the paper, for instance, by making the theoretical aspects more reader-friendly.

---

> ### Comment · Area_Chair_n84J · 2023-11-20
> **Respond to authors' rebuttal**
>
> Please, confirm that you have read the author's response and the other reviewers' comments and indicate if you are willing to revise your rating.

---

### Official Review · Reviewer_Qi4C · 2023-11-01

**Soundness:** 2 fair
**Presentation:** 3 good
**Contribution:** 2 fair
**Rating:** 6
**Confidence:** 4

**Summary:**

FROND is a method that uses concepts from fractional calculus applied to GNNs.
The method is based on defining the Caputo derivative and a solver that integrates the ODE.
The authors provide ample theory and several experiments to show the benefit of adding the fractional derivative component to GNNs.

**Strengths:**

1. The paper is well written. It was easy to follow and understand.

2. The authors show how the proposed method can encapsulate existing models such as GRAND or GraphCON.

3. The experiments show that adding a fractional derivative is useful.

**Weaknesses:**

1. Missing neural ODE literature: 'Stable Architectures for Deep Neural Networks' and 'A Proposal on Machine Learning via Dynamical Systems'.

2. Missing graph ODE literature: recent papers like 'Anti-Symmetric DGN: a stable architecture for Deep Graph Networks' and 'Ordinary differential equations on graph networks'.

3. In one of the main contributions, it is said that "We provide an interpretation from the perspective of a non-Markovian graph random walk when the model feature-updating dynamics is inspired by the fractional heat diffusion process. Contrasting with the traditional Markovian random walk implicit in traditional graph neural diffusion models whose convergence to the stationary equilibrium is exponentially swift, we establish that in FROND, convergence follows an algebraic rate.". Why is it true? if $\beta=2$ then the process is not diffusive at all. Rather, it is oscillatory, as shown in GraphCON (Rusch et al.)

4. In section 2.3, the authors should also discuss FLODE ('A Fractional Graph Laplacian Approach to Oversmoothing') which is very similar to this work and also uses fractional calculus.

5. In section 3.1 the authors discuss the initial conditions of the ODE. It is not clear to me how do you initialize $\beta$ time steps. From the text I can infer that it is the same condition as the input features. Is that was was actually done? If so, does it make sense from an ODE perspective? Have the authors tried other initialization procedures?

6. The authors mention that here only $\beta$ is only considered between 0 and 1. I wonder why. How would your model behave theoretically and practically if it larger than 1?

7. I am not sure it is correct that the model can have global or 'full path' properties if $\beta$ is smaller than 1. For example, I think it is fair to say that if $beta$ is indeed smaller than 1, then a second order process as in GraphCON cannot be represented by the model.

8. The experiments indeed show that the proposed method improves compared to baselines produced by the authors, but they are quite narrow and show a partial picture of the current state of the art and existing methods. I would expect that the authors compare their work (experimentally) with other methods like FLODE, CDE, GRAND++, as well as other recent methods like ACMII-GCN++ ('Is Heterophily A Real Nightmare For Graph Neural Networks To Do Node Classification?') or DRew ('DRew: Dynamically Rewired Message Passing with Delay').

9. The authors state that the proposed method can be applied to any ordinary differential equation GNN, so can the authors please also show the results when applied to other baseline methods as discussed in the paper?

10. $\beta$ is a hyperparameter. What would happen if you learn it? how will it influence the results and the stability of your solver? Is there a principles way to choosing the hyperparameter?

11. I am not certain that the method is novel, as it was also shown in 'Fractional Graph Convolutional Networks (FGCN) for Semi-Supervised Learning'.

12. Missing literature about skip connections in GNNs: 'Understanding convolution on graphs via energies'.

13. A general point - the focus of the paper is the mitigation of oversmoothing. But, also as the authors state, there are many methods that already do it. Then my question is what is the added value of using this mechanism ? Also, another important issue is oversquashing in graphs. Can the authors discuss how and if would the proposed method can help with that issue?

**Questions:**

I left questions in the review and I am looking forward to seeing the author's response.

---

> ### Author Response · Authors · 2023-11-16
> **Weakness 1, 2, 12: Missing literature**
>
> >Weakness 1: Missing neural ODE literature: 'Stable Architectures for Deep Neural Networks' and 'A Proposal on Machine Learning via Dynamical Systems'.
>
> >Weakness 2: Missing graph ODE literature: recent papers like ’Anti-Symmetric DGN: a stable architecture for Deep Graph Networks’ and ’Ordinary differential equations on graph networks’.
>
> >Weakness 12: Missing literature about skip connections in GNNs: ’Understanding convolution on graphs via energies’.
>
> **Response:** Thank you for your valuable literature suggestions. We have incorporated the additional related work you recommended into our revised manuscript. We invite the reviewer to examine the "Related Work Section", where these updates are highlighted in blue for ease of reference.
>
> We appreciate the opportunity to clarify our approach to the literature survey. In our initial manuscript submission, particularly on pages 16 to 17, we conducted a comprehensive exploration of the literature in the domain of graph neural ODEs and skip connections in GNNs. This survey encompasses pioneering studies dating back to 2018 and the most recent advancements, up to 2023, including references [R0-12] for (graph) neural ODEs and [R13-18] for skip connections in GNNs. The range of these citations, from foundational works to the latest research, underlines our commitment to providing a thorough and current literature survey.
>
> Additionally, in our updated manuscript, we have incorporated more references as recommended by you. We trust that these clarifications and additions demonstrate our rigorous approach to the literature review and our dedication to maintaining a comprehensive understanding of the field.
>
> [R0] Chen, Ricky TQ, et al. "Neural ordinary differential equations." In Advances Neural Inf. Process. Syst.2018.
>
> [R1] Avelar, Pedro HC, et al. "Discrete and continuous deep residual learning over graphs." arXiv preprint arXiv:1911.09554 (2019).
>
> [R2] Poli, Michael, et al. "Graph neural ordinary differential equations." arXiv preprint arXiv:1911.07532 (2019).
>
> [R3] Rusch, T. Konstantin, et al. "Graph-coupled oscillator networks", In Proc. Int. Conf. Mach. Learn., 2022.
>
> [R4] Chamberlain, Ben, et al. "Grand: Graph neural diffusion", In Proc. Int. Conf. Mach. Learn., pp. 1407–1418, 2021a.
>
> [R5] Chamberlain, Benjamin, et al. "Beltrami flow and neural diffusion on graphs", In Advances Neural Inf. Process. Syst., pp. 1594–1609, 2021b.
>
> [R6] Song, Yang, et al. "On the robustness of graph neural diffusion to topology perturbations", In Advances Neural Inf. Process. Syst., 2022.
>
> [R7] Bodnar, Cristian, et al. "Neural sheaf diffusion: A topological perspective on heterophily and oversmoothing in GNNs", In Advances Neural Inf. Process. Syst., 2022.
>
> [R8] Wang, Yuelin, et al. "ACMP: Allen-cahn message passing with attractive and repulsive forces for graph neural networks",  In Proc. Int. Conf. Learn. Representations, 2022b.
>
> [R9] Zhao, Kai, et al. "Graph neural convection-diffusion with heterophily", In Proc. Inter. Joint Conf. Artificial Intell., Macao, China, 2023.
>
>
> [R10] Thorpe, Matthew, et al. "GRAND++: Graph neural diffusion with a source term", In Proc. Int. Conf. Learn. Representations, 2022.
>
> [R11] Choi, Jeongwhan, et al. "GREAD: Graph Neural Reaction-Diffusion Networks", In Proc. Int. Conf. Mach. Learn., 2023.
>
> [R12] Maskey, Sohir, et al. "A Fractional Graph Laplacian Approach to Oversmoothing." arXiv preprint arXiv:2305.13084, 2023.
>
> [R13] Hongyang Gao and Shuiwang Ji. Graph u-nets. In Proc. Int. Conf. Mach. Learn., pp. 2083–2092, 2019.
>
> [R14] Xu, Keyulu, et al. "Representation learning on graphs with jumping knowledge networks", In Proc. Int. Conf. Mach. Learn., pp. 5453–5462, 2018.
>
> [R15] Chen, Ming, et al. "Simple and deep graph convolutional networks", In Proc. Int. Conf. Mach. Learn., pp. 1725–1735, 2020.
>
> [R16] Li, Guohao, et al. "Deepgcns: Can gcns go as deep as cnns?", In Proc. Int. Conf. Learn. Representations, pp. 9267–9276, 2019.
>
> [R17] Li, Guohao, et al. "Deepergcn: All you need to train deeper gcns." arXiv preprint arXiv:2006.07739, 2020a.
>
> [R18] Gutteridge, Benjamin, et al. "DRew: dynamically rewired message passing with delay", In Proc. Int. Conf. Mach. Learn., pp. 1225212267, 2023.

---

> ### Author Response · Authors · 2023-11-16
> **Weakness 3: Misunderstanding of our claim regarding random walk**
>
> >Weakness 3: In one of the main contributions, it is said that "We provide an interpretation from the perspective of a non-Markovian graph random walk when the model feature-updating dynamics is inspired by the fractional heat diffusion process. Contrasting with the traditional Markovian random walk implicit in traditional graph neural diffusion models whose convergence to the stationary equilibrium is exponentially swift, we establish that in FROND, convergence follows an algebraic rate.". Why is it true? if $\beta=2$ then the process is not diffusive at all. Rather, it is oscillatory, as shown in GraphCON (Rusch et al.)
>
> **Response:** We believe there has been a misunderstanding. Our assertion is that when the dynamic is **inspired by the fractional heat diffusion process**, this corresponds to the fractional GRAND model (F-GRAND) presented in Eq. (7) and (9) in our paper, where we clearly stated that $0<\beta \le 1$.
> Furthermore, the subsection title where we discuss this interpretation is Sec. 3.2 "Random Walk Perspective of F-GRAND-L", which clearly refers to the diffusive model.
>  This is different from the cases $\beta>1$ or $\beta=2$, which are oscillatory models similar to GraphCON. In the revision, we have emphasized this again in the contribution section, marked in blue color.

---

> ### Author Response · Authors · 2023-11-16
> **Weakness 4: More discussion of FLODE**
>
> >In section 2.3, the authors should also discuss FLODE ('A Fractional Graph Laplacian Approach to Oversmoothing') which is very similar to this work and also uses fractional calculus.
>
> **Response:** In our initial submission, we had addressed this point in the second paragraph on page 2 and in the "Graph Neural ODE Models" section of the Related Work on page 17. We kindly invite the reviewer to refer to these sections for detailed information. Additionally, in our revised version, we have included FLODE as a baseline and expanded the discussion in Section E.6 of the appendix. Fundamentally, FLODE differs from our work in the following key aspects:
>
> - FLODE employs the fractional (real-valued) power of $\mathbf{L}$, namely $\mathbf{L}^\alpha$.
> The feature evolution model used by FLODE, specifically in its first heat diffusion-type variant, is given by:
> \begin{align*}
> \frac{\mathrm{d}\mathbf{X}(t)}{\mathrm{d}t} = -\mathbf{L}^{\alpha}\mathbf{X}(t)\mathbf{\Phi}.  \quad  \quad  \quad   \quad \quad \text{[FLODE]}
> \end{align*}
> This is a graph spatial domain rewiring technique, as $\mathbf{L}^\alpha$ introduces dense connections compared to $\mathbf{L}$. As a result, FLODE introduces space-based long-range interactions during the feature updating process.
>
> - In contrast, our FROND model incorporates the time-fractional derivative $D_t^\beta$ to update graph node features in a memory-inclusive dynamical process. In this context, time acts as a continuous counterpart to the layer index, leading to significant dense skip connections between layers due to memory dependence. Thus, FROND induces time/layer-based long-range interactions in the feature update process. Note that FLODE does not utilize time-fractional derivatives.
> Our method is not only __compatible with various graph ODE modes, including FLODE (see F-FLODE below),__ but also extends them to graph fractional differential equation (FDE) models.
>
> In Section E.6, we introduce a new FROND model based on FLODE, called the F-FLODE model, which utilizes time-fractional derivatives for updating graph node features in FLODE:
> \begin{align*}
> D_t^\beta \mathbf{X}(t)=-\mathbf{L}^\alpha \mathbf{X}(t) \mathbf{\Phi},   \quad  \quad  \quad   \quad \quad \text{[F-FLODE]}
> \end{align*}
> Note the difference in the equations in [FLODE] and [F-FLODE], where the two are equivalent only when $\beta=1$.
> This example illustrates that the FROND framework encompasses the FLODE model as a special case when $\beta=1$. Our experimental results in Tabs 23 and 24 indicate that F-FLODE outperforms FLODE with the optimal $\beta\ne 1$ in general. For more detailed information, we refer the reviewer to the revised version of our paper.

---

> ### Author Response · Authors · 2023-11-16
> **Weakness 5: Discuss initial conditions**
>
> >Weakness 5: In section 3.1 the authors discuss the initial conditions of the ODE. It is not clear to me how do you initialize $\beta$ time steps. From the text I can infer that it is the same condition as the input features. Is that was was actually done? If so, does it make sense from an ODE perspective? Have the authors tried other initialization procedures?
>
> **Response:** FROND employs initial conditions analogous to those used in graph ODE models to allow it to be smoothly integrated with existing graph ODE models. Our selection of the Caputo fractional derivative is guided by this compatibility with the initial conditions of ODEs (cf. Sections B.3 and B.4.).
>
> - In typical first-order graph ODE models, preliminary node features $\mathbf{X}$, possibly processed through a learnable encoder, serve as the initial condition: $\mathbf{X}(0) = \mathbf{X}$.
>
> - The second-order graph ODE model GraphCON, described by the equation
> $$\mathbf{X}^{\prime \prime}=\sigma\left(\mathbf{F}_\\theta(\mathbf{X}, t)\right)-\gamma \mathbf{X}-\alpha \mathbf{X}^{\prime},$$
>
> utilizes the technique of introducing an auxiliary velocity variable, $\mathbf{Y}(t) = \mathbf{X}^{\prime}(t)$ to transform the second-order ODE into a first-order system (cf. Eq(3) in GraphCON). This is a common technique for handling higher-order ODEs:
> \begin{align*}
> \begin{aligned}
> & \mathbf{Y}^{\prime}=\sigma\left(\mathbf{F}_\theta(\mathbf{X}, t)\right)-\gamma \mathbf{X}-\alpha \mathbf{Y} \\\\
> & \mathbf{X}^{\prime}=\mathbf{Y}
> \end{aligned}
> \end{align*}
> Here, the initial condition is set as $\mathbf{X}'(0) = \mathbf{X}(0) = \mathbf{X}$ in this paper. _Thus, GraphCON can be effectively treated as a first-order ODE, with the initial feature dimension doubled by duplicating the initial features._
>
>
> - In our FROND framework, we adopt a similar approach for initial conditions when we have higher orders: $\mathbf{X}^{(\lceil\beta\rceil-1)}(0)=\ldots=\mathbf{X}(0)=\mathbf{X}$, paralleling the initial conditions in graph ODE models. Notably, for cases where $\beta>1$, akin to handling higher-order ODEs, we convert these into multiple lower-order FDEs [A1, Section 8], with a new $\beta$ value satisfying $0<\beta\le 1$.
>
>
> [A1] Kai Diethelm. The analysis of fractional differential equations: an application-oriented exposition using differential operators of Caputo type, volume 2004. Springer, 2010.

---

> > ### Comment · Reviewer_Qi4C · 2023-11-20
> > **Response**
> >
> > Thanks for the answer. I think that similarly to the discussion provided by the authors that GraphCON can be written as a first order model (similar to other higher order ODEs), it is also applicable to FROND. So I think that in this sense also FROND can be reduced to a first order model, making it similar to other methods like FLODE. Can the authors comment on that? Did the authors try to implement an evaluate an equivalent network ? is there an actual benefit to the proposed design this way?
> >
> > Also, while your response answers part of my question, it doesn't address different possible options to initialize the terms and if you experiment with it.

---

> ### Author Response · Authors · 2023-11-16
> **Weakness 6: Misunderstanding of order range**
>
> >Weakness 6:The authors mention that here only $\beta$ is only considered between 0 and 1. I wonder why. How would your model behave theoretically and practically if it larger than 1 ?
>
> **Response** In the general framework outlined in Equation (6), we do not assert that the parameter $\beta$ should be limited to the range between 0 and 1. This is also evident from our above detailed explanation regarding the initial conditions. The choice of the $\beta$ range depends on the application.
>
> In our experimental and theoretical investigation of the fractional versions of the first-order graph ODE models GRAND and CDE, we keep $\beta$ within the range $(0,1]$ so that our model is a diffusion-type differential equation to align with those used in GRAND and CDE. The initial condition is established as $\mathbf{X}(0)=\mathbf{X}$. As the reviewer correctly notes, a $\beta$ value exceeding 1 would shift these models from diffusion to oscillation, which will make comparison with GRAND/CDE difficult. In Section E.5. in our initial submission, we had investigated the performance of oscillatory models via F-GraphCON, a fractional oscillator extension of GraphCON. This extension essentially constitutes an FDE with an order $\le 2$.

---

> > ### Comment · Reviewer_Qi4C · 2023-11-20
> > **Response**
> >
> > Thank you for the response, which clarifies what I understood in the first phase. However it still doesn't answer my question of why not to experiment with higher values of $\beta$.

---

> ### Author Response · Authors · 2023-11-16
> **Weakness 7: Can our framework include GraphCON?**
>
> >Weakness 7: I am not sure it is correct that the model can have global or 'full path' properties if $\beta$ is smaller than 1. For example, I think it is fair to say that if beta is indeed smaller than 1, then a second order process as in GraphCON cannot be represented by the model.
>
>
>
> **Response:** Please see the responses to Weaknesses 5 and 6.

---

> ### Author Response · Authors · 2023-11-16
> **Weakness 8&9: Recommended experiments are already included in appendix.**
>
> >Weakness 8: The experiments indeed show that the proposed method improves compared to baselines produced by the authors, but they are quite narrow and show a partial picture of the current state of the art and existing methods. I would expect that the authors compare their work (experimentally) with other methods like FLODE, CDE, GRAND++, as well as other recent methods like ACMII-GCN++ ('Is Heterophily A Real Nightmare For Graph Neural Networks To Do Node Classification?') or DRew ('DRew: Dynamically Rewired Message Passing with Delay').
>
> >Weakness 9: The authors state that the proposed method can be applied to any ordinary differential equation GNN, so can the authors please also show the results when applied to other baseline methods as discussed in the paper?
>
>
> **Response:** It appears that the reviewer may have overlooked the experiments in our Appendix. In our original submission, we had presented F-GRAND, F-CDE, F-GRAND++, F-GREAD, and F-GraphCON, which are extensions of well-known graph ODE models recognized in the community. The referenced ACMII-GCN++ and DRew, however, are not graph ODE models. To further address the reviewer's concerns, we have included the fractional versions of F-FLODE during this rebuttal phase (see Section E.6 in our revised version).

---

> > ### Comment · Reviewer_Qi4C · 2023-11-20
> > **Response**
> >
> > Thank you for the response. In the first phase of the review I read the full paper including the appendix. I saw the results there, but as I said in my review the experiments are quite narrow and do not show a full picture that compares with the rest of the methods. The results in the appendix are partial in the sense that they always consider only one or two method on a specific benchmark, making it hard to fully understand the contribution of this work.

---

> > ### Comment · Reviewer_Qi4C · 2023-11-20
> > **Regarding DRew**
> >
> > It is true that DRew is not a full fledged ODE based GNN, but it also has (by design) memory properties, that in principle can be seen as a higher order ODE. I am therefore not convinced about the authors disregarding this work, which is also relevant in terms of oversquashing and SOTA performance.

---

> ### Author Response · Authors · 2023-11-16
> **Weakness 10: How to choose $\beta$?**
>
> >Weakness 10: $\beta$ is a hyperparameter. What would happen if you learn it? how will it influence the results and the stability of your solver? Is there a principles way to choosing the hyperparameter?
>
> **Response:** The hyperparameter $\beta$ is found by binary search. A significant challenge in using a learnable $\beta$ arises from the gamma function in the fractional derivative definition:
> \begin{align*}
> \Gamma(\beta)=\int_0^{\infty} t^{\beta-1} e^{-t} d t .
> \end{align*}
> Backpropagating through the gamma function becomes difficult primarily due to the integral. The gradient computations can become problematic and often unstable due to the following possible reasons.
>
> 1. Behavior and Gradient of the Gamma Function: The gamma function has poles (i.e., points of infinity) at certain values of $\beta$. As $\beta$ approaches these poles, the gradients can explode. The derivative of the gamma function with respect to $\beta$ involves the digamma function (the logarithmic derivative of the gamma function). This derivative can introduce rapid changes, especially near the poles of the gamma function, causing the gradients to be unstable.
>
> 2. Numerical Techniques: The gamma function is defined using an integral. Differentiating an integral, especially one that might not have a closed form for its derivative, can introduce numerical inaccuracies. These inaccuracies can become pronounced for certain values of $\beta$. In Pytorch/Tensorflow, we are relying on numerical methods to compute the gradient (e.g., numerical integration). There is potential for inaccuracies that could destabilize training, especially if $\beta$ moves into regions where the function behaves unpredictably.
>
>
> An alternative approach to learning $\beta$ is to utilize a distributed fractional-order differential equation [B1, B2]. Rather than designating a single $\beta$ that necessitates fine-tuning, we employ a learnable measure $\mu$ over a range $[a, b]$ for $\beta$. The distributed fractional-order operator applied to the function $X(t)$, represented as
> \begin{align*}
> \int_a^b D^\beta X(t) \mathrm{d} \mu(\beta),
> \end{align*}
>  can be perceived as the limiting case of a weighted summation over derivatives of multiple orders, denoted as $\sum_i w\left(\beta_i\right) D^{\beta_i} X(t)$. This operator addresses the limitation that a single fractional-order operator $D^\beta$ has limited capacity to accurately model complex physical phenomena despite its greater generality compared to its integer-order counterpart. The distributed fractional-order operator can be conceptualized as an amalgamation of behaviors depicted by distinct fractional-order operators across a particular range. The new approach is characterized by the following graph dynamical equation:
> \begin{align*}
> \int_a^b D^\beta \mathbf{X}(t) \mathrm{d} \mu(\beta)=\mathcal{F}(\mathbf{W}, \mathbf{X}(t)),  \quad  \quad  \quad   \quad \quad \text{[E.1]}
> \end{align*}
> where $[a, b]$ denotes the range of the order $\beta, \mu$ is a learnable measure of $\beta$, and $\mathcal{F}$ is a dynamic operator as in Eq(6) in our paper. The initial step is to approximate Eq[E.1] as follows:
> \begin{align*}
> \sum_{j=0}^n w_{j \mathrm{C}} D^{\beta_j} \mathbf{X}(t)=\mathcal{F}(\mathbf{W}, \mathbf{X}(t)) \quad  \quad  \quad   \quad \quad \text{[E.2]}
> \end{align*}
> where $D^{\beta_j} \in[a, b], j=1,2, \ldots, n$, are distinct interpolation points and $w_j$ are weights associated with the measure $\mu$. Reflecting the learnable nature of $\mu$ over $\beta$, $w_j$ can be directly set to be a learnable parameter in our implementation. The subsequent step is to solve the multi-term fractional differential equation given by [E.2]. As per [A1, Theorem 8.1.], the multi-term fractional differential equation can be converted into a system of equations of single order $D^\gamma$, where $\gamma:=1 / M$ with $M$ being the least common multiple of the denominators of $\beta_1, \beta_2, \ldots, \beta_n$ when they are rational numbers. The classical fractional solvers presented in our paper can resolve the resultant system of single-order equations.
>
> The new model above is out of the scope of our current work and will be investigated in detail in future work.
>
>
> [B1] Ding, Wei, et al. "Applications of distributed-order fractional operators: A review." Entropy 23.1 (2021): 110.
>
> [B2] Diethelm, Kai, and Neville J. Ford. "Numerical analysis for distributed-order differential equations." J. Computational Applied Math. (2009): 96-104.

---

> ### Author Response · Authors · 2023-11-16
> **Weakness 11: Novelty Clarification**
>
> >Weakness 11: I am not certain that the method is novel, as it was also shown in 'Fractional Graph Convolutional Networks (FGCN) for Semi-Supervised Learning'.
>
> **Response:** The referenced paper, 'Fractional Graph Convolutional Networks (FGCN) for Semi-Supervised Learning,' is unrelated to our work. While it employs the fractional power of the Laplacian matrix, denoted as the $L^\alpha$ operator, akin to the fractional operator in FLODE, it differs significantly from graph ODE models and FROND. Notably, this paper does not incorporate any differential equations or the time-fractional derivative in any form.

---

> ### Author Response · Authors · 2023-11-16
> **Weakness 13: More clarification on our approach**
>
> >Weakness 13: A general point - the focus of the paper is the mitigation of oversmoothing. But, also as the authors state, there are many methods that already do it. Then my question is what is the added value of using this mechanism ? Also, another important issue is oversquashing in graphs. Can the authors discuss how and if would the proposed method can help with that issue?
>
> **Response:** We address the issue of oversmoothing from a unique perspective by utilizing fractional differential equations, which allows us not only to tackle this problem but also to extend graph ODE models to their fractional counterpart, the graph FDE models. This approach sets our method apart from others, as it not only targets oversmoothing but also broadens the scope and applicability of graph ODE models, enabling them to function effectively in more complex scenarios. Unlike other methods aimed specifically at addressing oversmoothing, our approach can function as a plug-in for various graph ODE models with ease. For instance, our method can enhance the performance of the GraphCON model, designed to mitigate oversmoothing, across various datasets (refer to Table 22 in our paper). The random walk interpretation of F-GRAND-L also provides a theoretical underpinning to why oversmoothing mitigation is achieved in FROND, with a tight order bound. In particular, the result on algebraic convergence is of significant theoretical interest.
>
> While our paper is not explicitly designed to tackle the oversquashing problem, we note that FROND is a general framework by which any graph ODE model can be plugged into. This implies that a graph ODE model designed to tackle oversquashing can now be put into the FROND framework to simultaneously achieve oversmoothing mitigation. We will explore this possibility in future work.

---

> > ### Comment · Reviewer_Qi4C · 2023-11-20
> > **Response**
> >
> > I appreciate your answer. Indeed it is seen that FROND does not oversmooth but it still remains unclear to me what is the great advantage of the method, as there are already several methods that can avoid oversmoothing (whether they are ODE based or not), and I think that given the discussions of the authors that FROND has memory properties that do not exist in other ODE methods it would be interesting to understand the influence on oversquashing, which is presumably a more of on open challenge in GNNs, as compared to oversmoothing.  It would be interesting to see experiments with the LRGB datasets and some analysis to understand if FROND can help with this issue in GNNs.

---

> ### Author Response · Authors · 2023-11-19
> **any feedback?**
>
> Dear Reviewer Qi4C,
>
> We sincerely appreciate the time and effort you dedicated to reviewing our paper. **We believe there were several misunderstandings during the first review phase, which unfortunately led to a reject score.** After providing clarifications, we hope that the issues have been resolved. Could you kindly let us know if our responses have sufficiently addressed your concerns? We thank you again for your valuable time.
>
> Best Regards,
>
> Authors

---

> > ### Author Response · Authors · 2023-11-20
> >
> > Dear Reviewer Qi4C,
> >
> > As the discussion period draws to a close, we hope that we have successfully addressed all your comments.
> >
> > We deeply appreciate the time and effort you invested in reviewing our paper. We acknowledge that there may have been several misunderstandings during the initial review phase, which regrettably resulted in a reject score. Having provided further clarifications, it is our hope that these issues have been fully resolved. We kindly request your feedback to confirm if our responses have adequately addressed your concerns. Your time and insights are incredibly valuable to us, and we thank you once again for your dedication.
> >
> > Best Regards,
> >
> > The Authors

---

> ### Author Response · Authors · 2023-11-20
> **Response to "why not turn FROND (a fractional differential equation) to a first-order ordinary differential equation"**
>
> > Thanks for the answer. I think that similarly to the discussion provided by the authors that GraphCON can be written as a first order model (similar to other higher order ODEs), it is also applicable to FROND. So I think that in this sense also **FROND can be reduced to a first order model,** making it similar to other methods like FLODE. Can the authors comment on that? Did the authors try to implement an evaluate an equivalent network? is there an actual benefit to the proposed design this way?
>
> **Response:**
> Positive integer-order ODEs can be written as a system of first-order ODEs via the introduction of auxiliary variables. The same **cannot** be done for fractional-order differential equations due to the non-integer nature of the derivative order.
>
> Fractional calculus, the bedrock of FROND, is rooted in the concept of differentiation and integration to arbitrary, non-integer orders. Fractional differential equations can adeptly model processes exhibiting memory or hereditary characteristics unattainable by integer-order equations. Essentially, whereas a first-order ODE characterizes memoryless (Markovian) processes, a fractional differential equation can model processes influenced by their historical states.
>
> We note that F-GRAND, derived from a fractional differential equation, leads to a non-Markovian random walk on graphs. This is in stark contrast to GRAND or FLODE, which is based on a first-order ODE and results in a Markovian walk. This distinction emphasizes a vital point: **fractional differential equations are not simply extensions of integer-order models; they represent a fundamentally different approach, especially in their capacity to capture memory effects and complex dynamics absent in first-order ODEs.**

---

> ### Author Response · Authors · 2023-11-20
> **Response to "Why not try different possible initialization options or higher $\beta$ for the models like GRAND"**
>
> > Also, while your response answers part of my question, it doesn't address **different possible options to initialize** the terms and if you experiment with it.
>
> > Thank you for the response, which clarifies what I understood in the first phase. However it still doesn't answer my question of why not to experiment with higher values of $\beta$
>
>
> **Response:**
> We emphasize FROND is a framework that seeks to generalize a given graph ODE model. There is no standalone model named 'FROND'. This transformation leads to models like GRAND becoming F-GRAND, GraphCON to F-GraphCON, CDE to F-CDE, and FLODE to F-FLODE, among others.
>
> Our key contribution lies in demonstrating how the FROND transformations can lead to improved performance. To ensure a **fair comparison**, we choose suitable intervals for $\beta$ to maintain the same initial conditions as those used in the original models.
>
> In our framework, we can indeed employ higher-order values of $\beta$ in these transformations. For instance, we can let $\beta$ range between 1 and 2, leading to the F(oscillation)-GRAND-nl model:
>
> - GRAND-nl: $\frac{\mathrm{d} \mathbf{X}(t)}{\mathrm{d} t}=(\mathbf{A}(\mathbf{X}(t))-\mathbf{I}) \mathbf{X}(t)$
>
> - F-GRAND-nl: $D_t^\beta \mathbf{X}(t)=(\mathbf{A}(\mathbf{X}(t))-\mathbf{I}) \mathbf{X}(t), \quad 0<\beta \leq 1$
>
> - F(oscillation)-GRAND-nl: $D_t^\beta \mathbf{X}(t)=(\mathbf{A}(\mathbf{X}(t))-\mathbf{I}) \mathbf{X}(t), \quad 1<\beta \leq 2$
>
> _Different from GRAND and F-GRAND that use the initial condition $\mathbf{X}(0)=\mathbf{X}$, F(oscillation)-GRAND-nl is an oscillation-type differential equation and takes initial condition $\mathbf{X}^{\prime}(0)=\mathbf{X}(0)=\mathbf{X}$._  However, if we compare this model to GRAND, it is difficult to conclude if the difference in performance is due to the different initial conditions or the inclusion of fractional derivatives. That is also why in F-GRAND we keep $0<\beta\le 1$ which is the diffusion type equation that share the same initial condition with its counterpart first-order diffusion equation GRAND.
>
> We present some preliminary results of F(oscillation)-GRAND-nl in the following table.
>
>
> |                                | Cora          | Citeseer      | Airport       |
> |--------------------------------|---------------|---------------|---------------|
> | F(oscillation)-GRAND-nl:       | 82.2 ± 1.6    | 72.6 ± 1.3    | 91.7 ± 1.5    |
> | $\beta$ for F(oscillation)-GRAND-nl | 1.1     | 1.5           | 1.1           |
> | F-GRAND-nl                     | 83.2 ± 1.1    | 74.7 ± 1.9    | 96.1 ± 0.7    |
> | $\beta$ for FROND-nl           | 0.9           | 0.9           | 0.1           |
> | GRAND-nl                       | 82.3 ± 1.6    | 70.9 ± 1.0    | 90.9 ± 1.6    |
>
>
> **Table R3:** Comparison between GRAND-nl, F-GRAND-nl, and F(oscillation)-GRAND-nl

---

> ### Author Response · Authors · 2023-11-20
> **Response to "Why not include DRew or other baselines for the experiments in Appendix"**
>
> > Thank you for the response. In the first phase of the review I read the full paper including the appendix. I saw the results there, but as I said in my review the experiments are quite narrow and do not show a full picture that compares with the rest of the methods. The results in the appendix are partial in the sense that they always consider only one or two method on a specific benchmark, making it hard to fully understand the contribution of this work.
>
> > It is true that DRew is not a full fledged ODE based GNN, but it also has (by design) memory properties, that in principle can be seen as a higher order ODE. I am therefore not convinced about the authors disregarding this work, which is also relevant in terms of oversquashing and SOTA performance.
>
>
> **Response:**
> Note that FROND is a framework that seeks to generalize a base graph ODE model.
> It is important to note that the base graph ODE model is typically designed for different types of tasks. For example, CDE and GREAD are tailored for heterophilic graphs, while GRAND++ is optimized for graph learning in scenarios with a limited number of labeled nodes. In our paper, we have included a comprehensive comparison of these graph ODE models and their fractional counterparts across most datasets tested in their respective original works. We emphasize in our paper that "our primary aim is not to achieve state-of-the-art results, but rather to demonstrate the additional effectiveness of the FROND framework when applied to existing graph neural ODE models." **In our latest revision, in response to the reviewer's suggestions, we have expanded the baselines into detailed, extensive tables.** We refer the reviewer to the new Table 20, 22, 24 and 26 in Appendix.
>
> DRew is not a graph ODE model and, as far as we can tell, it cannot be formulated within a graph ODE framework. We are happy to be corrected if we are wrong. Consequently, our FROND framework, which is **specifically tailored for graph ODE models cannot be applied to DRew.** This is a limitation of the FROND framework: it is specifically designed to build upon existing graph neural ODE models. Future work can investigate designing a graph ODE model that can **approximate** DRew. If that is successful, then the FROND framework can be easily applied to this graph ODE model.

---

> ### Author Response · Authors · 2023-11-20
> **Response to "include oversquashing tasks"**
>
> >I appreciate your answer. Indeed it is seen that FROND does not oversmooth but it still remains unclear to me what is the great advantage of the method, as there are already several methods that can avoid oversmoothing (whether they are ODE based or not), and I think that given the discussions of the authors that FROND has memory properties that do not exist in other ODE methods it would be interesting to understand the influence on oversquashing, which is presumably a more of on open challenge in GNNs, as compared to oversmoothing. It would be interesting to see experiments with the LRGB datasets and some analysis to understand if FROND can help with this issue in GNNs.
>
> > I am therefore not convinced about the authors disregarding this work, which is also relevant in terms of oversquashing and SOTA performance.
>
>
> **Response:**
> The advantages of FROND include the following:
> - It can be easily applied to any existing graph ODE model.
> - With the FROND version of the GNN achieves better performance than the original graph ODE model due to its more general formulation and the flexibility to tune the fractional derivative order $\beta$. This is supported by extensive experiments in which we clearly demonstrated that the FROND versions of different graph ODE models outperform their respective original counterparts. This can form an important tool in the repertoire of a GNN practitioner.
> - Numerical experiments have indicated that the FROND version typically achieves **significantly** better oversmoothing mitigation than its corresponding base graph ODE model (cf. Table 14 in Appendix D.7.1).
> For example, the FROND version of FLODE achieves significantly better oversmoothing mitigation than FLODE, which is one of the graph ODE models for oversmoothing mitigation that is published in NeurIPS 2023 very recently. We direct the reviewer's attention to Table R1 in the response to Reviewer 1hAW. On the Airport dataset, it is observed that FLODE achieves an accuracy of 56.93\% with 256 layers. In contrast, F-FLODE significantly outperforms this, achieving an accuracy of 88.74\%. In our humble opinion, **oversmoothing in GNNs is still an open challenge that no one model can claim to have resolved satisfactorily.** FROND is a step towards this direction, as demonstrated by F-GRAND, which we show to have a theoretical algebraic convergence rate.
>
> As FROND is a framework to generalize a given base graph ODE model, its capabilities are dependent on those of the base graph ODE model.
> If the base graph ODE model has built-in oversquashing mitigation capabilities, this should carry over to the FROND version. Note that FROND is not specifically designed to address oversquashing, which is closely related to the graph topology. Indeed, memory properties are not directly related to the oversquashing phenomenon (but may help to improve a model's performance in this regard). A common technique to overcome oversquashing is rewiring, i.e., changing the graph topology.
>
> We have conducted preliminary tests using the GRAND and F-GRAND models on the 'Peptides-func dataset,' which is part of the LRGB dataset series. Our observations indicate that FROND does contribute to improving the performance of GRAND in terms of mitigating oversquashing. However, we observe that F-GRAND does not achieve SOTA results in oversquashing simply because GRAND itself is not SOTA in this regard. It is of interest in future research to study if further insights into developing optimal rewiring strategies can be derived via the memory properties of the FROND framework.
>
>
> | Model | GCN | GCNII | GINE | GatedGCN | GatedGCN+RWSE | GRAND-l | F-GRAND-l | DRew-GCN |
> |-------|-----|-------|------|----------|---------------|---------|-----------|----------|
> | Test AP | 0.5930±0.0023 | 0.5543±0.0078 | 0.5498±0.0079 | 0.5864±0.0077 | 0.6069±0.0035 | 0.5774±0.0063 | 0.6253±0.0015 | 0.6996±0.0076 |
>
>
> **Table R4:** Graph classification on Peptides-func dataset. Performance metric is Average Precision (AP).

---

> > ### Comment · Reviewer_Qi4C · 2023-11-21
> > **Thank you**
> >
> > I thank the authors for the detailed response and adding experimental results on oversqashinng.
> >
> > Your rebuttal has made significant improvements, and therefore I am happy to increase the score to 6. I still think that the paper can be improved by showing results where $\beta$ is higher than 1 or 2.

---

> > > ### Author Response · Authors · 2023-11-22
> > >
> > > Thank you for the thoughtful re-evaluation of our paper and for recognizing the clarifications provided in our responses! Your insightful feedback is greatly appreciated!

---

### Official Review · Reviewer_2jGF · 2023-11-02

**Soundness:** 3 good
**Presentation:** 4 excellent
**Contribution:** 3 good
**Rating:** 8
**Confidence:** 4

**Summary:**

The paper proposed a fractional variation for graph diffusion methods and its derivative methods. The author provides numerical solutions, theoretical support, and experimental data to support their claim that fractional variation performs better than vanilla graph diffusion methods.

**Strengths:**

- The method is novel and is a good direction for exploring graph neural diffusion methods.
- The paper is detailed and easy to read
- The paper has extensive comparisons between methods
- The paper answers the question about its computation cost with detailed experiments in appendix.

Overall, this paper is an updated version of a paper I've reviewed before. The authors have answered all my questions in this version. I think this paper starts from a nice idea and contains all the details required, so I would recommend acceptance.

**Weaknesses:**

N/A

**Questions:**

N/A

---

> ### Author Response · Authors · 2023-11-16
>
> Thank you for your careful review of our paper. We are grateful for your recognition of our novelty.
>
> Furthermore, we would be honored if you could lend your insights during the discussion phase. Your perspective would be invaluable in clarifying any potential misunderstandings about our work and in contributing to a more robust discussion.

---

> > ### Author Response · Authors · 2023-11-20
> > **Invitation for Enhanced Dialogue and Clarification of Key Points in Paper Review**
> >
> > Dear Reviewer 2jGF,
> >
> > We sincerely appreciate the time and expertise you have invested in reviewing our paper. Your support and insightful comments have been invaluable to our work. Recognizing your deep understanding and appreciation of our research, we would be honored if you could **lead the discussion for this paper.**
> >
> > Unfortunately, **there seems to be a significant misunderstanding of our work by Reviewer Qi4C.** We believe we have comprehensively addressed these concerns in our revisions. However, the lack of direct dialogue between authors and reviewers has limited our ability to clarify these misunderstandings fully. Your guidance in this matter would be immensely helpful.
> >
> > Thank you again for your valuable contribution to our work.
> >
> > Warm regards,
> >
> > Authors

---

> > > ### Comment · Reviewer_2jGF · 2023-11-21
> > >
> > > I have read the responces and I am keeping my score. Reviewer Qi4C's comment on that FROND can be seen as an ODE and / or written as a higher order method is actually incorrect. Although numerically it can be downscaled into an ODE, its qualitative behavior should not resemble any ODE, but rather, as proposed by the authors, a fractional order diffusion equation. In terms of experiments, I believe that the authors have provided enough experiments to adequately proven their point.

---

> > > > ### Comment · Reviewer_Qi4C · 2023-11-21
> > > > **Re writing as first order ODE**
> > > >
> > > > This transformation can be done while retaining the interpretation of an ODE, depending on the implementation, as also suggested by the authors (given that beta is an integer). Perhaps we could discuss this in a different occasion.

---

> ### Comment · Area_Chair_n84J · 2023-11-20
> **Respond to authors' rebuttal**
>
> Please, confirm that you have read the author's response and the other reviewers' comments and indicate if you are willing to revise your rating.

---

> ### Author Response · Authors · 2023-11-23
> **Final Clarification on ODE vs. FDE**
>
> > This transformation can be done while retaining the interpretation of an ODE, depending on the implementation, as also suggested by the authors (given that beta is an integer). Perhaps we could discuss this in a different occasion.
>
>
> We are profoundly grateful to all reviewers for their supportive and constructive engagement with our work. The insightful discussions with each reviewer have been invaluable in enhancing the overall quality and depth of our paper.
>
> We would like to provide a final clarification on whether our FROND can be solved or interpreted as an ODE, even with discretization in its implementation. This clarification is intended to further strengthen the justifications of our work and enhance its accessibility for all readers, including those in the future, in line with the spirit of the ICLR open review process.
>
>
>
> **Analytic Perspective:**
>
> Firstly, from an analytic standpoint, it is important to note that transforming a fractional differential equation (FDE) into an ordinary differential equation (ODE) of integer order is not feasible in general. The focus of the FDE community on developing specialized numerical solvers for FDEs over the past two decades, rather than relying on ODE solvers, underscores the challenges associated with such transformation techniques. A primary reason for this is that the Caputo fractional derivative lacks the semi-group property [C1, Section 3.1], unlike integer order derivatives. In other words, it is incorrect to assert that $D^{\beta_1+\beta_2} f = D^{\beta_1} (D^{\beta_2} f)$ when $\beta_1$ and $\beta_2$ are not integers. Our thorough review of fractional differential equation literature includes key observations from the paper [C2]: "Only with this structure many questions can be asked". It also notes that "Obviously, solving this kind of equations with such a general formulation is impossible, unless more properties on the structure are required." These include linearity and the vanishing at certain points. None of our models, including F-GRAND, F-CDE, F-GRAND, and others, satisfy these stringent conditions.
>
> **Numerical Discretization Perspective:**
>
> Thank you to Reviewer 7EpT for their suggestions. We have moved some implementation details from Appendix C to Section 3.3 and invite readers to examine them. Figure 1 and equation (18) illustrate the fractional explicit Adams–Bashforth–Moulton Basic Predictor under numerical discretization. This can be seen as a fractional Euler method, an extension of the Euler method used in ODE solver implementation. We note that the basic predictor in our framework includes nontrivial dense connections to $\left(\mathbf{X}(0), \ldots, \mathbf{X}\left(t_{n-1}\right)\right)$. In contrast, the ODE Euler solver includes only a direct connection to the preceding layer $\mathbf{X}\left(t_{n-1}\right)$. We also introduce the $K$ short memory solver, an approximation of the basic predictor, which utilizes skipping connections to $K$ preceding layers $\left(\mathbf{X}(t_{n-K}), \ldots, \mathbf{X}\left(t_{n-1}\right)\right)$ plus $\mathbf{X}\left(0\right)$.
>
> In GNN literature, skipping connections have been used to enhance model performance with increased layer depth. It remains unclear whether performance degradation with depth is primarily due to oversmoothing or the classical vanishing gradient problem. In our paper, from a continuous analog perspective, we conclude that oversmoothing persists and impairs model performance with only skip connections. Our theoretical analysis suggests that nontrivial dense connections between layers may mitigate the shift from exponential to slow algebraic convergence, a phenomenon empirically observed to aid in oversmoothing mitigation in works like [C3, C4].
>
>
> [C1] Kai Diethelm. The analysis of fractional differential equations: an application-oriented exposition using differential operators of Caputo type, volume 2004. Springer, 2010.
>
> [C2] Labora, D. C., and Rodriguez-Lopez, R. (2017). From fractional order equations to integer order equations. Fractional Calculus and Applied Analysis, 20(6), 1405-1423.
>
> [C3] Xu, Keyulu, et al. "Representation learning on graphs with jumping knowledge networks." International conference on machine learning. PMLR, 2018.
>
> [C4] Li, Guohao, et al. "Deepergcn: All you need to train deeper gcns." arXiv preprint arXiv:2006.07739 (2020).

---

### Meta-Review · Area_Chair_n84J · 2023-12-07

**Metareview:**

Summary:

The paper introduces the Fractional-Order graph Neural Dynamical network (FROND), a novel learning framework that enhances traditional graph neural ordinary differential equation (ODE) models by integrating the time-fractional Caputo derivative. This incorporation allows FROND to capture long-term memories in feature updating due to the non-local nature of fractional calculus, addressing the limitation of Markovian updates in existing graph neural ODE models and promising improved graph representation learning. The authors provide ample theory and several experiments to show the benefit of adding the fractional derivative component to GNNs.

Strengths:

- The method is novel and is a good direction for exploring graph neural diffusion methods.
- The paper is detailed and easy to read
- The paper has extensive comparisons between methods
- The paper answers the question about its computation cost with detailed experiments in appendix.
- The paper is well written. It was easy to follow and understand.
- The authors show how the proposed method can encapsulate existing models such as GRAND or GraphCON.
- The experiments show that adding a fractional derivative is useful.
- This is the first approach to directly generalize graph neural ODE to fractional derivatives and demonstrate its applicability in real-world datasets.
- The framework is general enough to be incorporated to a wide range of existing graph neural ODE in the literature, such as GRAND, GRAND++, GREAD, etc.
- Empirical study conducted is extensive and results are explained comprehensively. Showing competitiveness of the new framework over existing ones in many different dimensions.
- The paper is well-written, providing a clear and straightforward presentation of the content, which enhances the overall readability.
- The innovative integration of time-fractional derivatives into traditional graph ODEs is a novel approach that effectively addresses key issues like non-local interactions and over-smoothing.
- The proposal is supported by theoretical motivations.
- An extensive evaluation of the framework is presented, demonstrating its effectiveness and versatility across various settings and providing substantial empirical evidence of its performance.

Weaknesses:

- Some missing literature references.
- The experiments are quite narrow and show a partial picture of the current state of the art and existing methods.
- It would be more complete to have a discussion of this increased cost, if there are any, as well as techniques used to overcome it. This is crucial in scaling the approach to larger datasets.

Recommendation:

All reviewers vote for acceptance. I, therefore, recommend acccepting the paper and encourage the authors to use the feedback provided to improve the paper for the camera ready version.

**Justification For Why Not Higher Score:**

N/A

**Justification For Why Not Lower Score:**

Several reviewers provide very high ratings. The number of significant weaknesses identified by the reviewers is very low.

---

### Decision · Program_Chairs · 2024-01-16

Accept (spotlight)